**EMBO** *reports*

# Cerebral organoids expressing mutant actin genes reveal cellular mechanism underlying microcephaly

Indra Niehaus [1,2], Michaela Wilsch-Bräuninger [3], Felipe Mora-Bermúdez[3], Fabian Rost [4],
Mihaela Bobic-Rasonja[5], Velena Radosevic[6], Marija Milkovic-Perisa[7], Pauline Wimberger [8],
Mariasavina Severino [9], Alexandra Haase [10], Ulrich Martin [10], Karolina Kuenzel[11], Kaomei Guan [11],
Katrin Neumann[12], Noreen Walker[3], Evelin Schröck[1], Natasa Jovanov-Milosevic [5✉],
Wieland B Huttner [3,14✉], Nataliya Di Donato [1,2,14✉] & Michael Heide [3,13,14✉]

## Abstract

Actins are cytoskeletal proteins that are essential for multiple cellular processes. Mutations in the ACTB and ACTG1 genes, encoding the ubiquitous beta- and gamma-cytoskeletal actin isoforms, respectively, cause a broad spectrum of neurodevelopmental disorders, with microcephaly as the most frequent one. To investigate the pathogenesis underlying this cortical malformation, we studied patient-derived cerebral organoids from induced pluripotent stem cells of individuals with the Baraitser–Winter-CerebroFrontoFacial syndrome (BWCFF-S) carrying an ACTB/ACTG1 missense mutation. These organoids were reduced in size, showing a thinner ventricular zone (VZ) due to reduced VZ progenitor abundance. Strikingly, VZ progenitors in BWCFF-S cerebral organoids displayed a shift in the orientation of their cleavage plane from a predominantly vertical to a majoritarian horizontal orientation. The latter cleavage plane orientation is incompatible with increasing VZ progenitor abundance and instead promotes basal progenitor generation. Various cytoskeletal and morphological irregularities of BWCFF-S VZ progenitors, notably in the apical region, seemingly contribute to this change in cleavage plane orientation. Our results provide insight into the cell biological basis of the microcephaly associated with BWCFF-S caused by actin mutations.

**Keywords** Cerebral Organoids; Actin; Disease Modeling; Mitotic Spindle; Microcephaly
**Subject Categories** Cell Adhesion, Polarity & Cytoskeleton; Neuroscience; Stem Cells & Regenerative Medicine

## Introduction

Actin is an abundantly expressed protein found in virtually all eukaryotic cells, with multiple functions in critical cellular processes (Perrin and Ervasti, 2010; Pollard and Cooper, 2009). While it is often regarded as just "actin", human actin consists of six actin isoforms encoded by six genes with tissue-specific expression, *ACTA1* ($\alpha_{skeletal}$-actin), *ACTA2* ($\alpha_{smooth}$-actin), *ACTB* ($\beta_{cytoskeletal}$-actin, $\beta$CYA), *ACTC1* ($\alpha_{cardiac}$-actin), *ACTG1* ($\gamma_{cytoskeletal}$-actin, $\gamma$CYA), and *ACTG2* ($\gamma_{smooth}$-actin) (Vandekerckhove and Weber, 1978).

Pathogenic variants in the genes encoding for the muscle actin isoforms, *ACTA1, ACTA2, ACTC1*, and *ACTG2* result in different forms of either skeletal and visceral myopathies or vasculopathies (Parker et al, 2020). Pathogenic variants in the genes *ACTB* and *ACTG1*, encoding the ubiquitously expressed $\beta$CYA and $\gamma$CYA, are associated with the systemic Mendelian disorder called Baraitser–Winter-CerebroFrontoFacial syndrome (BWCFF-S) (Riviere et al, 2012; Verloes et al, 2015) (Appendix Fig. S1). BWCFF-S is characterized by a wide spectrum of variable congenital anomalies, with the central nervous system frequently being affected. Nearly all patients show intellectual impairment (Verloes et al, 2015; Cuvertino et al, 2017; Latham et al, 2018), and almost half of the BWCFF-S patients exhibit cortical malformations, with microcephaly and lissencephaly as the most common forms (Di Donato et al, 2016; Shitamukai and Matsuzaki, 2012).

[1]Institute for Clinical Genetics, Technische Universität Dresden, and National Cancer Center (NCT) Dresden, Dresden 01307, Germany. [2]Department of Human Genetics, Hannover Medical School, Hannover 30626, Germany. [3]Max Planck Institute of Molecular Cell Biology and Genetics, Dresden 01307, Germany. [4]Technische Universität Dresden, DRESDEN-Concept Genome Center, Center for Molecular and Cellular Bioengineering (CMCB), Dresden 01307, Germany. [5]Croatian Institute of Brain Research and Department of Biology, School of Medicine, University of Zagreb, Zagreb 10000, Croatia. [6]Department of Obstetrics and Gynecology, University Hospital Centre, Zagreb 10000, Croatia. [7]Department of Pathology and Cytology, School of Medicine, University of Zagreb and University Hospital Centre, Zagreb 10000, Croatia. [8]Department of Gynecology and Obstetrics, Technische Universität Dresden, and National Cancer Center (NCT) Dresden, Dresden 01307, Germany. [9]Neuroradiology Unit, IRCCS Istituto Giannina Gaslini, Genova 16147, Italy. [10]Leibniz Research Laboratories for Biotechnology and Artificial Organs (LEBAO), Department of Cardiac, Thoracic, Transplantation, and Vascular Surgery, REBIRTH – Research Center for Translational Regenerative Medicine, Hannover Medical School, Hannover 30625, Germany. [11]Institute of Pharmacology and Toxicology, Technische Universität Dresden, 01307 Dresden, Germany. [12]Center for Regenerative Therapies Dresden (CRTD, Stem Cell Engineering Facility), Technische Universität Dresden, Dresden 01307, Germany. [13]German Primate Center, Leibniz Institute for Primate Research, Göttingen 37077, Germany. [14]These authors contributed equally: Wieland B Huttner, Nataliya Di Donato, Michael Heide. ✉E-mail: njovanov@hiim.hr; huttner@mpi-cbg.de; didonato.nataliya@mh-hannover.de; mheide@dpz.eu

Currently, the available neuropathological data of BWCFF-S patients are limited to three reports, which however lack specific data on the cytoarchitecture of the BWCFF-S neocortex (Forman et al, 2005; Poirier et al, 2015; Vontell et al, 2019). Moreover, the developmental mechanisms underlying the BWCFF-S-associated cortical malformations are unknown.

BWCFF-S disease models could provide functional insights into the role of the above-mentioned two CYA isoforms during pathophysiological neocortex development. However, the available mouse *Actb* knock-out (KO) and *Actg1* KO models (Cheever and Ervasti, 2013; Belyantseva et al, 2009) do not faithfully reproduce the BWCFF-S-associated cortical malformations, which —besides the difference between an actin KO and expression of a mutant actin—is not surprising in light of the differences in cortical development between mouse and human. Therefore, other models of human corticogenesis are needed that better recapitulate human physiological and pathophysiological neocortex development. Recently, a promising model for this emerged in the form of brain organoids (Kadoshima et al, 2013; Lancaster et al, 2013). Brain organoids are organized three-dimensional cell aggregates generated from pluripotent stem cells that recapitulate many features of the developing brain (including the neocortex) (Lancaster et al, 2017; Kelava and Lancaster, 2016; Di Lullo and Kriegstein, 2017; Heide et al, 2018; Arlotta, 2018; Pasca et al, 2022). Moreover, several studies showed the usefulness of brain organoids in the modeling and functional characterization of cortical malformations (Lancaster et al, 2013; Gabriel et al, 2016; Li et al, 2017; Zhang et al, 2019; Esk et al, 2020; Iefremova et al, 2017; Bershteyn et al, 2017). Human brain organoids carrying pathogenic variants of the actin genes associated with BWCFF-S would have the potential to overcome the limitations of rodent models and could provide insight into the underlying mechanism of the formation of BWCFF-S-associated cortical malformations.

In this study, we generated patient-derived induced pluripotent stem cells (iPSCs) carrying a pathogenic gene variant of either *ACTB* or *ACTG1*, and used these to grow human cerebral organoids, a specific type of brain organoids, in order to model the BWCFF-S–associated microcephaly. For this purpose, we first confirmed that human cerebral organoids in essence recapitulate the widespread distribution of $\beta$CYA and $\gamma$CYA in fetal human cerebral cortex and can be used as a model for actinopathies, such as BWCFF-S. We then compared cerebral organoids derived from two different healthy control iPSC lines to those derived from the two BWCFF-S patient iPSC lines. We found that BWCFF-S-derived cerebral organoids exhibit a smaller size and an abnormal VZ morphology. Moreover, mitotic apical progenitors (APs; for the terminology of APs vs. VZ progenitors, see Methods) in the VZ shifted the cleavage plane from a predominantly vertical (control organoids) to a nearly randomized or majoritarian horizontal (BWCFF-S organoids) orientation. The latter is known to prevent the symmetric proliferative divisions of APs that are required to increase their abundance (Huttner and Kosodo, 2005; Mora-Bermudez and Huttner, 2015; di Pietro et al, 2016). Using transmission electron microscopy, we detected various cytoskeletal and morphological irregularities in VZ progenitors of BWCFF-S cerebral organoids, notably in the apical region of these cells. These irregularities presumably contribute, in a causative manner, to the abnormal cleavage plane orientation of mitotic APs in BWCFF-S organoids. In conclusion, our data suggest that the underlying mechanism of BWCFF-S-associated microcephaly is a nearly randomized or majoritarian horizontal cleavage plane orientation of mitotic APs, which likely is caused by various cytoskeletal and morphological irregularities in these cells.

# Results

## Cortical malformations in BWCFF-S patients

To broaden our view of the cortical malformations found in BWCFF-S patients, we reviewed the literature on the neurological manifestations in patients carrying pathogenic variants (or likely pathogenic variants) of *ACTB* and *ACTG1* genes (Riviere et al, 2012; Verloes et al, 2015; Di Donato et al, 2016; Di Donato et al, 2014; Accogli et al, 2020; Eker et al, 2014; Rossi et al, 2003). We combined this literature review with an analysis of brain imaging data of 29 not previously reported patients carrying pathogenic variants of *ACTB* (20 patients) and *ACTG1* (nine patients). We confirmed the previous observation (Verloes et al, 2015) that malformations of cortical development were present in almost 70% of BWCFF-S patients. Figure 1 shows representative brain MRI images illustrating malformations typically associated with the BWCFF-S in comparison to the images of a healthy individual (Fig. 1A–C). Two of the patients (Fig. 1D–F,J–L) each carry a point mutation in *ACTB* which results in a single amino acid substitution, of which the BWCFF-S patient with the actin variant NM_001101.5 (*ACTB*):c.359 C > T p.Thr120Ile (Fig. 1D–F) will serve as source for one set of the cerebral organoids studied here (see below). The other two patients (Fig. 1G–I,M–O) each carry a point mutation in *ACTG1* which also results in a single amino acid substitution, of which the BWCFF-S patient with the actin variant NM_001614.5(*ACTG1*):c.608 C > T p.Thr203Met (Fig. 1G–I) will serve as source for another set of the cerebral organoids studied here (see below).

The patient with the *ACTB* variant p.Thr120Ille was reported previously (Di Donato et al, 2014) and had microcephaly with a head circumference of –2.5 SD and severe intellectual disability with an IQ < 40. Briefly, the brain malformations in this patient included frontal predominant pachygyria (Fig. 1E) and periventricular nodular heterotopia located bilaterally in the occipital horns of the lateral ventricles (Fig. 1F). The patient carrying the variant *ACTG1*:p.Thr203Met was non-ambulant, showed severe microcephaly of –5.2 SD, and had no speech development with an IQ < 40. Brain MRI demonstrated frontotemporal pachygyria (Fig. 1H,I) accompanied by agenesis of the corpus callosum and cerebellar hypoplasia (Fig. 1G). Figure 1J–L shows a brain MRI from a second patient carrying a pathogenic *ACTB* variant, NM_001101.5(*ACTB*):c.571 A > G, p.Lys191Glu. This patient had microcephaly of –3.5 SD, and the brain MRI showed bilateral anterior-predominant pachygyria, with very prominent perivascular spaces (Fig. 1K). A second patient carrying a pathogenic *ACTG1* variant is presented in Fig. 1M–O. This patient was previously reported (Accogli et al, 2020) and carries the mutation NM_001614.5(*ACTG1*):c.767 G > A, p.Arg256Gln, located at a mutational hotspot. Briefly, the brain MRI images of this patient demonstrated partial agenesis of the corpus callosum (Fig. 1M) and a very specific pattern of cortical malformations consistent with

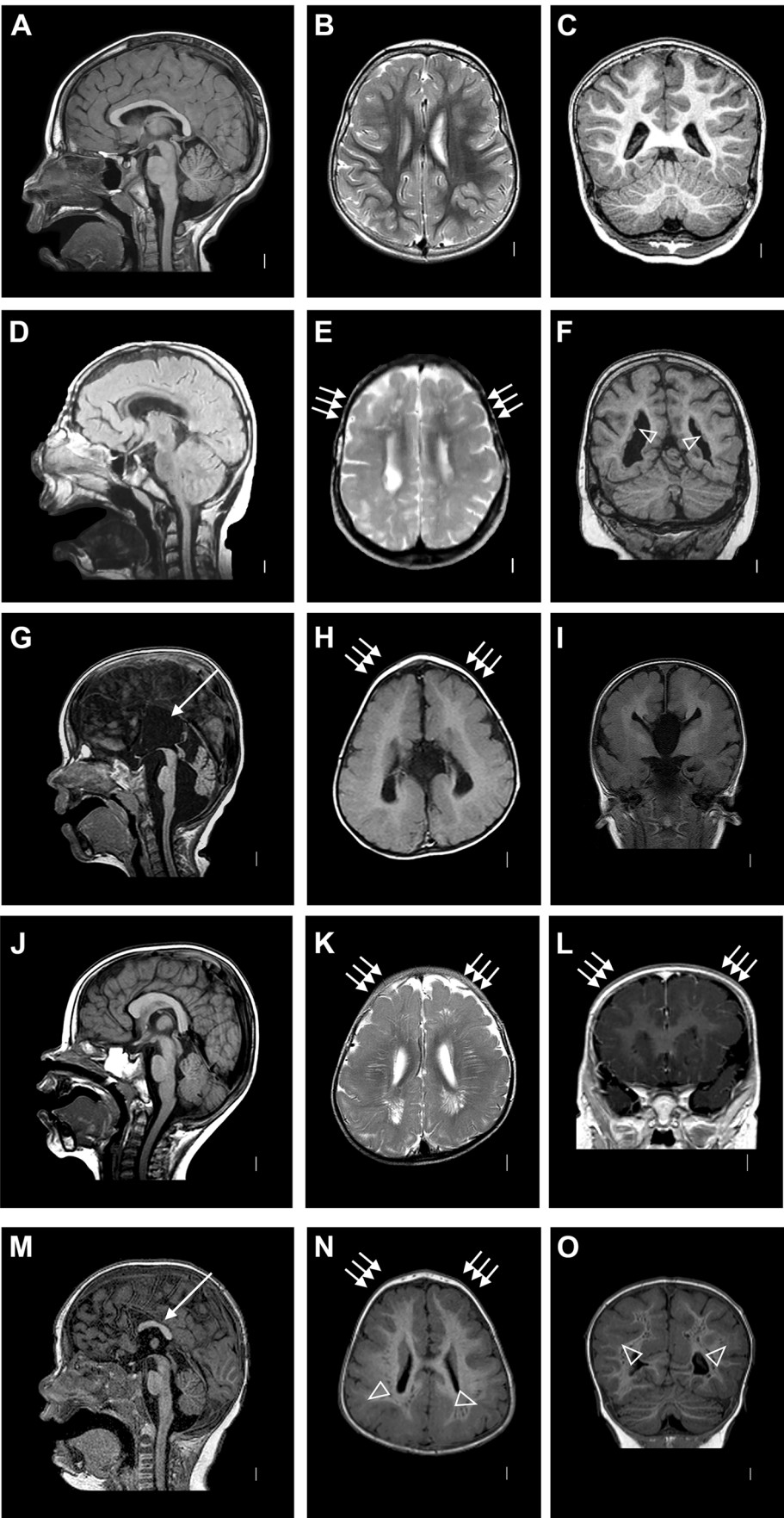

**Figure 1. Representative cases of malformations of cortical development in Baraitser–Winter-Cerebrofrontofacial Syndrome.**

(A–C) Representative MRI scans of a healthy individual at the age of 6 years; (A, B) reproduced from Brock et al, 2018 under CC BY 4.0. (D–F) MRI scans of a patient with the de novo variant in *ACTB* NM_001101.5:c.359 C > T p.Thr120Ile, at the age of 10 years. (D) Midline sagittal FLAIR image shows a thin corpus callosum; note that the thin appearance of the brainstem most likely is a positioning artifact. (E) T2-weighted axial image shows thick and less folded cortex (arrows); and (F) coronal T1 image demonstrates bilateral periventricular nodular heterotopia (arrowheads) in the posterior horns of the lateral ventricles. (G–I) MRI scans of a patient with the de novo variant in *ACTG1* NM_001614.5:c.608 C > T p.Thr203Met at the age of 1 year and 8 months. (G) Midline sagittal T1-weighted image shows complete agenesis of the corpus callosum (arrow) and hypoplasia of the cerebellar vermis; axial (H) and coronal (I) T1 images show frontal predominant pachygyria (arrows in (H)). (J–L) MRI scans of a patient with the de novo variant in *ACTB* NM_001101.5:c.571 A > G p.Lys191Glu at the age of 1 year and 1 month. (J) Midline sagittal T1-weighted image shows normal brain morphology; T2-weighted axial (K) and T1-weighted coronal (L) images show frontal predominant pachygyria (arrows). (M–O) MRI scans of a patient with the de novo variant in *ACTG1* NM_001614.5:c.767 G > A p.Arg256Gln at the age of 1 year and 5 months (patient S12 (30)). Midline sagittal T1-weighted image (M) shows partial agenesis of the corpus callosum (arrow); (N) axial image at the level of the lateral ventricles shows frontal pachygyria (arrows) and thin posterior subcortical band heterotopia (arrowheads); band heterotopia is more easily recognizable in the coronal image (arrowheads) (O).

frontal predominant pachygyria associated with a thin posterior band heterotopia (Fig. 1N,O).

## Widespread distribution of βCYA and γCYA in human cerebral organoids, similar to fetal human neocortex

Prior to generating cerebral organoids from BWCFF-S patient-derived iPSCs, we first determined whether the expression patterns of βCYA and γCYA in human cerebral organoids generated from control iPSCs would be similar, in principle, to those in fetal human neocortex, which we considered to be a prerequisite for using cerebral organoids as a model for BWCFF-S. As shown by immunohistochemistry in Appendix Figs. S2 and S3 for 12 and 16 post conception weeks (pcw) fetal human neocortex tissue, respectively, both βCYA and γCYA showed a widespread distribution across the developing cortical wall, from the VZ all the way to the marginal zone, with the peak of immunoreactivity in the apical-most region of the VZ, which is known to harbor the apical adherens junction (AJ) belt.

To investigate whether cerebral organoids show similar widespread βCYA and γCYA expression patterns as observed in fetal human neocortex tissue, we generated cerebral organoids from two human control iPSC lines, SC102A and CRTDi011-A. Immunohistochemistry of these organoids at day 30 of culture revealed an, in principle, similar widespread distribution of both βCYA and γCYA across the wall of the ventricle-like structures, from the ventricular surface to the outer surface (Appendix Fig. S4, c1 and c2). These ventricle-like structures are fluid-filled cavities surrounded by layers of cells within a given brain organoid, corresponding to the in vivo brain ventricles. The cellular layers surrounding these structures in cerebral organoids include a VZ and a subventricular zone (SVZ)/neuronal layer (NL), roughly corresponding to the different zones and layers of the developing cortical wall. As in fetal tissue, within the VZ of the organoids, βCYA and γCYA immunoreactivity were concentrated towards the apical surface.

Given this widespread distribution of both βCYA and γCYA in the control cerebral organoids, we checked the expression of canonical markers in these organoids under the present conditions, again using immunohistochemistry. Consistent with previous reports (Kadoshima et al, 2013; Lancaster et al, 2013; Lancaster and Knoblich, 2014), the control cerebral organoids after 30 days in culture were found to express markers of proliferating progenitors (SOX2), neurogenic basal progenitors (TBR2), and newborn neurons (Tuj1) including deep-layer neurons (Ctip2) (Appendix Figs. S5–S7, c1 and c2). Immunostaining for nestin revealed the

fibers of radially organized progenitors in the VZ, with densely packed nuclei as seen by DAPI staining or SOX2 immunostaining (Appendix Fig. S5, c1 and c2).

## Generation of cerebral organoids from BWCFF-S patient-derived iPSCs

Next, we sought to model BWCFF-S in cerebral organoids. To this end, we collected fibroblasts from two BWCFF-S patients, carrying either the actin variant NM_001101.5(*ACTB*):c.359 C > T p.Thr120Ile (Fig. 1D–F) or the actin variant NM_001614.5(*ACTG1*):c.608 C > T p.Thr203Met (Fig. 1G–I), and reprogrammed them into iPSCs. The two sets of iPSCs exhibited normal karyotype and expressed typical pluripotency markers (Appendix Figs. S8 and S9). We generated cerebral organoids from two clones of each patient-derived iPSC line.

Next, we explored whether cerebral organoids generated from the BWCFF-S patient-derived iPSCs showed a similar widespread βCYA and γCYA distribution and canonical marker expression as the control organoids and hence could be suitable models to study the disrupted early cortical development characteristic of such patients. Indeed, immunohistochemistry at day 30 of culture revealed that both types of organoids, expressing either the *ACTB* variant or the *ACTG1* variant, showed a similar widespread distribution of both βCYA and γCYA across the various zones of the wall of the ventricle-like structures (Appendix Fig. S4). For the *ACTG1* variant-expressing organoids, a concentration of βCYA and γCYA immunoreactivity at the apical surface of the VZ could be observed. Moreover, using western blot analysis, we found that both βCYA and γCYA have a similar relative abundance in BWCFF-S compared to control cerebral organoids (Appendix Fig. S10A,B). Also, phalloidin staining showed no apparent effects on F-actin in the BCWFF-S condition compared to control (Appendix Fig. S10C). Like control organoids, both the *ACTB* or *ACTG1* variant-expressing organoids showed SOX2, TBR2, Tuj1, NeuN, Ctip2, and nestin (Appendix Figs. S5–S7) immunoreactivity in the typical locations. We conclude that the cerebral organoids generated from the BWCFF-S patient-derived iPSCs can be suitable models to study the microcephaly observed in these patients.

## BWCFF-S cerebral organoids are significantly reduced in size

In the above-described immunostainings, we had noticed that the ventricle-like structures in the BWCFF-S organoids appeared to be

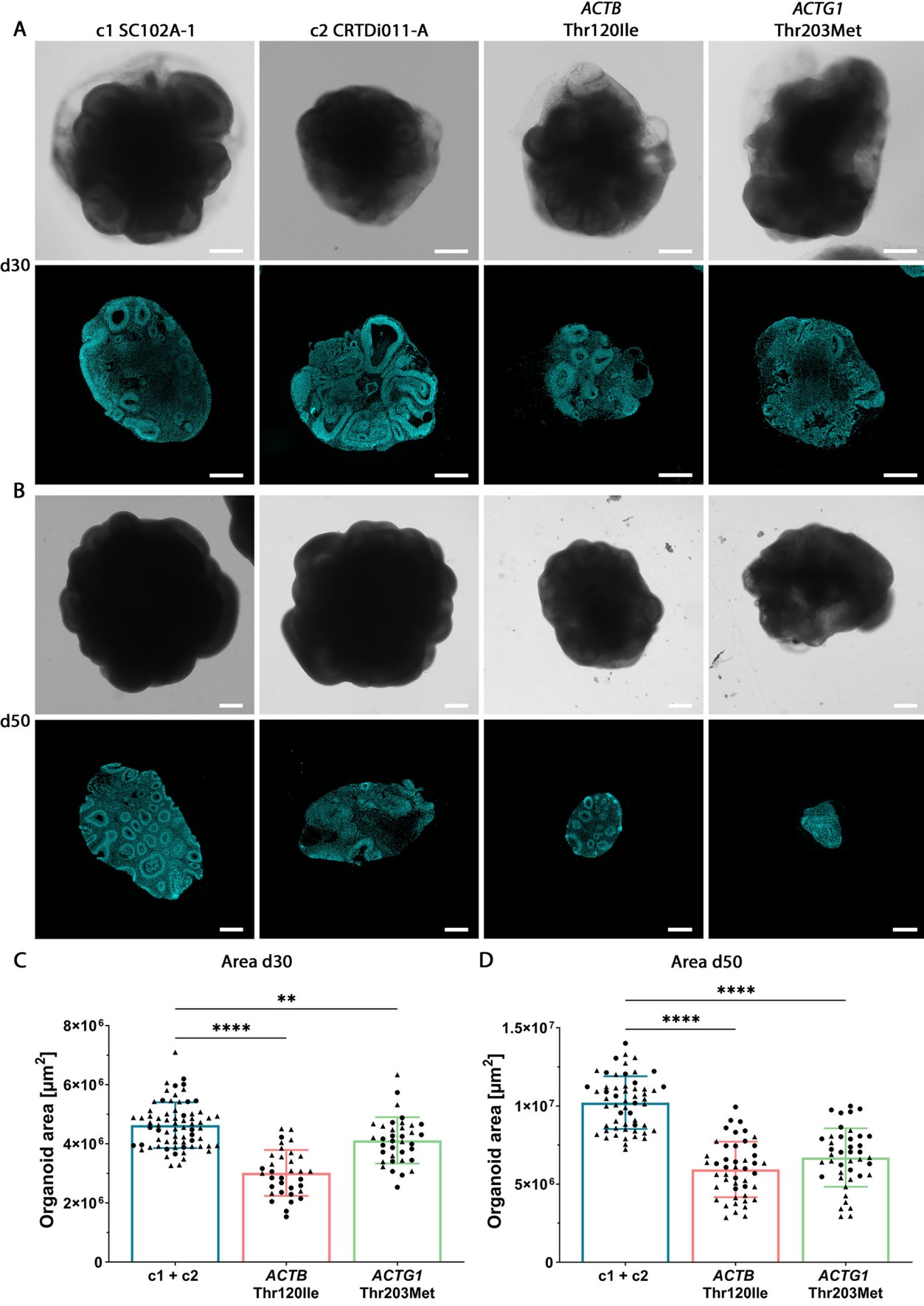

**Figure** (A) c1 SC102A-1, c2 CRTDi011-A, *ACTB* Thr120Ile, *ACTG1* Thr203Met at d30; (B) d50; (C) Area d30; (D) Area d50.

**Figure 2.  BWCFF-S cerebral organoids show a reduction in size at 30 and 50 days of culture.**

(A, B) Bright-field images (top rows in (A, B)) and DAPI-stained sections (bottom rows in (A, B)) of 30-day-old (A) and 50-day-old (B) cerebral organoids generated from two control iPSC lines (c1, SC102A-1 and c2, CRTDi011-A; left two columns), from BWCFF-S *ACTB* Thr120Ile patient-derived iPSCs (second column from right), and from BWCFF-S *ACTG1* Thr203Met patient-derived iPSCs (right column). Scale bars, 500 µm. (C, D) Quantification of the organoid area in control (c1, SC102A-1 and c2, CRTDi011-A; blue bars), BWCFF-S *ACTB* Thr120Ile (red bars), and BWCFF-S *ACTG1* Thr203Met (green bars) cerebral organoid sections at culture day 30 (C) and day 50 (D). (C) Data are the mean of 70 control (generated from two different iPSC lines; indicated by circles (c1, SC102A-1) and triangles (c2, CRTDi011-A)), 34 BWCFF-S *ACTB* Thr120Ile (generated from two different iPSC clones; indicated by circles and triangles) and 36 BWCFF-S *ACTG1* Thr203Met (generated from two different iPSC clones; indicated by circles and triangles) 30 days-old cerebral organoids of two to eight independent batches; error bars indicate SD; statistical significance was determined by one-way ANOVA (c1 + c2 vs. *ACTB* Thr120Ile: ****$P < 0.0001$; c1 + c2 vs. *ACTG1* Thr203Met: **$P = 0.0046$). (D) Data are the mean of 56 control (generated from two different iPSC lines; indicated by circles and triangles as in (C)), 46 BWCFF-S *ACTB* Thr120Ile (generated from two different iPSC clones; indicated by circles and triangles) and 42 BWCFF-S *ACTG1* Thr203Met (generated from two different iPSC clones; indicated by circles and triangles) 50-day-old cerebral organoids of 2–8 independent batches; error bars indicate SD; statistical significance was determined by one-way ANOVA (c1 + c2 vs. *ACTB* Thr120Ile: ****$P < 0.0001$; c1 + c2 vs. *ACTG1* Thr203Met: ****$P < 0.0001$). Source data are available online for this figure.

smaller than those in the control organoids. We therefore decided to quantitatively compare various size parameters of the BWCFF-S organoids vs. control organoids. As the first step in the analyses, we quantified the entire size of the cerebral organoids. At 30 days of cerebral organoid culture, the two types of BWCFF-S organoids were significantly smaller compared to control organoids (Fig. 2A,C). Moreover, at 50 days of organoid culture, a similar size difference was observed (Fig. 2B,D), suggesting that both the control and the two types of BWCFF-S cerebral organoids were growing at a similar speed between day 30 and day 50 of culture. These data are in line with the microcephaly observed in the two patients and in at least half of the whole BWCFF-S cohort (Verloes et al, 2015), and indicate that BWCFF-S cerebral organoids can recapitulate features of the patients' phenotype.

### Single-cell RNA-seq analysis reveals abnormal cell type composition in BWCFF-S cerebral organoids

To better understand the size reduction observed in BWCFF-S cerebral organoids, we performed single-cell RNA sequencing (scRNA-seq) on patient-derived organoids and control organoids at 30 and 50 days of culture. In addition, we analyzed cerebral organoids derived from control CRTDi011-A iPSCs, in which we introduced the *ACTB* Thr120Ile mutation using CRISPR/Cas9. In the following, we first focus on the patient-derived BWCFF-S cerebral organoids before turning to the CRISPR/Cas9-edited organoids later in the results.

Our scRNA-seq analysis identified 17 distinct clusters, representing different progenitor and neuronal populations but also mesenchymal-like, neural crest, and neuroectodermal-like cells expressing characteristic markers (Figs. 3A and EV1A,B; Appendix Fig. 11). Moreover, progenitor and neuronal populations could be distinguished based on their cell cycle phase, with neuronal clusters enriched in G1/G0 phase cells, while progenitor clusters were enriched in S/G2/M phase cells (Fig. 3B). To better understand the cell type composition, we combined clusters of the same cell types leading into six categories: radial glia (RG), basal progenitors (BPs), neurons, neuroectodermal-like cells, neural crest and mesenchymal-like cells (Figs. 3C and EV1C). Using this approach, we found that the percentage of RG relative to total cells was reduced in BWCFF-S cerebral organoids at 30 days of culture compared to controls (Fig. 3C). Moreover, we found an increased percentage of BPs in BWCFF-S *ACTB* Thr120Ile patient-derived cerebral organoids compared to controls. In BWCFF-S *ACTG1*

Thr203Met organoids, the BP percentage was comparable to controls. However, it is important to note that neural crest and neuroectodermal-like cells are overrepresented in BWCFF-S *ACTG1* Thr203Met organoids (Fig. 3C). To better assess BP output, we calculated the BP/RG ratio and found a strong increase in this parameter in day 30 BWCFF-S cerebral organoids, indicating an elevated BP output (Fig. 3D).

On day 50 of cerebral organoid culture, the relative contribution of RG in both control and BWCFF-S organoids was reduced, while the percentage of neurons increased (Figs. 3C and EV1C), in line with the neuron production between day 30 and 50. Interestingly, while there was a clear reduction in the proportion of neurons in BWCFF-S *ACTG1* Thr203Met organoids, BWCFF-S *ACTB* Thr120Ile patient-derived cerebral organoids showed a comparable proportion of neurons to the control (Fig. EV1C). However, two points need to be considered in this context. First, when examining individual neuronal populations, the relative contribution of dorsal neurons was reduced in both BWCFF-S *ACTG1* Thr203Met as well as *ACTB* Thr120Ile cerebral organoids (Fig. EV1B). Second, due to the strong size reduction of BWCFF-S cerebral organoids (Fig. 2B,D), even if the relative contribution of neurons was similar in control and BWCFF-S organoids (Fig. EV1C), total neuron numbers would still be reduced in BWCFF-S cerebral organoids. In summary, the scRNA-seq data indicate a strong effect of the BWCFF-S mutations on RG, affecting BPs and subsequently neurons.

### BWCFF-S cerebral organoids exhibit a massive reduction in VZ progenitors

Due to the strong reduction in the RG proportion in BWCFF-S organoids compared to control in our scRNA-seq data, we next analyzed the ventricle-like structures of these organoids, as they represent the compartment of apical radial glia as the primary RG and VZ progenitor type. These analyses revealed that the size of the ventricle-like structures and the morphology of the VZ in the two types of BWCFF-S organoids differed significantly from the controls (Fig. 4). To better characterize this abnormal morphology, we quantified different parameters of the VZ in the control and the BWCFF-S cerebral organoids at 30 days of organoid culture, as already at this time point, the size difference was obvious. For this purpose, we used DAPI-stained control and BWCFF-S cerebral organoids and first determined the radial thickness of the VZ (Fig. 4A). We found a significantly reduced VZ radial thickness for

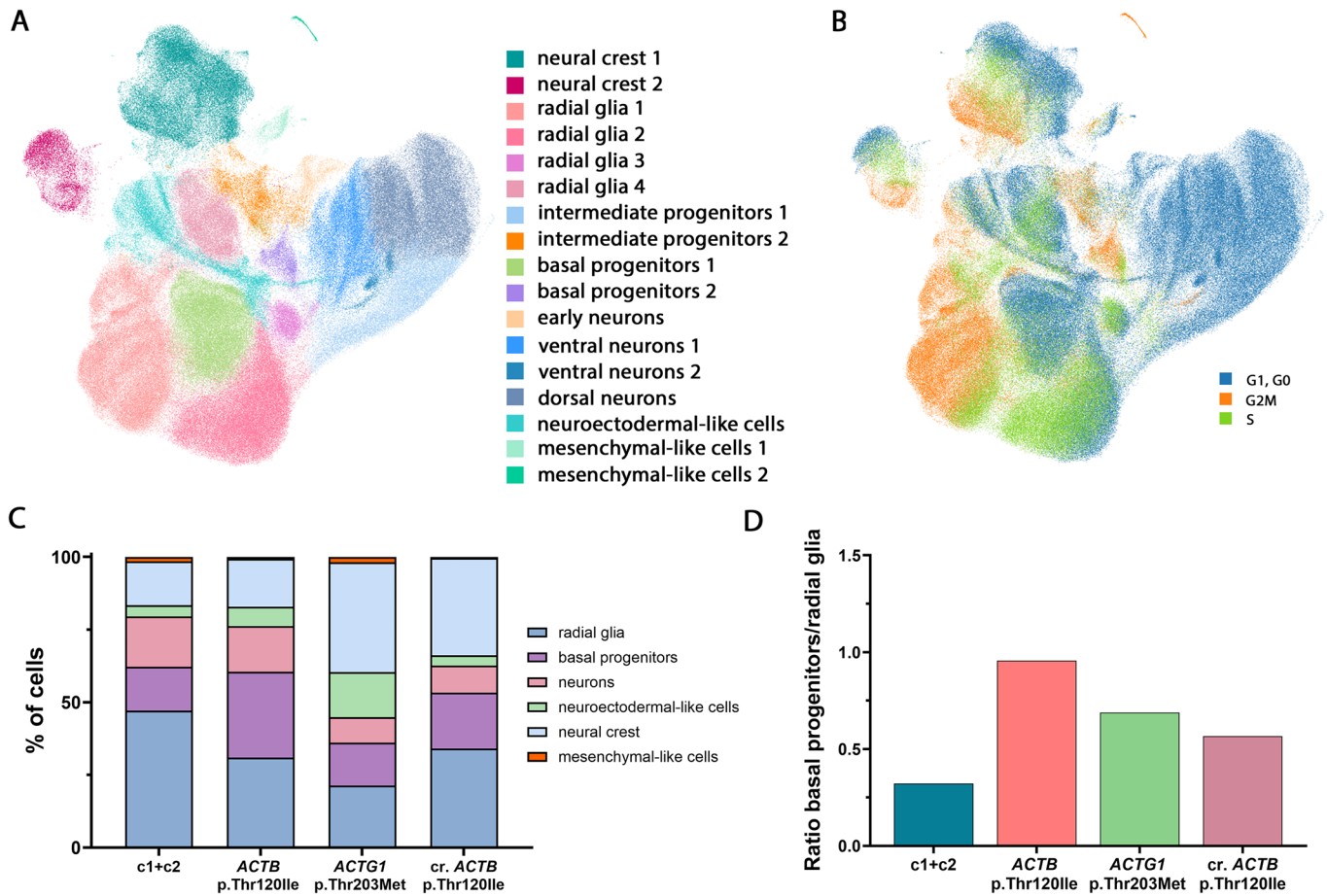

**Figure 3.   Single-cell RNA-seq analysis reveals an abnormal cell type composition of BWCFF-S cerebral organoids.**

(A, B) UMAP plot of single-cell RNA-seq data from control (generated from two different iPSC lines; SC102A-1 and CRTDi011-A)), BWCFF-S *ACTB* Thr120Ile (generated from two different iPSC clones; mut*ACTB*-1 and mut*ACTB*-2), BWCFF-S *ACTG1* Thr203Met (generated from two different iPSC clones; mut*ACTG1*-1 and mut*ACTG1*-2), and BWCFF-S–like *ACTB* Thr120Ile (generated from the CRISPR/Cas9-edited CRTDi011-A–derived iPSC clones CRTDi011-A–mut*ACTB*-1 and CRTDi011-A–mut*ACTB*-2, collectively referred to as cr. *ACTB* Thr120Ile). Each point represents a single cell, colored either by annotated cell type (A; specification on the right side of the panel) or by cell cycle phase (B; specification in the right lower corner of the panel). Data consists of 260,880 single cells of 80 control, 44 BWCFF-S *ACTB* Thr120Ile, 41 BWCFF-S *ACTG1* Thr203Met, and 20 cr. *ACTB* Thr120Ile 30 and 50 days-old cerebral organoids of 1–4 independent batches. (C) Quantification of the cell type composition in percent of scRNA-seq data from control (c1, SC102A-1 and c2, CRTDi011-A;), BWCFF-S *ACTB* Thr120Ile, BWCFF-S *ACTG1* Thr203Met, and cr. *ACTB* Thr120Ile cerebral organoids at culture day 30. Specification of the color-coded cell types on the right side of the panel. (D) Quantification of the ratio between basal progenitors and radial glia from scRNA-seq data of control (c1, SC102A-1 and c2, CRTDi011-A; blue bar), BWCFF-S *ACTB* Thr120Ile (red bar), BWCFF-S *ACTG1* Thr203Met (green bar), and cr. *ACTB* Thr120Ile (violet bar) cerebral organoids at culture day 30. (C, D) Data consists of 144,458 single cells of 40 control, 20 BWCFF-S *ACTB* Thr120Ile, 23 BWCFF-S *ACTG1* Thr203Met, and 10 cr. *ACTB* Thr120Ile 30-day-old cerebral organoids of 1–4 independent batches. Source data are available online for this figure.

the two types of BWCFF-S organoids compared to control organoids (Fig. 4B), indicating a thinner VZ likely caused by a reduced cell number. We next quantified the perimeter of the VZ, which also was smaller in the BWCFF-S VZ compared to the VZ of control organoids (Fig. 4C). Moreover, to corroborate the reduction in VZ size in the two types of BWCFF-S organoids compared to control, we measured the total area occupied by the VZ and found a significant reduction of VZ area in the two types of BWCFF-S cerebral organoids compared to control (Fig. 4D). To have an additional measurement which is independent of the placement of the basal boundary of the VZ, we quantified the length of the apical surface and found a significant reduction in the two types of BWCFF-S cerebral organoids compared to control (Fig. 4E). Interestingly, while these general VZ parameters were changed in both BWCFF-S conditions (*ACTB*

and *ACTG1*), the number of ventricle-like structures was only reduced in the actin variant NM_001101.5(*ACTB*):c.359 C > T p.Thr120Ile (Fig. 4F).

Despite the reduction in the above-mentioned VZ parameters, the density of cell nuclei in the VZ was similar for the two types of BWCFF-S organoids as compared to control organoids (Fig. 4G,H). Considering these VZ parameters together, these data strongly suggested that the reduction in the former VZ parameters (Fig. 4A–E) of the BCWFF-S organoids was likely caused by a decrease in VZ progenitors. Indeed, quantification of the DAPI-stained nuclei revealed a substantial decrease of the number of VZ progenitors per ventricle-like structure in the BWCFF-S organoids as compared to control organoids (Fig. 4I). Such a decrease can have various causes, including diminished proliferation of VZ progenitors or their increased apoptotic cell death. However,

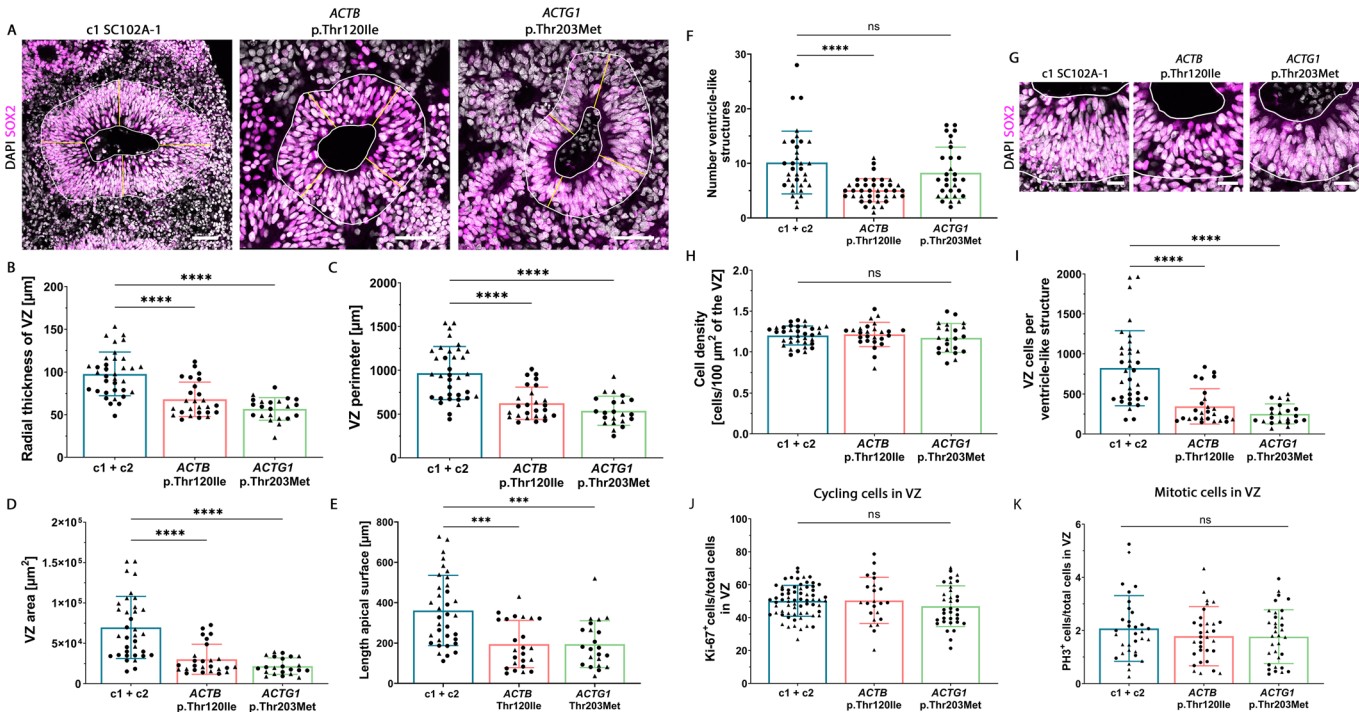

**Figure 4. BWCFF-S cerebral organoids show a reduction in VZ area and VZ progenitor number.**

(A–J) All cerebral organoids are at culture day 30. (A, G) SOX2 immunofluorescence (magenta), combined with DAPI staining (white), of sections containing ventricle-like structures of control (c1, SC102A-1, left image), BWCFF-S *ACTB* Thr120Ile (middle image), and BWCFF-S *ACTG1* Thr203Met (right image) cerebral organoids. High magnification images of the same samples are shown in (G). White lines indicate the area of the VZ; magenta lines indicate the radial thickness of the VZ at four different positions per ventricle-like structure. Scale bars, 50 μm (A) and 20 μm (G). (B–E, H, I) Quantification of various parameters of the VZ of control (c1, SC102A-1 and c2, CRTDi011-A; blue bars), BWCFF-S *ACTB* Thr120Ile (red bars), and BWCFF-S *ACTG1* Thr203Met (green bars) cerebral organoids. (B) Quantification of the radial thickness of the VZ (magenta lines in (A)). (C) Quantification of the VZ perimeter (outer white line in (A)). (D) Quantification of the VZ area (area between the inner and outer white lines in (A)). (E) Quantification of the length of the apical surface of a given ventricle-like structure (inner white line in (A)). (F) Quantification of the number of ventricle-like structures. Data are the mean of 34 control organoids (generated from two different iPSC lines; indicated by circles (c1, SC102A-1) and triangles (c2, CRTDi011-A), 345 ventricle-like structures in total), 40 BWCFF-S *ACTB* Thr120Ile organoids (generated from two different iPSC clones; indicated by circles and triangles, 201 ventricle-like structures in total) and 30 *ACTG1* Thr203Met organoids (generated from two different iPSC clones; indicated by circles and triangles, 248 ventricle-like structures in total), each from two to four independent batches; error bars indicate SD; statistical significance was determined by Kruskal–Wallis test (c1 + c2 vs. *ACTB* Thr120Ile: ****$P < 0.0001$; c1 + c2 vs. *ACTG1* Thr203Met: ns, not significant). (H) Quantification of the cell density in a 100 μm² field of the VZ. (I) Quantification of VZ cells per ventricle-like structure. (B–E, H, I) Data are the mean of 35 control organoids (generated from two different iPSC lines; indicated by circles (c1, SC102A-1) and triangles (c2, CRTDi011-A), 159 ventricle-like structures in total), 26 BWCFF-S *ACTB* Thr120Ile organoids (generated from two different iPSC clones; indicated by circles and triangles, 138 ventricle-like structures in total) and 22 *ACTG1* Thr203Met organoids (generated from two different iPSC clones; indicated by circles and triangles, 132 ventricle-like structures in total), each from two to three independent batches; error bars indicate SD; statistical significance was determined by Kruskal–Wallis test (B–E, I) and one-way ANOVA (H) (c1 + c2 vs. *ACTB* Thr120Ile: ****, $P < 0.0001$ (B–D, I), ***, $P = 0.0003$ (E); c1 + c2 vs. *ACTG1* Thr203Met: ****$P < 0.0001$ (B–D, I), ***$P = 0.0004$ (E); ns, not significant (H)). (J) Quantification of the proportion of DAPI⁺ cells that are also Ki-67⁺ in the VZ of control (c1, SC102A-1 and c2, CRTDi011-A; blue bar), BWCFF-S *ACTB* Thr120Ile (red bar), and BWCFF-S *ACTG1* Thr203Met (green bar) cerebral organoids at culture day 30. Data are the mean of 68 control (generated from two different iPSC lines; indicated by circles (c1, SC102A-1) and triangles (c2, CRTDi011-A), 23 BWCFF-S *ACTB* Thr120Ile (generated from two different iPSC clones; indicated by circles and triangles) and 34 BWCFF-S *ACTG1* Thr203Met (generated from two different iPSC clones; indicated by circles and triangles) VZs of 8–12 30-day-old cerebral organoids from 1–2 independent batches; error bars indicate SD; ns, not significant (one-way ANOVA). (K) Quantification of the proportion of DAPI⁺ cells that are also PH3⁺ in the VZ of control (c1, SC102A- 1 and c2, CRTDi011-A; blue bar), BWCFF-S *ACTB* Thr120Ile (red bar), and BWCFF-S *ACTG1* Thr203Met (green bar) cerebral organoids at culture day 30. Data are the mean of 59 ventricle-like structures of control organoids (generated from two different iPSC lines; indicated by circles (c1, SC102A-1) and triangles (c2, CRTDi011-A)), 48 ventricle-like structures of BWCFF-S *ACTB* Thr120Ile organoids (generated from two different iPSC clones; indicated by circles and triangles) and 38 ventricle-like structures of BWCFF-S *ACTG1* Thr203Met organoids (generated from two different iPSC clones; indicated by circles and triangles), of 16–17 30-day-old cerebral organoids from two independent batches each; error bars indicate SD; ns, not significant (Kruskal–Wallis test). Source data are available online for this figure.

caspase 3 staining showed that apoptotic cells were rare in both BWCFF-S and control organoids (Fig. EV2).

Quantification of Ki-67– and PH3–positive cells in the VZ revealed no significant differences in the proportion of these cells over total VZ cells between the BWCFF-S and control organoids (Figs. 4J,K and EV3). However, these cycling cell markers do not distinguish between the various modes of cell division in the VZ, that with regard to the two classes of progenitors found in the VZ

(see terminology in "Methods") apply only to APs; these modes of AP division may be either (i) symmetric–proliferative (1 AP – > 2 APs), resulting in an increase in AP and hence VZ progenitor number; (ii) asymmetric–self-renewing (1 AP – > 1 AP + 1 cell that leaves the VZ, typically a newborn BP), resulting in the maintenance of AP number but a transient increase in VZ progenitor number; or (iii) self-consuming (1 AP – > 2 cells that leave the VZ), resulting in a decrease of AP number but a transient

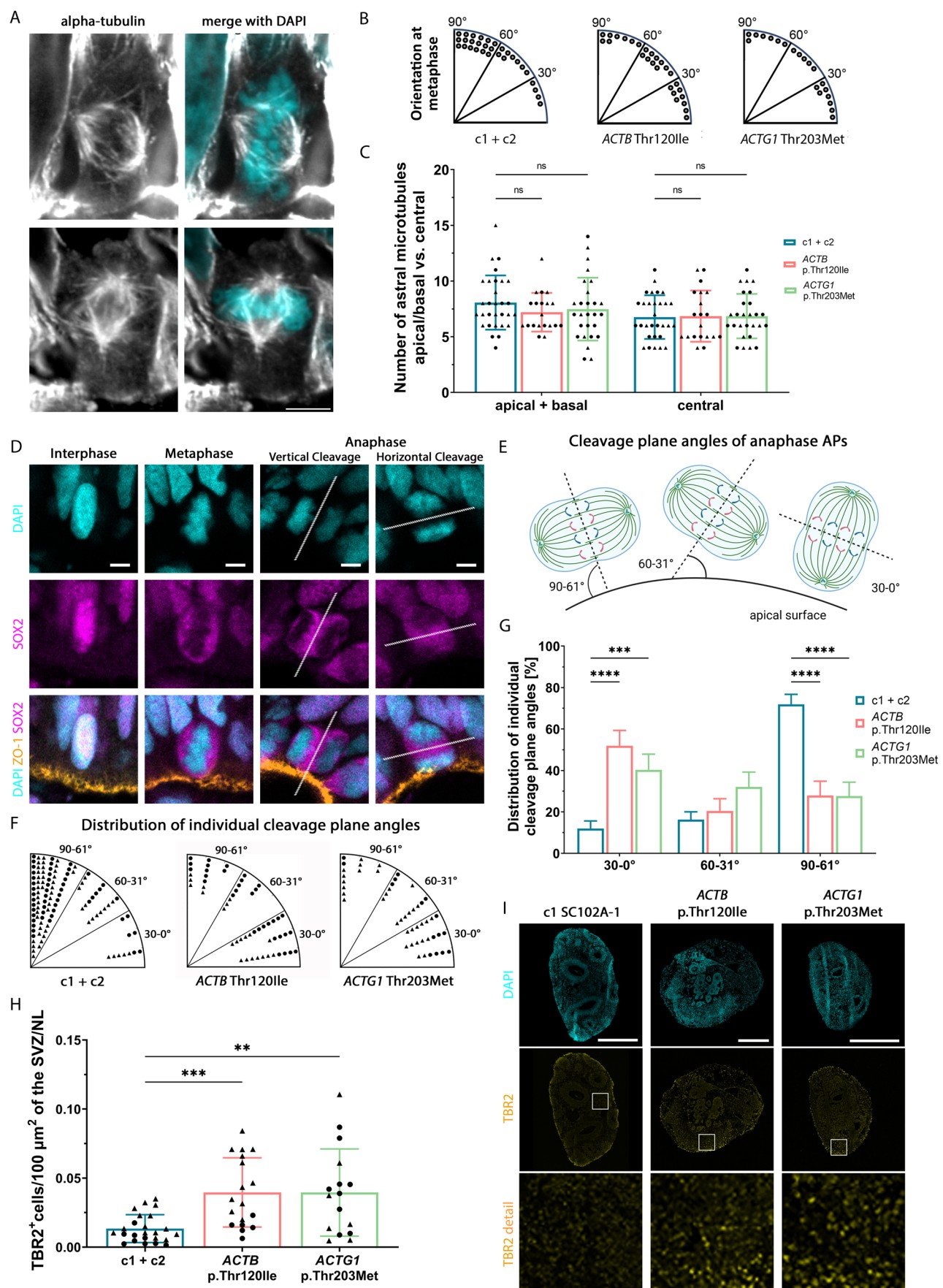

© The Author(s)

**Figure 5. Mitotic apical progenitors of BWCFF-S cerebral organoids predominantly show a horizontal rather than vertical cleavage plane.**

(A) Alpha-tubulin immunofluorescence (white) of 30-day-old human cerebral organoid sections showing mitotic microtubules, including astral microtubules, and counterstained with DAPI (blue). Upper images, vertical metaphase plate orientation, typical of mitotic APs in control organoids. Lower images, horizontal metaphase plate orientation, typical of mitotic APs in actin mutant organoids. Images are single 0.75 μm confocal sections. Apical surface is down. Scale bar, 5 μm. (B) Quantitation of the chromosome metaphase plate orientation in mitotic APs of control (left chart) and BCWFF-S (middle and right charts) cerebral organoids at culture day 30. Each dot represents a cell. 90° indicates a perfectly vertical metaphase plate orientation with respect to the local apical surface. Cells in the 90–61° bin have largely vertical orientations, cells in the 60–31° bin have largely oblique orientations, and cells in the 30–0° bin have largely horizontal orientations. (C) Quantification of apical basal (left) and central (right) astral microtubule fibers per mitotic AP of control (c1, SC102A-1 and c2, CRTDi011-A; blue bars), BWCFF-S *ACTB* Thr120Ile (red bars), and BWCFF-S *ACTG1* Thr203Met (green bars) cerebral organoids at culture day 30. Data are the mean of 30 control (generated from two different iPSC lines; indicated by circles (c1, SC102A-1) and triangles (c2, CRTDi011-A)), 20 BWCFF-S *ACTB* Thr120Ile (generated from two different iPSC clones; indicated by circles and triangles) and 27 BWCFF-S *ACTG1* Thr203Met (generated from two different iPSC clones; indicated by circles and triangles) mitotic APs; error bars indicate SD; ns, not significant (two-way ANOVA). (D) Double immunofluorescence for SOX2 (magenta) and ZO-1 (orange), combined with DAPI staining (blue), of sections of control (c2, CRTDi011-A) and BWCFF-S *ACTB* Thr120Ile 30-day-old cerebral organoids, as follows. Images show representative examples of interphase (left column, c2, CRTDi011-A) and metaphase (second column from left, c2, CRTDi011-A) APs, and APs in anaphase (two right columns) with vertical (c2, CRTDi011-A; second column from right) or horizontal (*ACTB* Thr120Ile; right column) cleavage plane angle; Scale bars, 5 μm. The axis of the cleavage plane is marked with a white dotted line. (E) Schematic depiction of the determination of anaphase AP cleavage plane angles relative to the apical surface (created with biorender.com). Cleavage plane angles are compiled into three groups as indicated. These three groups also apply to (F, G). (F, G) Quantification (F) and percentage distribution (G) of cleavage plane angles of anaphase APs of (i) control cerebral organoids (generated from two different iPSC lines; indicated by circles (c1, SC102A-1) and triangles (c2, CRTDi011-A) in (F) and by blue bars in (G)); (ii) BWCFF-S *ACTB* Thr120Ile cerebral organoids (generated from two different iPSC clones; indicated by circles and triangles in (F) and by red bars in (G)); and (iii) *ACTG1* Thr203Met cerebral organoids (generated from two different iPSC clones; indicated by circles and triangles in (F) and by green bars in (G)); all cerebral organoids are at culture day 30. Data consist of 49–114 anaphase APs, analyzed in (i) 23 control cerebral organoids (generated from 2 different iPSC lines; c1, SC102A-1: 10 cerebral organoids, 53 APs; c2, CRTDi011-A: 13 cerebral organoids, 61 APs); (ii) 21 *ACTB* Thr120Ile cerebral organoids (generated from two different iPSC clones; 33 and 21 APs per clone); and 23 *ACTG1* Thr203Met cerebral organoids (generated from two different iPSC clones; 15 and 34 APs, respectively); in each of the three conditions, the cerebral organoids are of 2–3 independent batches per clone. (G) For each of the three conditions, the numbers of mitotic AP cleavage plane angles in each group are expressed as a percentage of the total (set to 100). Error bars indicate SD; statistical significance was determined by two-way ANOVA (c1 + c2 vs. *ACTB* Thr120Ile: ****$P < 0.0001$; c1 + c2 vs. *ACTG1* Thr203Met: ***$P = 0.001$). Moreover, statistical analysis using the chi-squared test resulted in $P < 0.0001$. (H) Quantification of the number of TBR2+ cells in a 100 μm$^2$ field of the SVZ/NL of control (c1, SC102A-1 and c2, CRTDi011-A; blue bars), BWCFF-S *ACTB* Thr120Ile (red bars), and BWCFF-S *ACTG1* Thr203Met (green bars) cerebral organoid sections at culture day 30. Data are the mean of 24 control (generated from two different iPSC lines; indicated by circles and triangles), 18 BWCFF-S *ACTB* Thr120Ile (generated from two different iPSC clones; indicated by circles and triangles) and 16 BWCFF-S *ACTG1* Thr203Met (generated from two different iPSC clones; indicated by circles and triangles) 30-day-old cerebral organoids of 1–2 independent batches; error bars indicate SD; statistical significance was determined by one-way ANOVA (c1 + c2 vs. *ACTB* Thr120Ile: ***$P = 0.0004$; c1 + c2 vs. *ACTG1* Thr203Met: **$P = 0.0043$). (I) TBR2 immunofluorescence (yellow, top row), combined with DAPI staining (blue, middle row), of sections containing ventricle-like structures of control (c1, SC102A-1, left images), BWCFF-S *ACTB* Thr120Ile (middle images), and BWCFF-S *ACTG1* Thr203Met (right images) cerebral organoids. Bottom row, high magnification views of the insets shown in the middle row. Scale bars, 100 μm. Source data are available online for this figure.

increase in VZ progenitor number (Florio and Huttner, 2014; Heide and Huttner, 2023).

## Mitotic APs in BWCFF-S organoids show a massive reduction in the proportion of vertical cleavage planes

We therefore explored whether the mode of AP division was altered in the BWCFF-S organoids as compared to control organoids. Clues in this regard can be obtained from an analysis of the orientation of the metaphase plate and of the cleavage plane of mitotic APs in anaphase. The cleavage plane orientation of mitotic APs normally is predominantly vertical (i.e., parallel to the apical-basal axis of the cell); such a cleavage plane orientation is required for symmetric–proliferative divisions of APs but is also compatible with asymmetric–self-renewing divisions of these progenitors. In contrast, oblique cleavage plane orientation is typically associated with asymmetric–self-renewing divisions, and a horizontal cleavage plane orientation of mitotic APs (i.e., perpendicular to their apical-basal axis) is associated with either asymmetric–self-renewing divisions, or consumptive divisions yielding two non-AP daughters (Huttner and Kosodo, 2005; Mora-Bermudez and Huttner, 2015; Xing et al, 2021). Thus, while cleavage plane orientation does not strictly dictate daughter cell fate in the mammalian cerebral cortex, symmetric proliferative AP divisions require a vertical cleavage plane, as this is the only orientation that equally partitions the apicobasal cytoarchitecture, from the basal lamina contact to the apical endfoot, a hallmark of APs.

We first used high-resolution images of DAPI- and α-tubulin-stained AP metaphase nuclei to determine the mitotic AP metaphase plate orientation (Fig. 5A,B). We categorized AP metaphase plate orientation into three groups: (1) 90–61° as vertical orientation, (2) 60–31° as oblique orientation, and (3) 30–0° as horizontal orientation. We found a switch from predominantly vertical metaphase plate orientation to oblique and horizontal metaphase plate orientation in the mitotic APs of BWCFF-S cerebral organoids in comparison to those of control organoids (Fig. 5A,B). Orientation of the metaphase plate and later of the cleavage plane depends, among other things, on the abundance of astral microtubules (Mora-Bermudez and Huttner, 2015; di Pietro et al, 2016). Astral microtubules reaching the apical and basal cell cortex are more abundant in cells dividing in a vertical cleavage plane orientation than in cells dividing in an oblique or horizontal orientation. A reduced number of these microtubules would be indicative of increased oblique and horizontal cleavage plane orientation (Mora-Bermudez et al, 2014; Vargas-Hurtado et al, 2019; Da Silva et al, 2021). Accordingly, we next tested if the increase in oblique and horizontal metaphase plate orientation in mitotic APs of BWCFF-S cerebral organoids is caused by a reduced number of astral microtubules reaching the apical and basal cell cortex. For this purpose, we quantified the number of apical and basal (Fig. 5C, left) as well as central (Fig. 5C, right) astral microtubules in mitotic APs of control and BWCFF-S cerebral organoids. However, we did

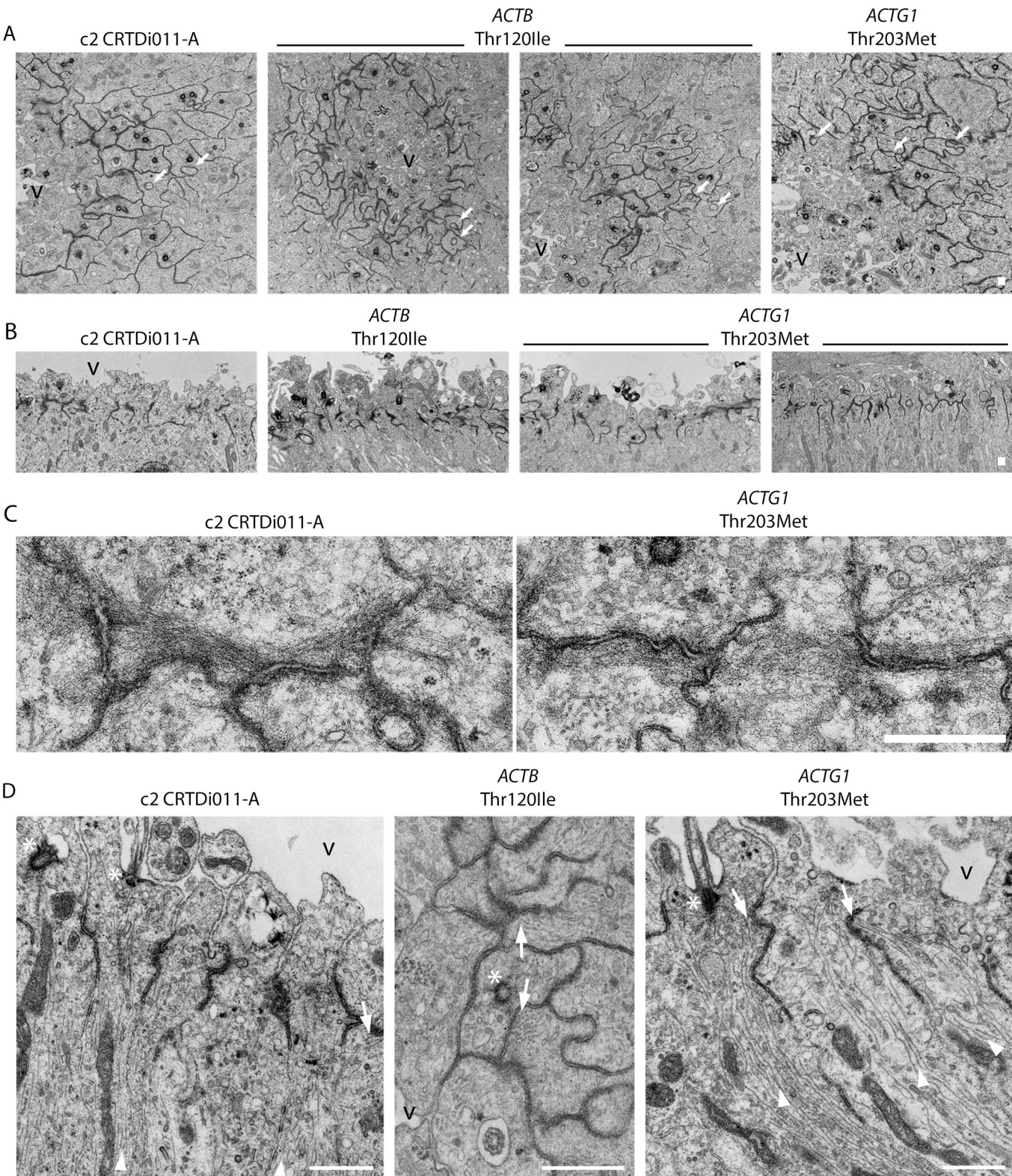

not observe any differences in the number of astral microtubules between the two conditions (Fig. 5C).

In light of this result and of the fact that mitotic spindles and metaphase plate can alter their orientation during metaphase (Haydar et al, 2003), we asked if the increase in oblique and horizontal metaphase plate orientation would culminate in a corresponding switch in cleavage plane orientation in anaphase. To analyze cleavage plane orientation, we used high-resolution images

**Figure 6. Cytoskeletal and morphological irregularities of VZ progenitors of BWCFF-S cerebral organoids.**

(A) Transmission electron micrographs of sections of control (c2, CRTDi011-A; left panel), BWCFF-S *ACTB* Thr120Ile (generated from two different iPSC clones, two middle panels), and *ACTG1* Thr203Met (right panel) 29- and 30-day-old cerebral organoids in cross-sectional views. Note the variable size and shape of the mutant cell circumference at the level of the apical adherens junction belt (dark line-like structures). Although protrusions from VZ progenitors into neighboring cells exist in control organoids, they are more frequently observed in the mutant situation (white arrows). Scale bar, 10 μm. V ventricle. (B) Transmission electron micrographs of sections of control (c2, CRTDi011-A; left panel), BWCFF-S *ACTB* Thr120Ile (second panel from left), and BWCFF-S *ACTG1* Thr203Met (generated from two different iPSC clones, two right panels) 29-day-old cerebral organoids; longitudinal cutting plane. Each cell is connected to its neighbors by an apical adherens junction belt (dark line-like structures) whose arrangement is preserved in the mutant cells. Scale bar, 10 μm. (C) High magnification transmission electron micrographs of apical adherens junction belts in control (c2, CRTDi011-A, left panel) and BWCFF-S *ACTG1* Thr203Met 29-day-old cerebral organoids. Note the presence of actin bundles adjacent to the junctions, both in control and *ACTG1* Thr203Met mutant APs. Scale bar, 1 μm. (D) Transmission electron micrographs showing either longitudinal views (left and right panels) or a cross-sectional view (middle panel) of ultrathin sections containing the apical cell cortex of VZ progenitors, likely APs (note the apical primary cilia), of control (c2, CRTDi011-A; left panel), *ACTB* Thr120Ile (middle panel) and *ACTG1* Thr203Met (right panel) 29-day-old cerebral organoids. Note the higher density of microtubule bundles (white arrowheads) in the mutant cells. Microtubules in the mutant cells are often anchored to the apical adherens junction belt (white arrows) rather than being nucleated at the basal body of the primary cilium (white asterisk). Scale bars, 1 μm. V ventricle. Source data are available online for this figure.

of the DAPI- and SOX2-stained AP anaphase nuclei at the ventricular surface (stained for ZO-1), and determined mitotic AP cleavage plane angles relative to the apical surface (Fig. 5D,E). Similar to the analysis of metaphase plate orientation, we categorized AP anaphase cleavage plane angles into three groups: (1) 90–61° as vertical cleavage, (2) 60–31° as oblique cleavage, and (3) 30–0° as horizontal cleavage. We found that in contrast to the predominantly vertical cleavage plane orientation of mitotic APs in control cerebral organoids, mitotic APs of BWCFF-S cerebral organoids exhibited a nearly randomized or majoritarian horizontal cleavage plane orientation (Fig. 5F,G), in line with the results of the analysis of the metaphase plate orientation. Symmetric proliferative AP divisions require a vertical cleavage plane, as this is the only way to equally divide, and distribute to the daughter APs, the full apicobasal cytoarchitecture (see above). Therefore, the change of BWCFF-S AP cleavage planes to a nearly randomized or majoritarian horizontal orientation is expected to prevent most APs from undergoing symmetric proliferative divisions to expand their pool size. Moreover, although both vertical and horizontal cleavage planes can be associated with asymmetric–self-renewing divisions, a shift away from vertical cleavages increases the probability that one or both daughter cells will delaminate from the VZ. In this way, the observed shift in cleavage plane orientation in BWCFF-S APs is likely to contribute to a reduction in the size of the VZ progenitor pool.

Next, we asked whether the shift in cleavage plane orientation in mitotic anaphase APs of BWCFF-S cerebral organoids results in an increased number of delaminated AP progeny cells, which in humans are predominantly BPs. To this end, we analyzed changes in BP numbers using immunofluorescence for the BP marker TBR2 (Appendix Fig. S6) and quantified the number of TBR2-positive BPs in the SVZ/NL. We found a significantly higher number of TBR2-positive BPs in the SVZ/NL of BWCFF-S cerebral organoids compared to control organoids (Fig. 5H). This finding is in line with our scRNA-seq data, which pointed to an elevated BP output (Fig. 3D). Taken together, these data indicate that the VZ progenitor pool in BWCFF-S cerebral organoids is diminished, likely due to the shift in cleavage plane orientation of mitotic anaphase APs, which increases the probability of progenitor delamination and thereby promotes BP generation at the expense of the primary progenitor type, the AP.

## VZ progenitors of BWCFF-S organoids display various cytoskeletal and morphological irregularities

As actin is a major component of the cytoskeleton affecting multiple structural and morphological features of the cell, we next sought to analyze VZ progenitors of BWCFF-S organoids in detail for cytoskeletal and morphological changes using transmission electron microscopy. First, we examined the size and shape of VZ progenitors. We found that at the level of the apical adherens junction belt, the size of BWCFF-S VZ progenitors and the shape of their cell circumference appeared to be more variable in comparison to control (Fig. 6A). In addition, protrusions from VZ progenitors into neighboring cells were more frequently observed in BWCFF-S than control organoids (Fig. 6A; Movies EV1–EV6).

As these effects were particularly noticeable at the level of the apical adherens junction belt, we next examined this belt in greater detail. We found that a key feature of the apical adherens junction belt, that is, to connect a cell to its neighbors, appeared to be maintained in VZ progenitors of BWCFF-S cerebral organoids (Fig. 6B,C). In line with this observation, upon immunofluorescence for ZO-1—a tight junction protein that is also associated with adherens junctions—the apical adherens junction belt appeared as a thin line in the ventricle-like structures of control as well as BWCFF-S cerebral organoids. (Fig. EV4A). Any abnormal adherens junction belt morphology was observed to a similar degree in both control and BWCFF-S conditions (Fig. EV4B). However, in the BWCFF-S organoids, we observed a higher density of microtubule bundles at the apical cell cortex that appeared to be anchored to the apical adherens junction belt rather than being nucleated at the basal body of the primary cilium (Fig. 6D).

To further assess this observation, we performed tubulin immunohistochemistry. This indicated an increased concentration of both alpha- and beta-tubulin at the apical cell cortex of VZ progenitors in the BWCFF-S organoids (Fig. EV5). Taken together, using transmission electron microscopy, we identified several reproducible cytoskeletal and morphological irregularities in VZ progenitors of BWCFF-S cerebral organoids, notably in the apical region of these cells. While these findings do not establish a causal relationship, they provide reproducible observations (further images are available as source data in BioStudies) of altered cell morphology that may be relevant in the context of abnormal

cleavage plane orientation of BWCFF-S mitotic APs (see "Discussion" for details).

## BWCFF-S-like *ACTB* Thr120Ile mutant cerebral organoids generated from CRISPR/Cas9-edited control CRTDi011-A iPSCs show essentially the same phenotypes as BWCFF-S *ACTB* Thr120Ile patient-derived cerebral organoids

In light of the (unavoidable) facts (see "Methods") that (i) the reprogramming of the control or BWCFF-S patient-derived fibroblasts had to be carried out at three different institutes, (ii) the culture conditions after reprogramming until the isolation of the iPSC clones were similar but not identical, and (iii) the genetic backgrounds of the healthy donor and the two BWCFF-S patients were (by definition) different, it was essential to demonstrate that the phenotypes observed in the BWCFF-S cerebral organoids as compared to control organoids were not due to these differences but indeed caused by the point mutations in *ACTB* or *ACTG1*. To this end, we subjected the control CRTDi011-A iPSCs to CRISPR/Cas9-mediated genome editing to introduce a single point mutation into *ACTB* that would selectively result in the Thr120Ile change. We then isolated two clones after this genome editing, referred to as CRTDi011-A–mut*ACTB*-1 and CRTDi011-A–mut*ACTB*-2, and used these to generate cerebral organoids, collectively referred to as cr. *ACTB* Thr120Ile organoids (see Fig. 7A for an overview of 30-day-old control and cr. *ACTB* Thr120Ile organoids). The two clones exhibited normal karyotype and expressed typical pluripotency markers (Appendix Fig. S12). Analyses in comparison to 30 days-old control CRTDi011-A iPSC–derived organoids of (i) the organoid area (Fig. 7B), (ii) the VZ radial thickness, perimeter, and area (Fig. 7C–E), (iii) the length of the apical surface (Fig. 6F) (iv) the number of VZ cells per ventricle-like structure (Fig. 6G), and (v) anaphase AP cleavage plane angles (Fig. 7H,I), revealed essentially the same phenotypes—with exception of the number of ventricle-like structures (Fig. 7J)—in the 30 days-old cr. *ACTB* Thr120Ile organoids as observed in the BWCFF-S *ACTB* Thr120Ile patient-derived organoids (see Figs. 2, 4, and 5). Moreover, scRNA-seq analysis of 30 and 50-day-old cr. *ACTB* Thr120Ile organoids revealed a cell type composition similar to that of BWCFF-S cerebral organoids (Figs. 3 and EV1; Appendix Fig. S11). These data indicate that the phenotypes observed in the BWCFF-S *ACTB* Thr120Ile patient-derived cerebral organoids were indeed caused by the point mutation in *ACTB*.

### Conclusion

In conclusion, our results indicate that cerebral organoids expressing BWCFF-S-associated mutant *ACTB* or *ACTG1* genes provide insight into the cellular mechanism underlying the cortical malformations seen in BWCFF-S patients. Thus, BWCFF-S organoids exhibit several cytoskeletal and morphological irregularities in VZ progenitors, notably in the apical region of these cells. Likely reflecting consequences of these irregularities, mitotic BWCFF-S APs show a striking deviation of the orientation of the mitotic spindle, which normally is parallel to the ventricular surface. This in turn is incompatible with AP proliferation increasing AP abundance, and likely results in an increased delamination of cells from the VZ due to the change in the orientation of the cleavage plane of mitotic APs from

predominantly vertical to a nearly randomized or majoritarian horizontal orientation. This lack of AP proliferation and increase in cell delamination from the VZ, on the one hand, is associated with a decrease in the size of the VZ within the ventricle-like structures of the BWCFF-S cerebral organoids and, on the other hand, with an increased abundance of BPs in the SVZ/NL of BWCFF-S organoids. Since APs in the VZ constitute the primary progenitor type in the developing brain, a reduction of their pool size due to delamination at a stage when these cells normally still self-amplify or at least self-renew will, in the short run, be consistent with an increased BP abundance. In the long run, such a reduction in the AP pool could limit the source from which BPs are generated, potentially decreasing neuronal output—one mechanism that would be consistent with microcephaly.

## Discussion

The present study provides crucial insight into the cell biological basis underlying the cortical malformations that are observed in patients afflicted with Baraitser–Winter-CerebroFrontoFacial syndrome (BWCFF-S) and are associated with the pathogenic variants of the *ACTB* and *ACTG1* genes encoding the actin isoforms βCYA and γCYA, respectively. Patients with BWCFF-S exhibit a variety of cortical anomalies; the most common are microcephaly and malformations of the lissencephaly spectrum (Di Donato et al, 2017). In short, the use of cerebral organoids grown from BWCFF-S patient-derived iPSCs carrying either *ACTB* or *ACTG1* mutations has revealed that one of the likely early steps in the formation of the BWCFF-S-associated microcephaly is cytoskeletal disorganization, changing AP mitotic spindle orientation. This prevents the proliferation of APs and likely increases cell delamination from the VZ, resulting in a reduction in the size of the VZ progenitor pool of these organoids. Three aspects of our study deserve particular attention: (i) the appropriateness of cerebral organoids grown from BWCFF-S patient-derived iPSCs to model the pathogenesis of the cortical malformation observed in BWCFF-S patients; (ii) a likely mechanism explaining how the mutant actin isoforms βCYA and γCYA cause the formation of BWCFF-S-associated microcephaly; (iii) the nature of the *ACTB* and *ACTG1* mutations and the functional relationship between the actin isoforms.

First, multiple studies demonstrated that cerebral organoids exhibit many aspects of in vivo corticogenesis (Kadoshima et al, 2013; Lancaster et al, 2013; Quadrato et al, 2017; Qian et al, 2016) (for review, see (Heide et al, 2018)). Although cerebral organoids have been successfully used to model various types of malformations of cortical development, the modeling of an actin-related pathology required additional caution as actin isoforms display a tightly regulated time- and tissue-specific expression pattern (Ampe and Van Troys, 2017). In the fetal human neocortex, both actin isoforms, βCYA and γCYA, show a widespread distribution across the entire cortical wall. Similar to this, cerebral organoids show a widespread distribution across the entire wall of the ventricle-like structures, in principle allowing the study of neurodevelopment defects of actinopathies, such as BWCFF-S, in this 3D model of cortical development. Indeed, the BWCFF-S-associated microcephaly is recapitulated in BWCFF-S patient-derived cerebral organoids by a significant reduction in organoid

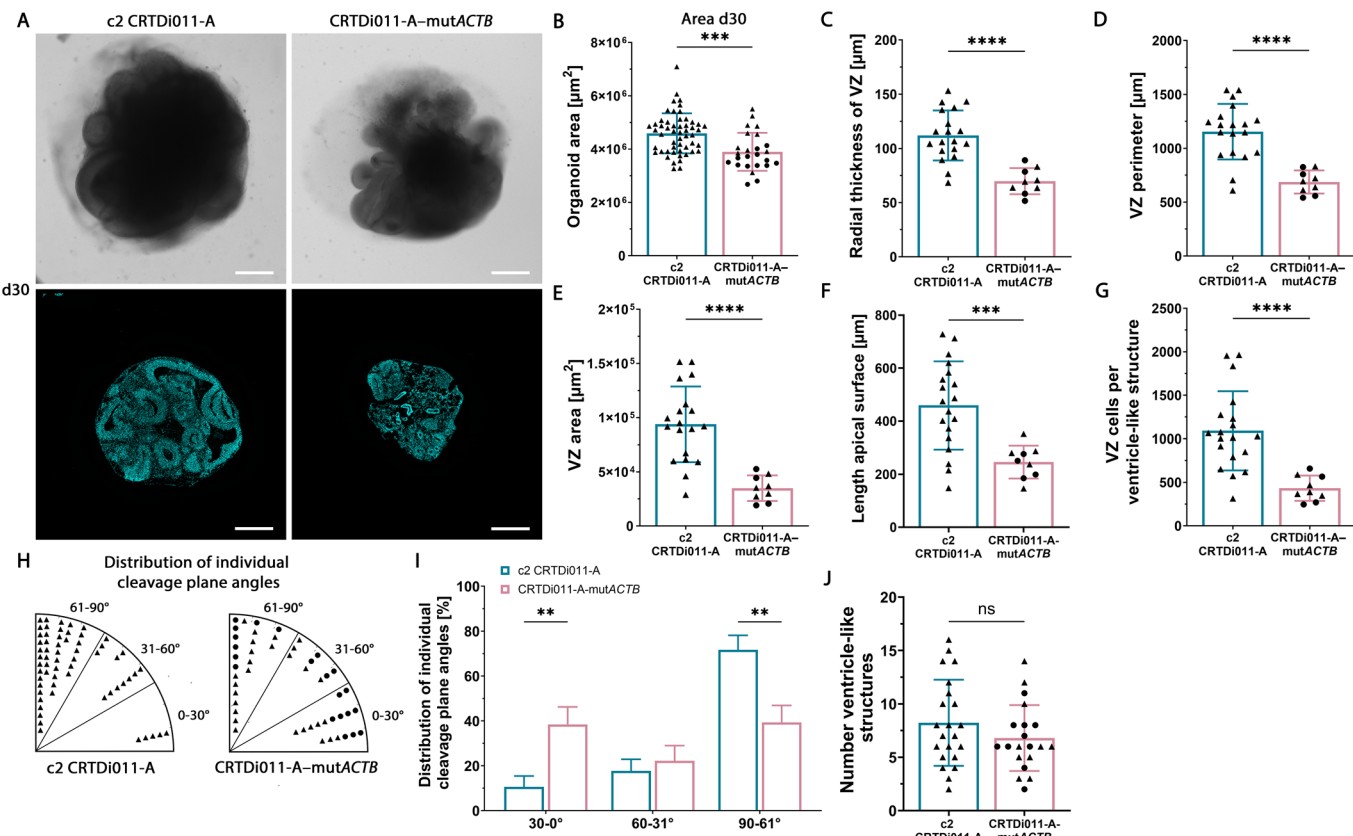

**Figure 7. BWCFF-S-like *ACTB* Thr120Ile mutant cerebral organoids generated from CRISPR/Cas9-edited control CRTDi011-A iPSCs show essentially the same phenotypes as BWCFF-S *ACTB* Thr120Ile patient-derived cerebral organoids.**

(A) Bright-field images (top rows) and DAPI-stained sections (bottom rows) of 30-day-old control cerebral organoids generated from the control iPSC line CRTDi011-A (left column) or the BWCFF-S–like *ACTB* Thr120Ile mutant cerebral organoids generated from a CRISPR/Cas9-edited CRTDi011-A–derived iPSC clone (CRTDi011-A-mut*ACTB*, right column). Scale bars, 500 μm. (B) Quantification of the area of 30-day-old cerebral organoids generated either from the control clone CRTDi011-A (blue bar) or from the CRISPR/Cas9-edited CRTDi011-A–derived iPSC clones CRTDi011-A-mut*ACTB*-1 (circles) and CRTDi011-A-mut*ACTB*-2 (triangles), collectively referred to as cr. *ACTB* Thr120Ile organoids (red bar). Data are the mean of 53 control and 25 cr. *ACTB* Thr120Ile organoids of 1–8 independent batches; error bars indicate SD; statistical significance was determined by unpaired Student's *t* test (c2 CRTDi011-A vs. CRTDi011-A–mut*ACTB*: ***P = 0.0003). (C-G) Quantification of various parameters of the VZ of 30-day-old control (CRTDi011-A; blue bars) and cr. *ACTB* Thr120Ile (red bars) cerebral organoids. (C) Quantification of the radial thickness of the VZ. (D) Quantification of the VZ perimeter. (E) Quantification of the VZ area. (F) Quantification of the length of the apical surface of a given ventricle-like structure. (G) Quantification of VZ cells per ventricle-like structure. (C-G) Data are the mean of 19 control organoids (77 ventricle-like structures in total) and 9 cr. *ACTB* Thr120Ile organoids (symbols as in (B), 114 ventricle-like structures in total), each from one to three independent batches; error bars indicate SD; statistical significance was determined by unpaired Student's *t* test (C-E) and Mann–Whitney *U* test (F,G) (c2 CRTDi011-A vs. CRTDi011-A–mut*ACTB*: ****P < 0.0001 (C-E, G), P = 0.0008 (F)). Note that for the control organoids, the same data points are depicted as those indicated by triangles in Fig. 3B–E,I (c2). (H, I) Quantification (H) and percentage distribution (I) of cleavage plane angles of anaphase APs of (i) control cerebral organoids (CRTDi011-A, indicated by triangles in (H) and by blue bars in (I)); and (ii) cr. *ACTB* Thr120Ile cerebral organoids (symbols in (H) are as in (B), red bars in (I)); all cerebral organoids are at culture day 30. Data consists of 61 anaphase APs of 13 control cerebral organoids and 49 (circles, 21; triangles, 28) anaphase APs of 15 cr. *ACTB* Thr120Ile cerebral organoids (generated from 2 different iPSC lines; CRTDi011-A-mut*ACTB*-1: 7 cerebral organoids; c2, CRTDi011-A-mut*ACTB*-2: 8 cerebral organoids); the cerebral organoids are of 1–2 independent batches per clone. (I) For each of the two conditions, the numbers of mitotic AP cleavage plane angles in each group are expressed as a percentage of the total (set to 100). Error bars indicate SD; statistical significance was determined by two-way ANOVA (c2 CRTDi011-A vs. CRTDi011-A-mut*ACTB*: 0–30°, **P = 0.0075, 61–90°, **P = 0,0014). Note that for the control organoids, the same data points are depicted as those indicated by triangles in panel C and by blue bars in Fig. 4D (c2). (J) Quantification of the number of ventricle-like structures. Data are the mean of 22 control organoids (181 ventricle-like structures in total) and 20 cr. *ACTB* Thr120Ile organoids (symbols as in (B), 130 ventricle-like structures in total), each from one to three independent batches; error bars indicate SD; ns, not significant (Mann–Whitney *U* test). Source data are available online for this figure.

size. In summary, cerebral organoids grown from BWCFF-S patient-derived iPSCs are suitable to model the microcephaly observed in BWCFF-S patients.

Second, with regard to a likely mechanism underlying BWCFF-S-associated cortical malformations, only very few neuropathological reports are currently available (for review, see (Brock et al, 2021)). In one of these reports, the fetal brain from a pregnancy terminated after the diagnosis of BWCFF-S was analyzed (Vontell

et al, 2019). This brain showed a reduced number and poor organization of neurons in the cortical plate (Vontell et al, 2019). Interestingly, the published fetus carried exactly the same missense variant in *ACTG1*:p.Thr203Met as the patient who donated the skin biopsy for iPSC reprogramming and cerebral organoid generation in this study. In line with the clinical and neuropathological data, cerebral organoids with the *ACTG1*:p.Thr203Met variant as well as organoids with the *ACTB* mutation had significantly reduced size,

recapitulating microcephaly in BWCFF-S patients. As a potential mechanism of this size reduction in BWCFF-S patient-derived organoids, we identified a switch of the mitotic AP cleavage plane from a predominantly vertical orientation to a nearly randomized or majoritarian horizontal orientation, preventing the proliferation of APs that would increase their abundance and thereby the VZ progenitor pool, and likely leading to increased cell delamination from the VZ of BWCFF-S organoids.

In this context, it is important to emphasize that the phenotypes observed in the BWCFF-S cerebral organoids when compared to control cerebral organoids were not due to differences other than the mutations in the actin genes, such as differences in the generation of the control and patient-derived iPSC clones used to generate the cerebral organoids, or differences in the genetic backgrounds between the healthy donor and the two BWCFF-S patients. Specifically, organoids generated from genome-edited clones that were derived from the control iPSC line CRTDi011-A, in which a single point mutation into *ACTB* had been introduced that would selectively result in the Thr120Ile change, showed essentially the same phenotypes compared to control as those observed in the BWCFF-S *ACTB* Thr120Ile patient-derived organoids, demonstrating that the latter phenotypes were indeed caused by the point mutation in *ACTB*.

In light of the various cytoskeletal and morphological irregularities that we observed in the BWCFF-S VZ progenitors, several explanations for the change in cleavage plane orientation of mitotic APs can be envisioned. One possible factor is that the increased concentration of tubulin and microtubules at the apical cell cortex of the BWCFF-S VZ progenitors could result, upon AP mitotic entry, in an altered positioning of the mitotic spindle resulting in the abnormal, nearly randomized or majoritarian horizontal cleavage plane orientation of BWCFF-S VZ progenitors. This could be either a direct effect of the increased microtubule concentration (see also below) or a combined effect together with the actin cytoskeleton. Several seminal studies have established that during cell division, actin and microtubules work closely together and should be considered as a unified system in which subcomponents co-regulate each other (Dogterom and Koenderink, 2019). Our observation of horizontal cleavage planes prevailing in mitotic APs of BWCFF-S cerebral organoids not only provides a potential mechanism explaining the reduced brain size in BWCFF-S patients, but could also indicate an extensive actin–microtubule crosstalk. Microtubules emanating from the spindle poles physically interact with actin filaments, and this interaction is required for proper spindle positioning and orientation prior to anaphase (Pimm and Henty-Ridilla, 2021). Such interactions are mediated, for example, by formins (Bartolini and Gundersen, 2010). It was previously shown that mutations in actin (Palmer et al, 1992), mutations in formin (Lee et al, 1999), or low doses of the F-actin poison latrunculin A (Theesfeld et al, 1999), cause spindle orientation defects. In such a situation, the actin mutations of BWCFF-S could either directly affect the actin filaments or the binding of formins, and thereby the interaction with the mitotic spindle.

Another contributing factor may be the increased variability in cell shape of BWCFF-S VZ progenitors, which could affect the orientation of the mitotic spindle. One classical example of such a scenario is the atypical myosin Dachs, which controls spindle orientation in the developing *Drosophila* wing. During wing development, Dachs drives cell shape changes, which in turn direct

the orientation of the spindle (Mao et al, 2011). In the case of BWCFF-S, the mutations in *ACTB* and *ACTG1* could change the arrangement of the actin cell cortex, which results in a change of the cell shape and an altered spindle orientation. Another possibility is that the altered cell shape leads to changes in mechanical forces acting on the mitotic spindle, and thereby changes its orientation (Nestor-Bergmann et al, 2014).

In addition, protrusions from VZ progenitors into neighboring cells, more frequently observed in BWCFF-S than in control organoids, could influence the orientation of the mitotic spindle. Obviously, these protrusions likely affect the cell shape and also the mechanical forces of the protrusion-sending as well as the protrusion-receiving cell, which as described above could contribute to the altered mitotic spindle orientation of BWCFF-S mitotic APs. In addition, these protrusions could affect the cell adhesion between neighboring cells, also resulting in a change of mitotic spindle orientation.

Moreover, although the apical adherens junction belt seems to be largely intact in the VZ of BWCFF-S cerebral organoids, the observed increased concentration of tubulin and microtubules at the adherens junction-near, apical cell cortex of the BWCFF-S VZ progenitors could affect the orientation of the mitotic spindle. This increased concentration could likely result, upon AP mitotic entry, in an altered occupation of astral microtubule binding sites (despite the similar number of astral microtubules observed), such that a positioning of the mitotic spindle parallel to the ventricular surface is hindered.

Finally, it is conceivable that a combination of some, or all, of the above-described irregularities could influence the orientation of the mitotic spindle. Mitotic APs normally violate Hertwig's rule that mitotic spindles are preferentially oriented along the longest axis of a dividing cell, which results in a horizontal cleavage plane orientation (default situation) (Hertwig, 1884; Mora-Bermudez and Huttner, 2015). In other words, the vertical cleavage plane orientation of mitotic APs reflects the non-default state. To achieve this non-default state, VZ progenitors need to "force" the cellular and sub-cellular mechanics to reach a condition that allows a vertical cleavage plane orientation. As soon as this condition gets disturbed to a degree that it cannot be maintained anymore, the cleavage plane orientation may revert back to the default state (horizontal cleavage plane orientation) following Hertwig's rule. In the case of BWCFF-S, the various cytoskeletal and morphological irregularities described here could sum up such that the cytomechanic condition of the cell for a horizontal spindle cannot be maintained anymore, and BWCFF mitotic APs fall back into the default state of horizontal cleavage plane orientation. Future studies need to address the combined and individual contributions of the various cytoskeletal and morphological irregularities of BWCFF-S VZ progenitors to uncover the precise mechanism underlying BWCFF-S abnormal cleavage plane orientation.

Third, a critical consideration pertains to the molecular nature of the *ACTB* and *ACTG1* mutations and the extent to which the beta- and gamma-actin isoforms exhibit functional redundancy or divergence within neural progenitor cells. As both *ACTB* and *ACTG1* mutations are missense variants, they likely exert dominant effects rather than simple loss of function. Indeed, loss of ACTB function produces distinct phenotypes unrelated to BWCFF-S (Cuvertino et al, 2017; Vedula et al, 2017). The long-standing debate on whether cytoskeletal beta and gamma actin isoforms have

overlapping or distinct cellular functions remains unresolved (Dugina et al, 2009; Tondeleir et al, 2012; Lechuga et al, 2020). Despite their high sequence similarity and co-expression, their functions are likely not entirely redundant. Our study reinforces this view by demonstrating that pathogenic variants in either isoform similarly disrupt cortical development.

Taken together, our results demonstrate the suitability of cerebral organoids as models for actinopathies and for identifying the underlying mechanisms for the development of cortical malformations of one of these actinopathies, BWCFF-S. Our observed change in cleavage plane orientation as the underlying cause of the reduced cerebral organoid size and likely the BWCFF-S microcephaly adds additional support to the observation that mutations in different cytoskeletal components all result in a similar spectrum of malformations of cortical development (for review, see (Koenig et al, 2021)). For example, two organoid-based studies of the Miller–Dieker syndrome (heterozygous deletion of chromosome 17p13.3 involving the genes *LIS1* and *YWHAE*) reported a switch from vertical to horizontal cleavage plane orientation of APs (Iefremova et al, 2017; Bershteyn et al, 2017), essentially the same pathology that we observed in BWCFF-S associated with actin mutations. Hence, our work provides further evidence that a substantial number of malformations of cortical development represent cytoskeletal disorders with mutations affecting different cytoskeletal components, disrupting the same cellular functions.

# Methods

### Reagents and tools table

| Reagent/ resource | Reference or source | Identifier or catalog number | RRID | Dilution factor |
|---|---|---|---|---|
| **Experimental models** | | | | |
| SC102A-1 | System Biosciences | | CVCL_IT66 | |
| mut*ACTB*-1 | This study | | | |
| mut*ACTB*-2 | This study | | | |
| mut*ACTG1*-1 | This study | | | |
| mut*ACTG1*-2 | This study | | | |
| CRTDi011-A | Center for Regenerative Therapies Dresden (CRTD) | | CVCL_C1P8 | |
| CRTDi011-A–mut*ACTB*-1 | This study | | | |
| CRTDi011-A–mut*ACTB*-2 | This study | | | |
| **Antibodies** | | | | |
| βCYA | Bio-Rad | CMCA5775GA | AB_2571580 | 1:50 |
| γCYA | Bio-Rad | MCA5776GA | AB_2571583 | 1:100 |
| alpha-tubulin | Sigma-Aldrich | T6199 | AB_477583 | 1:1000 |
| beta-tubulin | Abcam | ab179513 | AB_3073861 | 1:500 |
| Ki-67 | Abcam | ab15580 | AB_443209 | 1:300 |

| Reagent/ resource | Reference or source | Identifier or catalog number | RRID | Dilution factor |
|---|---|---|---|---|
| Nestin | Sigma-Aldrich | N5413 | AB 1841032 | 1:100 |
| Pan-actin | Novus Biologicals | NB600-535 | AB_2222881 | 1:200 |
| Pan-cadherin | Sigma-Aldrich | C1821 | AB_476826 | 1:300 |
| PH3 | Abcam | ab10543 | AB_2295065 | 1:300 |
| SOX2 | R + D Systems | AF2018 | AB_355110 | 1:300 |
| TBR2 | Abcam | ab23345 | AB_778267 | 1:300 |
| Caspase3 | Abcam | ab208161 | | 1:200 |
| CREST | Antibodies Incorporated | 15-235 | AB_2939059 | 1:300 |
| Ctip2 | Abcam | ab18465 | AB_2064130 | 1:300 |
| Tuj1 | BioLegend | 801201 | AB_2313773 | 1:200 |
| ZO-1 | Invitrogen | 61-7300 | AB_138452 | 1:300 |
| DAPI | Roche | 10236276001 | | 1:1000 |
| Goat anti-mouse | Jackson Immunoresearch | 115-545-205 | AB_2338854 | 1:200 |
| Goat anti-mouse | Jackson Immunoresearch | 115-175-207 | AB_2338717 | 1:50 |
| Donkey anti-mouse | Thermo Fisher | A-21202 | AB_141607 | 1:500 |
| Donkey anti-rabbit | Thermo Fisher | A-31572 | AB_162543 | 1:500 |
| Donkey anti-goat | Thermo Fisher | A-21447 | AB_2535864 | 1:500 |
| Donkey anti-rat | Thermo Fisher | A-21208 | AB_2535794 | 1:500 |
| **Oligonucleotides and other sequence-based reagents** | | | | |
| ssODN) sequence for c.359 C > T mutation | This study, IDT | "Methods" CRISPR/Cas9 | | |
| ssODN sequence for c.359 C > C reference maintenance | This study, IDT | "Methods" CRISPR/Cas9 | | |
| PCR primers | This study | "Methods" CRISPR/Cas9 | | |
| **Software** | | | | |
| Loupe Browser 8 | 10x Genomics | | | |
| GraphPad Prism 10.3.0 | Graphpad Software, Inc. | | | |

## Reprogramming of somatic cells (fibroblasts) to generate control and BWCFF-S patient-derived induced pluripotent stem cells (iPSCs)

After obtaining a positive statement (EK127032017 and EK44012019) from the ethics council of TU Dresden, fibroblasts were isolated, using a previously described procedure (Latham et al, 2018), from skin punch biopsies of a healthy adult female donor

(control) and of two female BWCFF-S patients carrying either the NM_001101.5(*ACTB*):c.359 C > T p.Thr120Ile or the NM_001614.5(*ACTG1*):c.608 C > T p.Thr203Met mutation. The presence of both BWCFF-S mutations was confirmed with locus-specific Sanger sequencing as previously described (Di Donato et al, 2014).

The three batches of isolated human fibroblasts were put into culture under identical conditions (in BIO-AMF™-2 medium (Biological Industries)), grown for 2–3 passages, and then subjected to reprogramming to generate iPSCs. Fibroblasts from the BWCFF-S mutant *ACTB* patient were frozen in liquid nitrogen before reprogramming. Due to differences in the availability of the healthy donor and the two BWCFF-S patients providing the skin punch biopsies, this reprogramming was performed correspondingly at three different locations. The human control fibroblasts were reprogrammed at the CRTD Stem Cell Engineering Facility at the *Technische Universität Dresden*. The fibroblasts from the BWCFF-S patient carrying the NM_001101.5(*ACTB*):c.359 C > T p.Thr120Ile mutation were reprogrammed at the *Medizinische Hochschule Hannover*. The fibroblasts from the BWCFF-S patient carrying the NM_001614.5(*ACTG1*):c.608 C > T p.Thr203Met mutation were reprogrammed at the *Medizinisch Theoretisches Zentrum Dresden*. All reprogrammings were performed using the same type of kit, the CytoTune-iPS 2.0 Sendai Reprogramming Kit (Thermo Fisher Scientific).

Human control fibroblasts were transduced according to the reprogramming kit manufacturer's instructions. After 5 days following transduction, cells were plated onto irradiated CF1 Mouse Embryonic Fibroblasts (Thermo Fisher Scientific) and cultured in ReproTeSR (StemCell Technologies), After 10 days of culture, individual iPSC colonies were mechanically picked and expanded as clonal lines on Matrigel-coated cell culture dishes (Corning) in mTeSR™1 (StemCell Technologies). IPSC lines were passaged using ReLeSR (StemCell Technologies).

BWCFF-S *ACTB* Thr120Ile patient-derived fibroblasts were thawed in fibroblast medium I (DMEM (Gibco), supplemented with 10% FCS (PAA), 1% MEM non-essential amino acids (Gibco), and 1 mM L-glutamine (Gibco), cultured for 3 days, and then transduced according to the reprogramming kit manufacturer's instructions. After 4 days following transduction, cells were plated on mitotically inactivated murine embryonic feeder cells and cultured in iPSC medium (knockout-DMEM (Gibco), supplemented with 20% knockout serum replacement (Gibco), 1 mM L-glutamine (Gibco), 0.1 mM 2-mercaptoethanol (Gibco), 1% MEM non-essential amino acids (Gibco), and 8 ng/ml bFGF-2 (Pepro-Tech)). After 8 days of culture, individual iPSC colonies were mechanically picked and expanded as clonal lines on murine embryonic feeder cells in iPSC medium. IPSC lines were passaged using collagenase IV (Gibco).

BWCFF-S *ACTG1* Thr203Met patient-derived fibroblasts were transduced according to the reprogramming kit manufacturer's instructions. Directly after transduction, cells were cultured in fibroblast medium II (DMEM/F-12 (Gibco), supplemented with 1% GlutaMAX Supplement (Gibco), 20 ng/ml hbFGF (PeproTech), 1% 2-mercaptoethanol (Serva), 10% FBS (Sigma-Aldrich), and 1% MEM non-essential amino acids (Gibco). After 7 days of culture, the cells were passaged on Geltrex-coated cell culture dishes (Thermo Fisher Scientific) in fibroblast medium II. One day later, the medium was changed to Essential 8™ Medium (Gibco).

Individual iPSC colonies were mechanically picked and expanded as clonal lines on Geltrex-coated cell culture dishes in Essential 8™ Medium, supplemented with 2 µM thiazovivin for 24 h (Sigma-Aldrich). IPSC lines were passaged using Versene solution (Thermo Fisher Scientific).

Although there were some differences in cell culture conditions after the reprogramming step, as can be tracked above, these are unlikely be responsible for the phenotypic differences observed with the various cerebral organoids described in the Results section. The same applies to the different genetic background of the healthy donor and the two BWCFF-S patients, in light of the following considerations. All iPSC clones obtained from the healthy donor- and the two BWCFF-S patient-derived fibroblasts were treated identically after the clonal expansion step (see below). All iPSC clones were tested for the removal of Sendai virus genomes and transgenes by reverse transcription-PCR analysis, and further checked for pluripotency marker expression, differentiation to three germ layers, and an intact karyotype by G-banding (Appendix Figs. S8, S9, and S13). One fully characterized clone of the control iPSC line (referred to as CRTDi011-A) and two clones per mutation (referred to as mut*ACTB*-1 and mut*ACTB*-2, and as mut*ACTG1*-1 and mut*ACTG1*-2) were used for cerebral organoid generation. Moreover, the presence of the NM_001101.5(*ACTB*):c.359 C > T p.Thr120Ile and NM_001614.5(*ACTG1*):c.608 C > T p.Thr203Met mutation was additionally confirmed by locus-specific Sanger sequencing of the two clones each used in this study.

Furthermore, strong evidence that the phenotypic differences between the mut*ACTB*-1-, mut*ACTB*-2-, mut*ACTG1*-1-, and mut*ACTG1*-2-derived cerebral organoids and the control organoids are caused by the mutations in *ACTB* and *ACTG1*, respectively, is provided by our finding that introducing the Thr120Ile change into *ACTB* in the control iPSC clone CRTDi011-A using the CRISPR/Cas9 technology is sufficient to cause essentially the same cerebral organoid phenotype as observed in the BWCFF-S *ACTB* Thr120Ile patient-derived cerebral organoids (Fig. 7).

## CRISPR/Cas9-mediated generation of *ACTB* Thr120Ile mutant iPSCs from control CRTDi011-A iPSCs

For the target region NC_000007.14:g.5529165 G > A (NM_001101.5(*ACTB*):c.359 C > T p.Thr120Ile), the sequence was analyzed for appropriate guide RNA binding sites using the Geneious software and the CRISPOR design webtool (http://crispor.tefor.net/). The selected target sequence plus PAM was: 5'-AGGTAGCGGGCCACTCACCTGGG-3'. Guide RNA was ordered as crRNA from IDT (Integrated DNA Technologies). Linear donor templates with 50-bp homology arms were purchased from IDT as PAGE Ultramer ssDNA oligonucleotides with three phosphorothioated DNA bases on each side. In order to generate a heterozygous iPSC line, two oligonucleotides were designed; one contained the point mutation c.359 C > T, whereas the other contained the sequence encoding the reference amino acid sequence, with both oligonucleotides containing silent mutations to prevent recutting of either targeted allele. The ss-oligodeoxynucleotide (ssODN) sequence for the c.359 C > T mutation was: 5'-AGGAGCACCCCGTGCTGCTGACCGAGGCCCCCC TGAACCCCAAGGCCAACAGGGAAAAAATGA**T**AC AGGTGA GTGGCCCGCTACC**T**CTTCTGGTGGCCGCCTCCCTCCTTCC TGG-3'. The ssODN sequence for the c.359 C > C reference maintenance was: 5'-AGGAGCACCCCGTGCTGCTGACCGAGG

CCCCCCTGAACCCCAAGGCCAACAGGGAAAAAATGA**C**ACA
GGTGAGTGGCCCGCTACCTCTTCTGGTGGCCGCCTCCCTC
CTTCCTGG-3′ (underlined bases indicate match to guide RNA).
CRTDi011-A cells were transfected with RNP (46 pmol guideRNA
and 31 pmol Cas9) and 0.5 µl of each ssODN (100 µM stock,
yielding 50 pmol final amount) using the NEON transfection
system (settings: 1000 V, 20 ms, 1 pulse). Cells were single-cell
sorted by FACS, and heterozygous single-cell clones were identified
by Sanger Sequencing of the target region (primer_fwd: 5′-
GTCACCAACTGGGACGACAT-3′, primer_rev: 5′-
GCTAAGTGTGCTGGGGTCTT-3′). Two such clones, referred to
as CRTDi011-A–mut*ACTB*-1 and CRTDi011-A–mut*ACTB*-2, were
used for the generation of cerebral organoids, after successful
validation of pluripotency marker expression (Appendix Fig. S12).

## iPSC culture

The clonal control iPSC lines SC102A-1 (System Biosciences) and
CRTDi011-A (this study), the mutated CRTDi011-A-derived clones
CRTDi011-A–mut*ACTB*-1 and CRTDi011-A–mut*ACTB*-2, and the
BWCFF-S patient-derived iPSC lines BWCFF-S *ACTB* Thr120Ile
(clones mut*ACTB*-1 and mut*ACTB*-2) and BWCFF-S *ACTG1*
Thr203Met (clones mut*ACTG1*-1 and mut*ACTG1*-2) (all this study)
were subjected to an identical culture protocol under feeder-free
conditions on Matrigel-coated plates (Corning) in mTeSR™1
(StemCell Technologies) at 37 °C, in a humidified atmosphere of
5% $CO_2$ and 95% air. Medium was changed daily, and cells were
passaged every third day, using ReLeSR™ (StemCell Technologies).

## Generation of cerebral organoids

Cerebral organoids were generated as previously described (Lan-
caster et al, 2013; Lancaster and Knoblich, 2014; Kanton et al, 2019;
Mora-Bermudez et al, 2016) with minor modifications. Patient-
derived organoids were generated from 2 clones each per mutation
(clones mut*ACTB*-1 and mut*ACTB*-2, and clones mut*ACTG1*-1 and
mut*ACTG1*-2), patient-like organoids were generated from the
mutated CRTDi011-A-derived clones CRTDi011-A–mut*ACTB*-1
and CRTDi011-A–mut*ACTB*-2, control organoids were generated
from two different iPSCs (line SC102A-1 and clone CRTDi011-A)
using feeder-free conditions. Briefly, 9,000 cells/well were seeded
into an ultra-low-attachment 96-well plate (Corning) in mTeSR™1
(StemCell Technologies) supplemented with 50 µM ROCK inhi-
bitor Y-27632 (StemCell Technologies). After 48 h, the medium
was changed to mTeSR™1 without ROCK inhibitor Y-27632. On
day 5, mTeSR™1 was replaced by neural induction medium
(DMEM/F12 (Gibco), 1% $N_2$ supplement (Gibco) 1% GlutaMAX
supplement (Gibco), 1% MEM non-essential amino acids (Gibco),
and 1 µg/ml heparin (Sigma-Aldrich)) and changed every
other day. Five days later, embryoid bodies were embedded in
Matrigel, medium was changed to differentiation medium (DMEM/
F12 and Neurobasal (Gibco, ratio 1:1), 0.5% $N_2$ supplement
(Gibco) 1% GlutaMAX supplement (Gibco), 0.5% MEM non-
essential amino acids (Gibco), insulin (2.875 ng/ml final concen-
tration Sigma-Aldrich), 1% B27 supplement (without vitamin
A, Gibco), 0.00035% 2-mercaptoethanol (Serva), and 1%
penicillin–streptomycin (Gibco)), and cerebral organoids were
further cultivated on an orbital shaker. Medium was changed every
other day. On day 16, medium was changed to differentiation

medium containing 1% B27 supplement with vitamin A (Gibco),
and organoids were cultivated under these conditions until day 30
and 50 after seeding.

## Fixation and cryosectioning of cerebral organoids

Cerebral organoids were fixed after 30 and 50 days in culture in 4%
paraformaldehyde (PFA) in 120 mM phosphate buffer (pH 7.4) for
1 h at room temperature and then washed in PBS for at least 12 h.
Organoids were then incubated overnight in phosphate-buffered
saline (PBS) containing 30% sucrose at 4 °C, embedded in Tissue-
Tek OCT (Sakura, Netherlands), and frozen on dry ice. Cryosec-
tions of 12 µm thickness were cut on a cryostat (Microm HM 560,
Thermo Fisher Scientific) and stored at –20 °C until further use.

## Fixation, paraffin embedding, and sectioning of fetal human brain tissue

Fetal human brain samples were collected in accordance with the
Helsinki Declaration 2000 and approval of the Ethical Committees of
the School of Medicine, University of Zagreb, and University Hospital
Centre Zagreb (641-01/19-02/01; 8.1-7/170-02; 02/21AG). The brain
specimens were obtained from fetuses following spontaneous abor-
tions (with 12 h postmortem delay, or less). The deaths were attributed
to respiratory distress syndrome or non-fetal causes. Two human
fetuses at the age of 12 pcw and two human fetuses at the age of 16 pcw
were clinically evaluated by a gynecologist and a pathologist and
included in this study. The fetuses showed no evidence of growth
restriction or developmental malformations; the fetal age was
estimated based on the crown-rump length, pregnancy records, and
assessment of histological sections. Whole fetal brains were fixed in 4%
PFA in 0.1 M PBS pH 7.4 for 48 to 96 h, depending on the brain size,
and afterwards embedded in paraffin according to standard
procedures. The brain tissue was cut into 20-µm-thick sections,
mount on glass slides, deparaffinized, and rehydrated prior to
immunohistochemistry as described below.

## Fixation and cryosectioning of fetal human brain tissue

Fetal human brain tissue was obtained from the Klinik und
Poliklinik für Frauenheilkunde und Geburtshilfe, Universitätskli-
nikum Carl Gustav Carus of the Technische Universität Dresden
with maternal agreement and permission of the Ethics Committee
at the TU Dresden. After pregnancy termination, the tissue was
directly transferred to the lab, and fragments of the cerebral cortex
were identified by morphology and fixed in 4% PFA in 120 mM
phosphate buffer pH 7.4 for 3 h at room temperature, followed by
24 h at 4 °C. The specimen was then incubated overnight in PBS
containing 30% sucrose at 4 °C and processed for cryosectioning as
described above for cerebral organoids.

## Immunohistochemistry

For immunohistochemical detection of βCYA and γCYA, cerebral
organoids and fetal brain sections were washed twice with PBS and
permeabilized with ice-cold methanol (–20 °C) for 5 min. After two
washing steps in PBS, samples were incubated in blocking buffer
(2% BSA/PBS) for 1 h. Primary antibodies were diluted in blocking
buffer and incubated for 1 h, followed by a 1 h incubation with

secondary antibodies also diluted in blocking buffer (Latham et al, 2018). Immunostained tissue sections were counterstained with DAPI (1:1000 in PBS) and were mounted in Mowiol (Millipore). Other immunohistochemical stainings were performed as previously described (Kanton et al, 2019) using antigen retrieval in 0.01 M sodium citrate buffer (pH 6.0) for 1 h at 70 °C. The following primary antibodies were used: mouse monoclonal IgG$_1$ anti-βCYA (Bio-Rad, clone 4C2, RRID:AB_2571580), mouse monoclonal IgG$_{2b}$ anti-γCYA (Bio-Rad, clone 2A3, RRID:AB_2571583), mouse anti-Caspase3 (abcam, #ab208161), CREST anti-centromere (kinetochore) (Antibodies Incorporated, #15-235, RRID:AB_2939059), rat anti-Ctip2 (abcam, #ab18465, RRID:AB_2064130), rabbit anti-Ki-67 (abcam, #ab15580, RRID:AB_443209), rabbit anti-Nestin (Sigma-Aldrich, #N5413, RRID:AB 1841032), mouse anti-pan-actin (Novus Biologicals, #NB600-535, RRID:AB_2222881), mouse anti-pan-cadherin (Sigma-Aldrich, #C1821, RRID:AB_476826), rat anti-PH3 (abcam, #ab10543, RRID:AB_2295065), goat anti-SOX2 (R + D Systems, #AF2018, RRID:AB_355110), rabbit anti-TBR2 (abcam, #ab23345, RRID:AB_778267), mouse anti-alpha-tubulin (Sigma-Aldrich, #T6199, RRID:AB_477583), rabbit anti-beta-tubulin (abcam, #ab179513, RRID:AB_3073861), rabbit anti-tubulin (Sigma-Aldrich, #T3526, RRID:AB_261659), mouse anti-Tuj1 (BioLegend, #801201, RRID:AB_2313773) and rabbit anti-ZO-1 (Invitrogen, #61-7300, RRID:AB_138452). All fluorescent-conjugated secondary antibodies were purchased from Thermo Fisher Scientific.

## Image acquisition

Images were acquired as Z-stacks of thirty 0.3-μm optical sections in the case of βCYA and γCYA immunostainings or of twelve 1-μm optical sections in the case of all other immunostainings using a Zeiss LSM 880 microscope with a 63×/1.3 LCI Plan-Neofluar, W/Glyc, DIC (Zeiss) objective. Prior to the image analysis, 10 adjacent slices in the center of the stack of the thirty 0.3-μm optical sections of the βCYA and γCYA immunostainings, and three adjacent slices in the center of the stack of the twelve 1-μM slices of all other immunostainings, were averaged. Overview images of the cerebral organoids were acquired using a Keyence's bench-top BZ-X810 all-in-one fluorescence microscope with 2× Plan-Apochromat, air, and 10x Plan-Fluor, air objectives (Keyence). Haze reduction function (Keyence) for the elimination of fluorescence blurring was used for the overview images of DAPI-stained cerebral organoids. Images to detect astral microtubules were acquired with an LSM 780 NLO laser scanning microscope, using a 63× Plan-Apochromat 1.4 N.A. oil objective (Carl Zeiss). The stacks acquired were of 512 × 512 pixels × 15–20 optical sections (xyz sampling: 0.09 × 0.09 × 0.75 μm). Images were analyzed with ImageJ (http://imagej.nih.gov/ij/) and Zen software (Carl Zeiss). The brightness and contrast of images were recorded and adjusted linearly.

## Analysis of the βCYA and γCYA distribution

Analysis of the βCYA and γCYA distribution was performed with custom Jython scripts written for Fiji (Schindelin et al, 2012). The input image stacks containing the γCYA isoform images in the magenta channel, the βCYA isoform images in the green channel, and an additional DAPI channel were first corrected for shifts caused by chromatic aberrations. Then, background intensity was subtracted per channel, and incompletely acquired images at the beginning and end of the stack were removed. These preprocessed image stacks were re-sliced along the $z$ axis for visualization purposes. To obtain the βCYA and γCYA distributions in the $x$–$y$ plane, 10 adjacent slices in the center of the stack were averaged. The gray-scale images to visualize the difference in localization pattern of γCYA and βCYA were obtained by first normalizing the intensities of each channel (histogram normalization). Then, the green channel signals were subtracted from the magenta channel signals, and a gray lookup table was applied. Thus, bright regions corresponded to areas in which γCYA was localized relatively strongly (compared to the localization distribution of βCYA). Conversely, in dark regions, βCYA was predominantly localized. Note that the generated images visualize relative localization patterns and do not quantify absolute concentrations of γCYA or βCYA.

In a second analysis step, the differences between the γCYA and βCYA localization patterns were quantified. Within each of the various zones (AJ, VZ, iSVZ, oSVZ, IZ, CP, MZ) of fetal human neocortical tissue, a region of interest (ROI) was defined for which the relative localization of βCYA and γCYA was compared. To this end, the ratio between the mean signal intensities of the respective ROI was calculated for each channel separately. Differences in the value of the ratio for the two channels then indicated a different localization preference of γCYA versus βCYA. For statistical analysis, intensity ratios of multiple images were obtained and compared with GraphPad Prism. The relative intensities of γCYA and βCYA were normalized to the mean fluorescence intensity of all positions.

## Quantifications

### Progenitor terminology
We use the term "VZ progenitors" as a collective term for both (i) the apical progenitors (APs) in the VZ, i.e., the canonical class of progenitors residing in the VZ, and (ii) AP-derived newborn BPs, which are also found in the VZ prior to their migration to the subventricular zone (SVZ). We restrict the term "APs" to those VZ progenitors that undergo mitosis at the ventricular surface, which is a defining criterion for APs.

### Quantification of cerebral organoid and VZ parameters
The contour of a cerebral organoid as it appeared in a bright-field image (see, for example, the top rows in Fig. 2A,B) was used to determine the cerebral organoid area. For determining the radial thickness of the VZ, the mean of 4 measurements of the VZ, made at orthogonal and opposite positions within a ventricle-like structure (see Fig. 4A), was calculated. The VZ perimeter was defined as the basal boundary of the VZ of a given ventricle-like structure. To calculate the area of the VZ, the area encompassed by the VZ perimeter was measured, and the area of the ventricular lumen (see Fig. 4A) was subtracted. For determining the number of VZ progenitors per ventricle-like structure and the cell density of the VZ, the number of DAPI-stained nuclei in the VZ of all ventricle-like structures of the number of cerebral organoids indicated in the Figure legends was counted. For quantifying these five parameters, the data of all ventricle-like structures per single organoid were averaged prior to calculating the mean for the organoids of a given condition. For quantification of cleavage plane

angles, the indicated number of mitotic APs in anaphase were analyzed from control iPSC- (lines SC102A-1 and CRTDi011-A) derived organoids, patient-like organoids from the mutated CRTDi011-A–derived clones CRTDi011-A–mut*ACTB*-1 and CRTDi011-A–mut*ACTB*-2, and patient-derived cerebral organoids (clones mut*ACTB*-1 and mut*ACTB*-2, and clones mut*ACTG1*-1 and mut*ACTG1*-2). The proportion of cycling cells in the VZ was defined as the percentage of DAPI-stained nuclei in the VZ that were Ki-67[+]. Mitotic cells in the VZ, i.e., mitotic APs, were expressed as the number of PH3[+]-stained cells per VZ area. All quantifications were performed blindly by using Fiji (Schindelin et al, 2012).

### Quantification of distinct populations of astral microtubules

Apical, basal, and central astral microtubules were stained, imaged, and analyzed as described (Mora-Bermudez et al, 2014). In short, comprehensive stacks of serial confocal sections (see "Image acquisition") were acquired and examined for each detected AP in metaphase. Each set of confocal sections of a metaphase cell soma was divided into three regions: an apical, a central, and a basal region. The apical region was defined from the apical surface lining the ventricle to the parallel plane just before the first centromere on the metaphase plate. The central region that followed was defined to contain all centromeres. The basal region extended after the last centromere until the basal end of the cell soma. All detectable astral microtubules that emanated from the two centrosomes and reached the cell periphery in each of the three regions were counted and classified accordingly.

## Conventional transmission electron microscopy (EM)

Electron microscope analysis was performed as previously described (Dubreuil et al, 2007; Meinhardt et al, 2014; Camp et al, 2015). Cerebral organoids were shortly pre-fixed in 0.2% glutaraldehyde in culture medium, before being fixed in 1% glutaraldehyde, 2% PFA in 0.1 M Pipes pH 7.4, 0.09 mM CaCl$_2$ for 1 h at room temperature. After washing steps, the samples were transferred into 4% low-melting agarose in PBS. Sections of 200 µm thickness through the cerebral organoids were cut on a vibratome (Leica VT1200S). Sections were post-fixed with 2% osmium tetroxide (OsO$_4$), 1.5% potassium ferrocyanide for 30 min at room temperature. Contrast was enhanced by subsequent incubation (without washing) with 1% OsO$_4$ for 1 h. After washing, the samples were stained en bloc with 0.5% uranyl acetate overnight at 4 °C. Samples were dehydrated by an ascending series of ethanol (15 min for each step) and three times in pure ethanol. After gradual infiltration with resin, the samples were flat-embedded in Epon replacement (Embed-812, Science Services) and polymerized at 60 °C.

Selected regions were re-mounted on resin stubs, and 70 nm-thick sections were cut on an ultramicrotome (Leica UCT). Sections were post-stained with uranyl acetate and lead citrate according to standard protocols. Images were acquired on a Tecnai 12 Biotwin transmission electron microscope (Thermo Fisher Scientific/FEI) at 100 kV with a TVIPS F416 CMOS camera and SerialEM software (Mastronarde, 2005). Image processing was performed in Fiji (Schindelin et al, 2012).

For electron tomography, 300 nm-thick sections were cut on an ultramicrotome (Leica UCT) and post-stained as described above.

Both sides of the section were coated with 15 nm colloidal gold beads (BBI solutions). Tilt series at +/− 64° with a 1° increment were acquired on a Tecnai F30 transmission electron microscope (Thermo Fisher Scientific/FEI) at 300 kV with a Gatan OneView camera and SerialEM software. Second axis tilt series were acquired after a 90° sample rotation. Electron tomograms were reconstructed using the IMOD software package. The arrangement of the adherens junctions (and centrioles/basal bodies) in the reconstructed tomograms was visualized by 3D-rendering with inverted grey values in ORS Dragonfly software (Comet Technologies Canada Inc., Montreal, Canada; software available at https://www.theobjects.com/dragonfly).

Quantification of the EM data shown in Fig. 6 was not performed due to technical limitations. The apical-most neuroepithelium in both wild-type and mutant organoids exhibits a highly complex three-dimensional architecture, making accurate measurements of cell circumference or cytoskeletal attachments at adherens junctions unreliable in single 2D sections. The orientation of individual sections (e.g., diagonal vs. transverse cuts) and potential membrane folds could not be determined with certainty. Furthermore, the resolution of the tomograms was in some cases insufficient to unambiguously identify individual actin filaments (~5 nm) or microtubules (~20 nm) within 300-nm sections. Reliable quantification of these elements would require volumetric imaging.

## Western blot analysis

Organoid protein lysates were obtained from organoid cell cultures after 30 days of culture using the RIPA lysis buffer system (Santa Cruz). Samples with a protein amount of 4 µg were prepared using 4× Protein Sample Loading Buffer (Li-Cor), denatured for 10 min at 70 °C, and loaded onto NuPAGE® Novex 4–12% Bis-Tris gels (Thermo Fisher Scientific) for separation. Separation was performed by running the gels for ~2.5 h at 90 V in 1× NuPAGE™ MOPS SDS running buffer (lower buffer chamber) and 1× NuPAGE™ MOPS SDS running buffer supplemented with NuPAGE® Antioxidant (upper buffer chamber) (Thermo Fisher Scientific). Proteins were transferred to iBlot™ PVDF Transfer Stacks (Thermo Fisher Scientific) for 6 min at 25 V. After blotting, membranes were dried at 37 °C for 10 min and rehydrated in TBS. For total protein detection, membranes were stained using Revert™ 700 Total Protein Stain (Li-Cor). After destaining, membranes were blocked in 5% milk (Roth) in TBST (1× TBS, 0.1% Tween-20) and incubated overnight at 4 °C while rotating. After three washes with 1× TBST, membranes were incubated with primary antibodies (mouse anti-human actin beta antibody, clone 4C2, or mouse anti-human actin gamma antibody, clone 2A3, Bio-Rad) at a 1:7500 dilution for 2 h at room temperature while rotating. Following three additional washes with 1× TBST, membranes were incubated in the dark for 1 h with a goat anti-mouse IgG H&L (IRDye® 800CW) secondary antibody (Li-Cor) at a 1:15000 dilution at room temperature while rotating. Washing with 1× TBST and 1× TBS was repeated. For visualization, a Li-Cor Odyssey® Imager was used. Image Lab (Bio-Rad) was used to analyze the protein bands.

## Single-cell RNA sequencing (scRNA-seq) analysis

For scRNA-seq of cerebral organoids, 3–6 organoids (depending on size) were used per iPSC line or clone, derived from one to three

independent batches, at 30 and 50 days of culture. For dissociation into single cells, the organoids were cut with a scalpel and washed three times with HBSS (Thermo Fisher Scientific) to remove apoptotic cells from the core. Subsequently, the organoids were incubated with Solution A (50 µL Enzyme P + 1900 µL Buffer X) from the Neural Tissue Dissociation Kit (P) (Miltenyi Biotec) for 15 min at 37 °C in a water bath. After the addition of Solution B (10 µL Enzyme A + 20 µL Buffer Y), the organoids were dissociated using a 1000 µL pipette tip and incubated for 10 min at 37 °C. The mechanical dissociation and incubation steps were repeated four times before filtering the cell suspension through a 30-µm filter and washing it three times with HBSS.

ScRNA-seq was then performed based on the 10x Genomics Single-cell transcriptome workflow (Zheng et al, 2017). Specifically, accurate cell numbers were obtained using a Neubauer Hemocytometer. Cell samples were carefully mixed with Reverse Transcription Reagent (Chromium Single Cell NextGem 3′ Library Kit v3.1) and loaded onto a Chromium Single Cell G Chip to reach a recovery of up to 10,000 cells per sample. The samples were processed further following the guidelines of the 10x Genomics user manual for single-cell 3′ RNA-seq v3.1. In short, the droplets were directly subjected to reverse transcription, the emulsion was broken, and cDNA was purified using silane beads (Chromium Single Cell 3′ Gel Bead Kit v2). After amplification of cDNA with 11 cycles, samples were purified with 0.6x volume of SPRI select beads to deplete DNA fragments smaller than 400 bp, and cDNA quality was monitored using the Agilent FragmentAnalyzer 5200 (NGS Fragment Kit). In total, 10 µL of the resulting cDNA was used to prepare scRNA-seq libraries, involving fragmentation, dA-tailing, adapter ligation, and 10–13 cycles of indexing PCR, following the manufacturer's guidelines. After quantification, libraries were sequenced on multiple Illumina NovaSeq 6000 S4 flow cells in 100 bp paired-end mode, aiming for 30,000 fragments per cell. The initial sequencing data were processed using the count command of the CellRanger software (v6.1.2) (https://support.10xgenomics.com/single-cell-gene-expression/software).
To build the reference, the human genome (hg38) and gene annotation (Ensembl 104) were used. The annotation was filtered using the mkgtf command of the CellRanger tool, as suggested by 10x Genomics, to include only genes with the types 'protein_coding', 'lincRNA', 'antisense', 'IG_LV_gene', 'IG_V_gene', 'IG_V_pseudogene', 'IG_D_gene', 'IG_J_gene', 'IG_J_pseudogene', 'IG_C_gene', 'IG_C_pseudogene', 'TR_V_gene', 'TR_V_pseudogene', 'TR_D_gene', 'TR_J_gene', 'TR_J_pseudogene', 'TR_C_gene'. The human genome and the filtered annotation were then used as input for the CellRanger mkref command to build a CellRanger reference.

The resulting expression tables were then analyzed using the Scanpy software (v1.9.1) following best-practice guidelines (Wolf et al, 2018; Luecken and Theis, 2019). For each 10x barcode, a Doublet Score was calculated using the Scrublet tool (Wolock et al, 2019). Quality control was performed, and only cells with at least 5000 reads, at least 2500 detected genes, a maximum of 30% of reads in the 50 most highly expressed genes, a maximum of 15% of reads in mitochondrial genes, and a Doublet Score of at most 0.25 were included in the analysis. This resulted in a dataset with 260,880 cells. Only genes expressed in at least 10 cells were used for further analysis.

All functions mentioned below are from Scanpy. Unless otherwise specified, standard parameters were used. CPM normalization and log-transformation were performed using the normalize_total and log1p functions (base=10). The 5000 most variable genes were selected using highly_variable_genes. A principal component analysis (PCA) was performed with the function pca. Sample integration was carried out with harmony_integrate (Korsunsky et al, 2019). On the Harmony-corrected principal components, a k-nearest neighbors (k-NN) network was computed with neighbors. This network was used for UMAP calculation with umap. Clustering was performed using Leiden (resolution=0.4), which resulted in 17 clusters. Marker genes for each cluster were computed using rank_genes_groups. The source code for the analysis is available upon request. The complete software stack is available as a Singularity container (https://gitlab.hrz.tu-chemnitz.de/dcgc-bfx/singularity/singularity-single-cell/container_registry/, v1.1.0).

## Statistical analysis

GraphPad Prism version 9.3.1 for Windows (GraphPad Software, La Jolla California USA, http://www.graphpad.com/) was used for graphical illustration and statistical analysis of all data. All data were tested for normal distribution (Shapiro–Wilk test) and further analyzed with unpaired *t* test, Mann–Whitney *U* test, one-way or two-way ANOVA, or Kruskal–Wallis test (*$P < .05$; **$P < .01$; ***$P < 0.001$; ****$P < 0.0001$).

# Data availability

There are no restrictions on the published data. All data are accessible in this article or in the supplementary material. The single-cell RNA sequencing data from this publication have been deposited in the German Human Genome-Phenome Archive (https://data.ghga.de/browse) and assigned the identifier GHGAS92787677252496.

The source data of this paper are collected in the following database record: biostudies:S-SCDT-10_1038-S44319-025-00647-7.

# Peer review information

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

## Acknowledgements

We thank J Peychl and his team of the Light Microscopy Facility at MPI-CBG for help with microscopy and the Light Microscopy Facility, a Core Facility of the CMCB Technology Platform at TU Dresden, for scanning of cryosections. We thank Christina Eugster Oegema and Jula Peters of the Organoid and Stem Cell Facility at MPI-CBG for the help in parallel culture of multiple organoid batches as well as sample preparation and Jana Meissner and Jula Peters for their help in sample preparation for EM. We also thank Ilka Reinhardt and Mihail Sarov of the Genome Engineering Facility at MPI-CBG for the generation of *ACTB* Thr120Ile mutant iPSCs from control CRTDi011-A iPSCs. We thank A. Dahl and his team of the DRESDEN-concept Genome Center for the single-cell RNA sequencing service and bioinformatical support. Additionally, we acknowledge Mateo Bastidas Betancourt and Sabrina Heide, both members of MH's lab, for their assistance with the analysis of the scRNA-seq data and TBR2 immunofluorescence, respectively. IN was a doctoral student jointly supervised by NDD and MH. ES was a fellow of MPI-CBG. Work in the NJM group (collection and research on the postmortem fetal human brains) was funded by

Croatian Science Foundation project IP-2024-05-4135 at the Scientific Centre of Excellence for Basic, Clinical and Translational Neuroscience (GA KK01.1.1.01.0007) funded by the European Union through the European Regional Development Fund. Collaboration between NDD and NJM was funded throughout COST Action NeuroMIG 16118. Work in the laboratory of WBH was supported by grants from the DFG (SFB 655, A2), the ERC (250197), and ERA-NET NEURON (MicroKin). Work in the NDD group was supported by grants from the DFG (DI 2170/3-1 and DI 2170/5-1), EJPRD JTC 2019 (PredACTINg; BMBF 01GM1922A), and Else-Kröner-Fresenius Stiftung (2020_EKES.04). MH was supported by an ERC starting grant (PRIMAZINC, 101039421).

## Author contributions

**Indra Niehaus**: Data curation; Investigation; Visualization; Methodology; Writing—original draft. **Michaela Wilsch-Bräuninger**: Investigation; Methodology; Writing—original draft. **Felipe Mora-Bermúdez**: Validation; Investigation; Methodology; Writing—review and editing. **Fabian Rost**: Investigation; Methodology. **Mihaela Bobic-Rasonja**: Resources; Investigation; Visualization. **Velena Radosevic**: Resources. **Marija Milkovic-Perisa**: Resources. **Pauline Wimberger**: Resources. **Mariasavina Severino**: Resources; Visualization. **Alexandra Haase**: Resources. **Ulrich Martin**: Resources. **Karolina Kuenzel**: Resources. **Kaomei Guan**: Resources. **Katrin Neumann**: Resources. **Noreen Walker**: Methodology. **Evelin Schröck**: Writing—review and editing. **Natasa Jovanov-Milosevic**: Resources; Supervision; Funding acquisition; Methodology. **Wieland B Huttner**: Conceptualization; Supervision; Funding acquisition; Methodology; Writing—original draft; Writing—review and editing. **Nataliya Di Donato**: Resources; Supervision; Funding acquisition; Writing—original draft; Project administration. **Michael Heide**: Conceptualization; Supervision; Funding acquisition; Methodology; Writing—original draft; Writing—review and editing.

Source data underlying figure panels in this paper may have individual authorship assigned. Where available, figure panel/source data authorship is listed in the following database record: biostudies:S-SCDT-10_1038-S44319-025-00647-7.

## Disclosure and competing interests statement

The authors declare no competing interests.

# Expanded View Figures

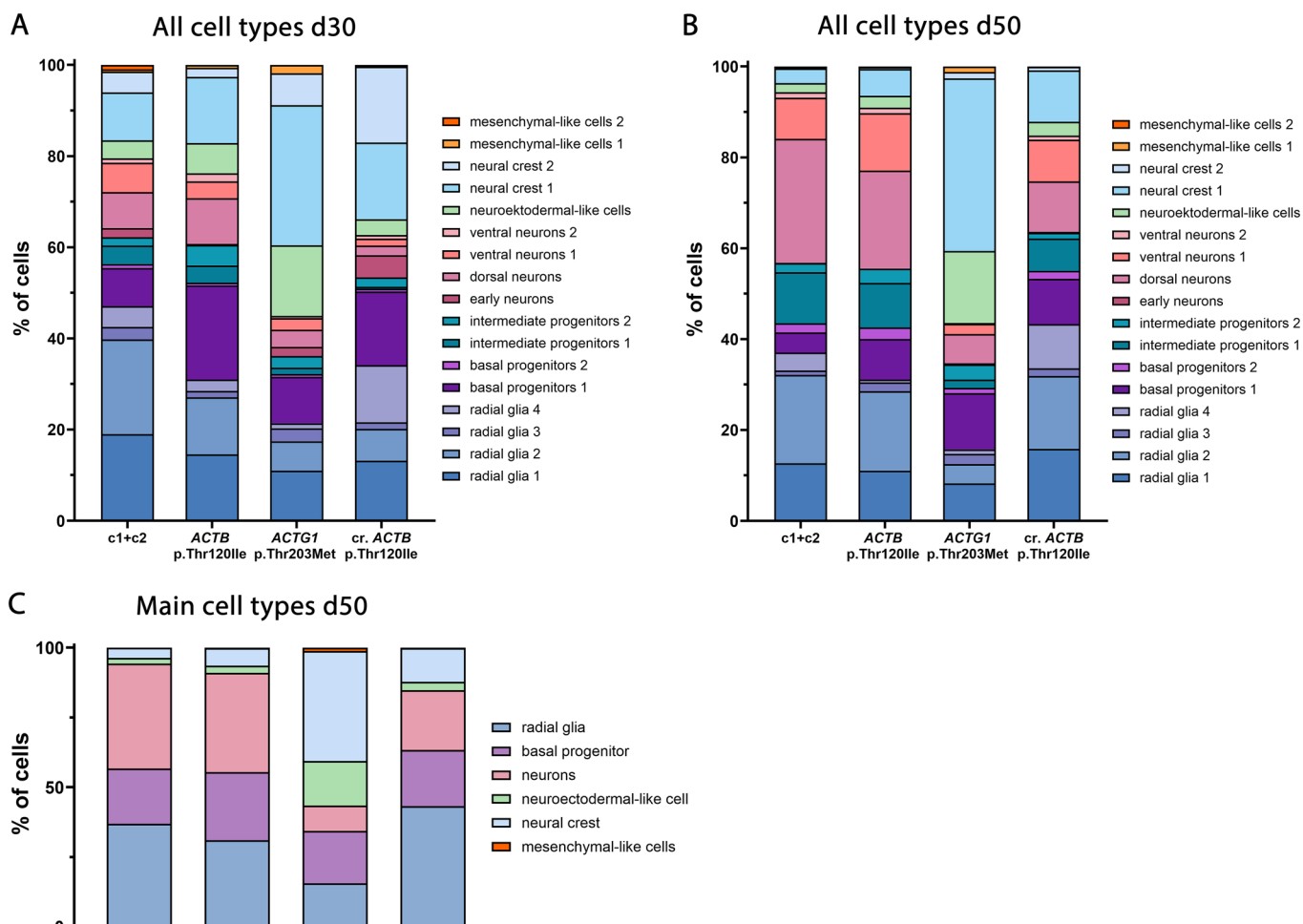

Figure EV1. Cell type composition of control and BWCFF-S cerebral organoids on culture day 30 and 50.

(A, B) Quantification of the cell type composition in percent of scRNA-seq data from control (c1, SC102A-1 and c2, CRTDi011-A;), BWCFF-S *ACTB* Thr120Ile, BWCFF-S *ACTG1* Thr203Met, and cr. *ACTB* Thr120Ile cerebral organoids at culture day 30 (A) and 50 (B). Specification of the color-coded cell types on the right side of the panel. (C) Quantification of the composition of the combined cell types (i.e., radial glia, basal progenitors, neurons, neuroectodermal-like cells, neural crest and mesenchymal-like cells) in percent of scRNA-seq data from control (c1, SC102A-1 and c2, CRTDi011-A;), BWCFF-S *ACTB* Thr120Ile, BWCFF-S *ACTG1* Thr203Met, and cr. *ACTB* Thr120Ile cerebral organoids at culture day 50. Specification of the color-coded cell types on the right side of the panel.

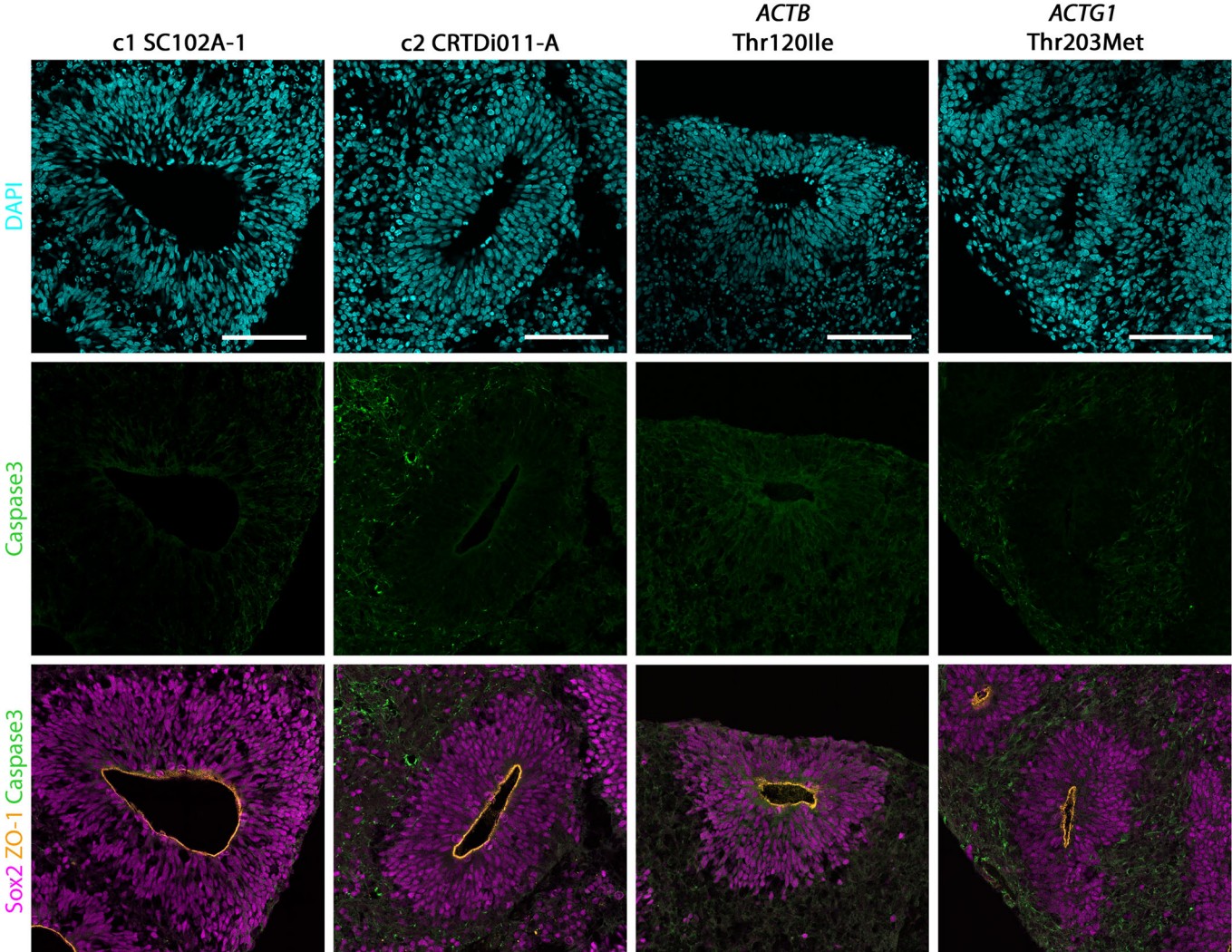

**Figure EV2. No difference in apoptosis between control and BWCFF-S cerebral organoids.**

Triple immunofluorescence for Caspase 3 (green), SOX2 (magenta) and ZO-1 (orange), combined with DAPI staining (blue), of sections of control (c1, SC102A-1 and c2, CRTDi011-A; two left columns), BWCFF-S *ACTB* Thr120Ile (second column from right) and BWCFF-S *ACTG1* Thr203Met (right column) 30-day-old cerebral organoids showing ventricle-like structures. Scale bars, 100 μm.

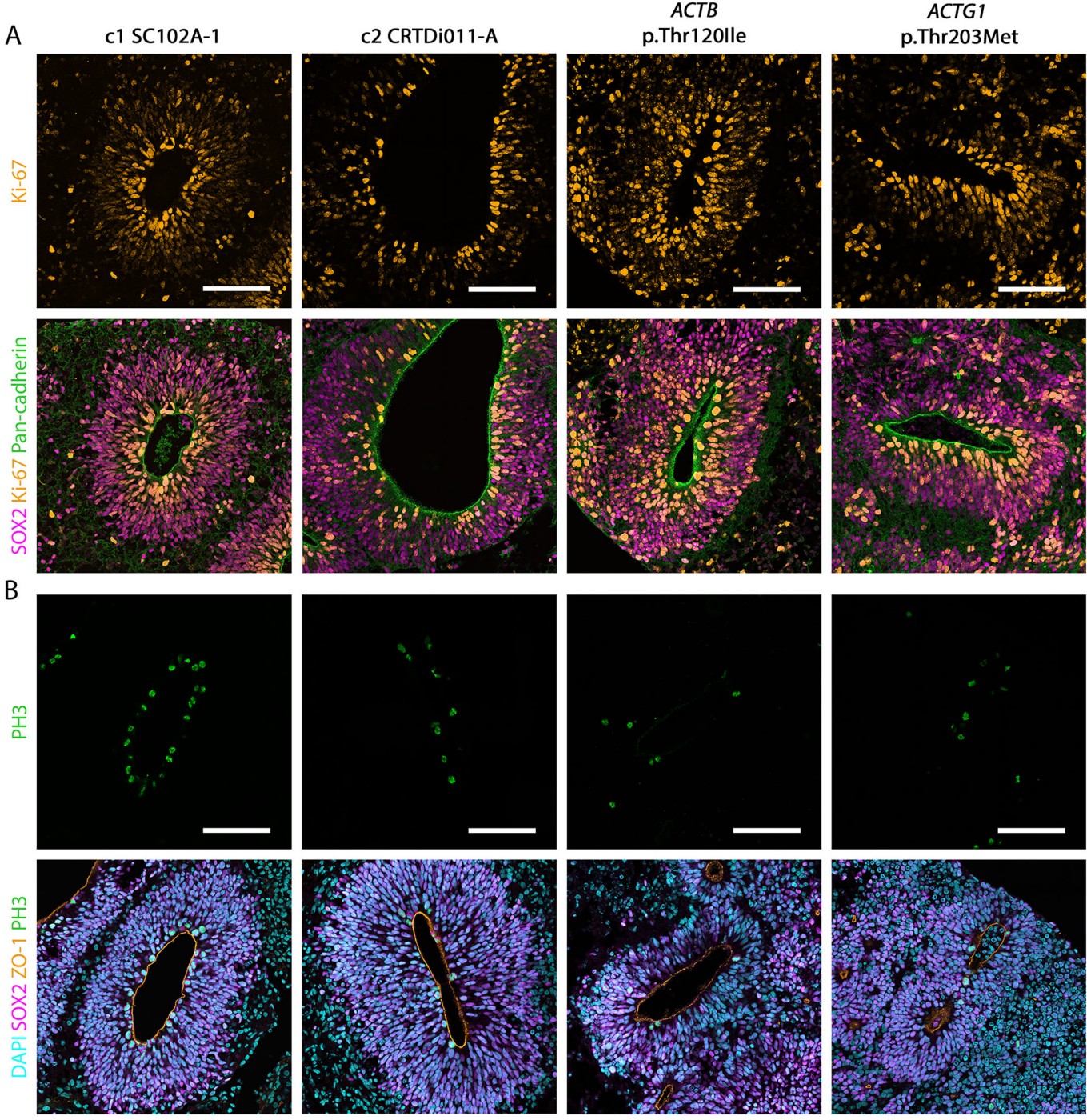

**Figure EV3.   No difference in the number of cycling VZ progenitors and mitotic APs between control and BWCFF-S cerebral organoids.**

(**A**) Triple immunofluorescence for Ki-67 (orange), SOX2 (magenta) and pan-cadherin (green) of sections of control (c1, SC102A-1 and c2, CRTDi011-A; two left columns), BWCFF-S *ACTB* Thr120Ile (second column from right) and BWCFF-S *ACTG1* Thr203Met (right column) 30-day-old cerebral organoids showing ventricle-like structures. Scale bars, 100 μm. (**B**) Triple immunofluorescence for SOX2 (magenta), phosphohistone H3 (PH3, green) and ZO-1 (orange), combined with DAPI staining (blue), of sections of control (c1, SC102A-1 and c2, CRTDi011- A; two left columns), BWCFF-S *ACTB* Thr120Ile (second column from right) and BWCFF-S *ACTG1* Thr203Met (right column) 30-day-old cerebral organoids showing ventricle-like structures. Scale bars, 100 μm.

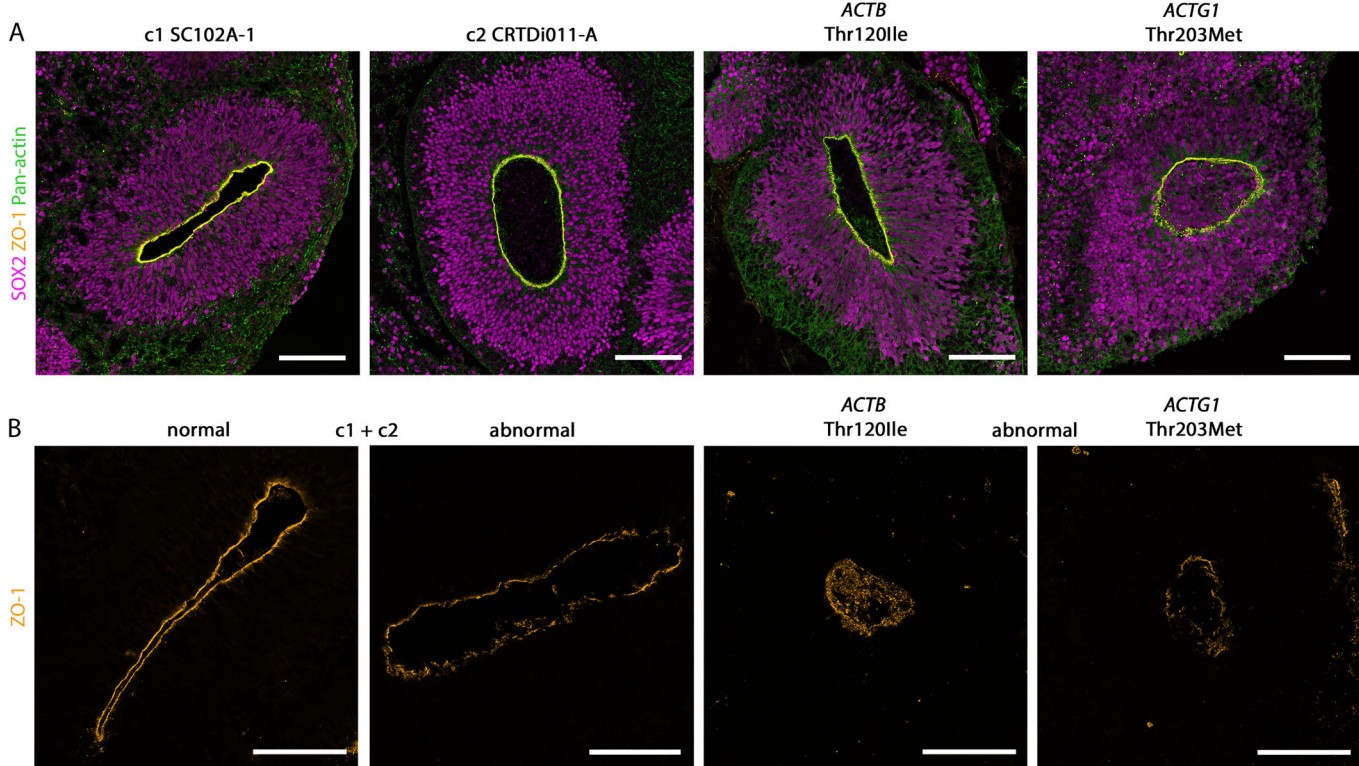

**Figure EV4. Morphology of apical junctional complexes in control and BWCFF-S cerebral organoids.**

(**A**) Triple immunofluorescence for SOX2 (magenta), ZO-1 (orange) and pan-actin (green) of sections of control (c1, SC102A-1 and c2, CRTDi011-A; two left panels) and BWCFF-S *ACTB* Thr120Ile (second panel from right) and *ACTG1* Thr203Met (right panel) 30-day-old cerebral organoids; Scale bars, 100 µm. (**B**) Exemplary images of ZO-1 immunofluorescence of sections of control (c1, SC102A-1 and c2, CRTDi011-A; two left panels), BWCFF-S *ACTB* Thr120Ile (second panel from right) and *ACTG1* Thr203Met (right panel) 30-day-old cerebral organoids showing normal (left panel) and abnormal (second, third and fourth panel from left) adherent junction belt morphology. Scale bars, 100 µm.

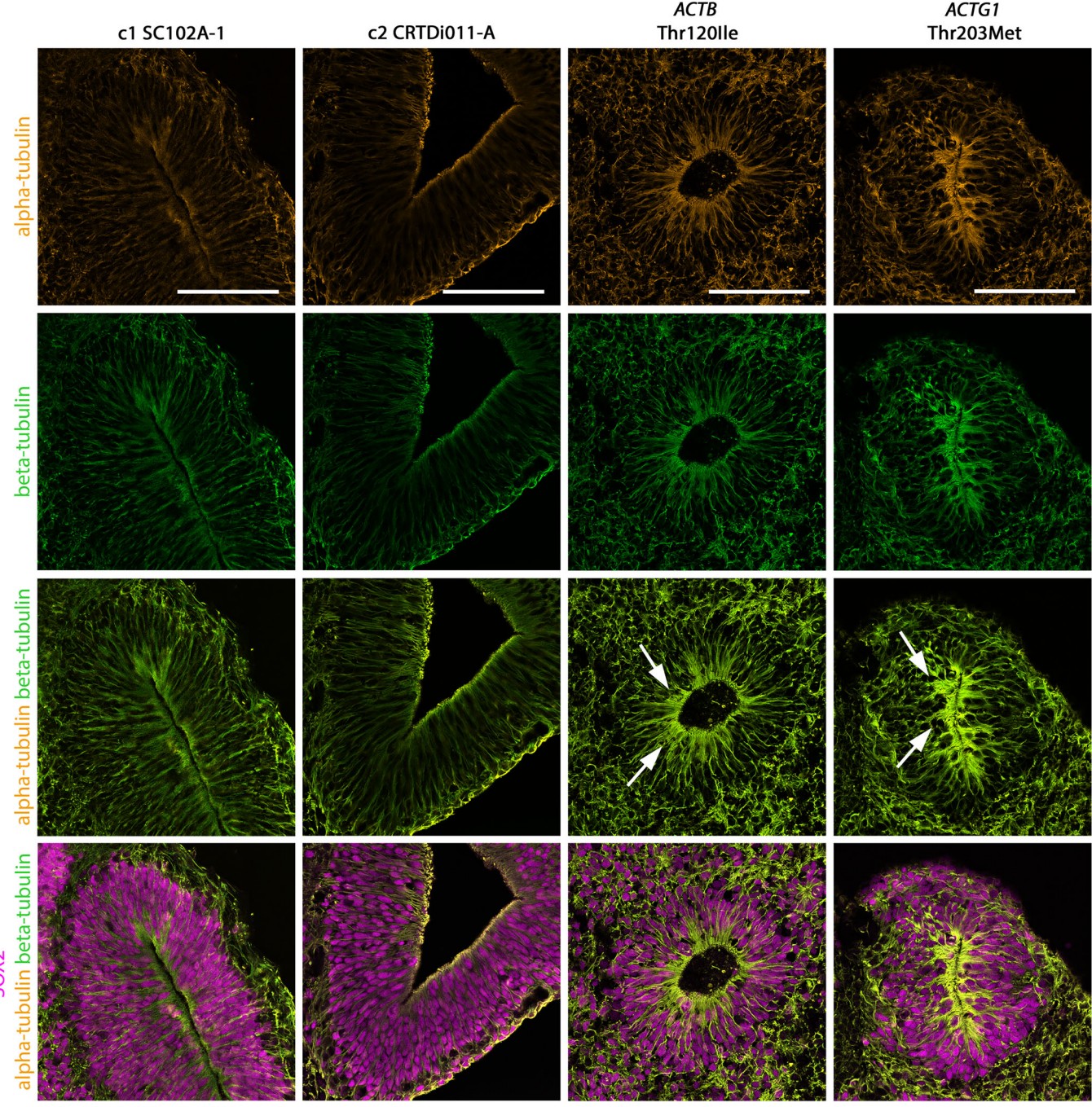

**Figure EV5. BWCFF-S cerebral organoids show an increased localization of alpha- and beta-tubulin at the apical cell cortex.**

Triple immunofluorescence for alpha-tubulin (orange, shown in rows 1, 3 and 4), beta-tubulin (green, shown in rows 2–4) and SOX2 (magenta, shown in row 4) of control (c1, SC102A-1 and c2, CRTDi011-A; two left columns), BWCFF-S *ACTB* Thr120Ile (second column from right) and BWCFF-S *ACTG1* Thr203Met (right column) 30-day-old cerebral organoids. Note the strong alpha-tubulin and beta-tubulin fluorescence signal at the apical cell cortex of BWCFF-S VZ progenitors (white arrows); Scale bars, 100 µm.

