## [Peer Review File · EMBO Reports]

Cerebral organoids expressing mutant actin genes reveal cellular mechanism underlying microcephaly

Indra Niehaus, Michaela Wilsch-Bräuninger, Felipe Mora-Bermúdez, Fabian Rost, Mihaela Bobić Rasonja, Vlena Radošević, Marija Milković Periša, Pauline Wimberger, Mariasavina Severino, Alexandra Haase, Ulrich Martin, Karolina Kuenzel, Kaomei Guan, Katrin Neumann, Noreen Walker, Evelin Schrock, Nataša Jovanov-Milošević, Wieland Huttner, Nataliya Di Donato, and Michael Heide

Corresponding author(s): Michael Heide (Mheide@dpz.eu), Wieland Huttner (huttner@mpi-cbg.de), Nataliya Di Donato (didonato.office@mh-hannover.de)

Review Timeline:

Submission Date:	14th Jun 24
Editorial Decision:	20th Sep 24
Resubmission Consultation:	10th Oct 24
Invitation to Resubmit:	23rd Oct 24
Revision Received:	10th Apr 25
Editorial Decision:	12th Aug 25
Revision Received:	25th Sep 25
Accepted:	6th Nov 25

Editor: Bernd Pulverer

Transaction Report:

Dear Dr. Heide

Thank you for the submission of your research manuscript to EMBO Reports. I apologize for the excessive delay in sending this decision and thank you for your patience. We struggled to obtain two thorough reports over the summer holiday season but, given significant criticisms raised in the first two reports, decided to obtain a third report to ensure that the decision would be fair and optimally informed.

As you will see, despite the basic interest in the approach and clear support for the overall project, the three expert referees all recommend against publication, at least at this time.

Briefly, Ref 1 raises no major technical issues, but suggests further mechanistic analysis and notes some crucial overstatements. The referee suggest you present a 'nice tool', but suggests the data remain 'superficial' and 'premature': 'a handful of anatomical and cell biological characterizations are performed, based mostly on immunohistochemical stains...one cannot really draw any conclusion as to the mechanism of action of the genes mutated, nor that the cellular phenotypes observed really explain the complex malformations observed in the patients...'.

The report suggests to test some of the mechanistic hypotheses proposed, in particular to address two current overstatements: "The present study provides crucial insight into the pathogenesis (...)" & "suggests an increase in the delamination of cells from the VZ in BWCFF-S cerebral organoids, providing a likely underlying mechanism for the reduction in the size of the VZ progenitor pool of these organoids".

We agree with the referee that in particular the last sentence would need to be supported by definitive data, as also noted by the other referees (see below).

Further, referee 1 recommend to investigate further the claim "...the decrease in VZ size does not appear to be compensated by a corresponding size increase of non-VZ tissue" - again as noted by the referee below.

Ref 2: notes the work is valuable and in principle suitable for the broader readership of EMBO Report, but again that the work is too preliminary without mechanistic insights: 'key data is poor and thin... the message of why and how the progenitors are lost and what kind of cell types are indeed produced instead of the progenitors is missing'.

More specifically, referee suggests to add transcriptomics data (sc or bulk), whole-mount imaging, and either WB to quantify actins between organoids or a live actin dye to calculate the uptake.

In line with the other referee, s/he requires improvements how VZ progenitor proliferation are measured.

Ref 3 notes 'The subject is very interesting, and the phenotypes reported appear clear' but substantial improvement of the quantifications are needed:

- show at least three independent replicas.
- Measure many more cells for spindle orientation (fig 4).
- Improve VZ definition (cf. ref 2)
- What becomes of delaminated progenitors (cf. ref 1).
- Effect of mutation on actin.

in our view these revision would take the project to a new level and we cannot expect them for a 'realistic revision', which we define as 3-4 month experimental work. if you feel you have in the meantime progressed the project and can address all of the experimental deficits listed above (and a couple of specific additional one in the referee report eg on statistics), we are open to discuss a future resubmission here or at our partner journal Life Science Alliance.

I am sorry to disappoint you on this occasion, and hope that the referee comments are helpful in your continued work in this area.

Yours sincerely

Bernd Pulverer

Referee #1:

This manuscript aims at understanding the developmental mechanisms underlying the brain defects observed in Baraitser-Winter-CerebroFrontoFacial syndrome (BWCF-S), characterized by a combination of microcephaly and malformations of the frontal cerebrum, such as periventricular heterotopia or pachygyria (loss of folds). The study is an analysis of cerebral organoids generated with patient-derived iPSC lines, which is nice but unfortunately a bit too superficial. Organoids are cultured for one fixed period (30 days) only, and then a handful of anatomical and cell biological characterizations are performed, based mostly on immunohistochemical stains. The authors show that organoids derived from patient cells, and also from hiPSCs from healthy donors genetically modified to carry only the disease point mutation, have reduced overall size, reduced size of the germinal tissue and reduced number of cells in it, and change in cleavage plane orientation of apical germinal cells. Using electron microscopy, the authors also show some disorganization of subcellular aspects of these apical germinal cells. The authors conclude that their study provides crucial insight into the pathogenesis of BWCF-S.

The authors have generated a nice tool, and overall I see nothing wrong with the data presented here, which shows that their organoids have merit to be an interesting model of this syndrome. However, in my opinion the study is too premature for publication. Characterizations are well performed and the results are clear, but one cannot really draw any conclusion as to the mechanism of action of the genes mutated, nor that the cellular phenotypes observed really explain the complex malformations observed in the patients. In Discussion, the authors present multiple possibilities and hypotheses as to the mechanisms driving brain developmental malformations in these patients, and yet no mechanistic experiments are performed beyond basic characterization of mutant organoids. It would improve the manuscript very significantly if mechanistic experiments were performed to test some of these ideas.

These are some of my specific comments on the manuscript:

In line 320, the authors conclude: "This change in cleavage plane orientation no longer allows most APs to undergo symmetric-proliferative divisions to increase their pool size, and strongly suggests an increase in the delamination of cells from the VZ in BWCF-S cerebral organoids, providing a likely underlying mechanism for the reduction in the size of the VZ progenitor pool of these organoids." - The concept of cleavage plane orientation dictating the fate of daughter cells is strong and valid in *Drosophila*, but it has been long since it was shown not so in mammals. Mounting studies over the last 20 years (Noctor, Kriegstein, Knoblich, Huttner, Götz, Borrell) have shown that mitotic plane orientation does not strictly predict nor dictate daughter fate in the mammalian cerebral cortex, so the first part of the above statement is not true. The same is true for delamination of apical cells, which is under strong regulation but not directly by mitotic plane orientation. Therefore, the last part of the statement is merely speculative.

Line 353: "we detected several cytoskeletal and morphological irregularities in VZ progenitors of BWCF-S cerebral organoids, notably in the apical region of these cells. These irregularities could contribute to the abnormal cleavage plane orientation of BWCF-S mitotic APs (see discussion for details)." - This is a rather weak argument at this point, as the analysis of astral microtubules revealed no differences in the mitotic spindles and associated structures.

Discussion begins with an overstatement: "The present study provides crucial insight into the pathogenesis (...)" - no real insight is provided beyond observations of changes in organoid growth and some cell biology.

The authors also emphasize the appropriateness of their organoid model to model the pathogenesis of cortical malformations observed in their patients. They only show that their model may be appropriate to model the microcephaly aspect, but not other patient defects shown in Fig 1. It is the combination of these defects, and not one at a time, that characterize this disease.

In Discussion, the authors state that "the decrease in VZ size does not appear to be compensated by a corresponding size increase of non-VZ tissue" - The authors should investigate this: Why there is not an increase in size of non-VZ tissue? What happens to delaminated progenitors? Do they exit cell cycle? Is the cell cycle of delaminated progenitors blocked somehow? If delaminated progenitors remain in cell cycle, there is no reason for microcephaly. If delamination causes cell cycle exit, then one should see increased neurogenesis transiently in mutant organoids, while in WT organoids progenitors continue amplifying...

Data shown in Fig 5D is very unclear. What are the white arrows pointing at in Thr120Ile? What are the arrowheads pointing at in all three images? These are all different structures and nothing seems like microtubule bundles.

For some unknown reason, there are lots of data in supplementary figures, which is unnecessary given the small size of several main figures. Accordingly, data in Suppl Fig 7 should be included in main Fig 3, and Suppl Fig. 8 should be part of main Fig 4.

ANOVA test is not appropriate for quantifications in Fig 4D, because values are only from 0 to 100. Chi-square should be used instead.

Referee #2:

The work by Niehaus et al. identified mutations in two common Actins and has generated cerebral organoids and decoded the potential mechanisms of microcephaly, a developmental disorder mainly stemming from the depletion of actively proliferating neural progenitors. This work adds to the current understanding of genetic mutations causing cortical abnormalities. It is undoubtedly a valuable work for the community and is suitable for a broader readership of EMBO Report. While the concept and the finding are straightforward, the work is too preliminary as it is presented. Most importantly, the mechanistic insights are missing. In precise terms, why/how actin mutations affect the progenitors in the developing brain organoids. As far as I see, the most striking data is calculating the kinetics of progenitor proliferation. The paper is built on it (in this aspect, the amount of key data is poor and thin). However, this data is too weak to make a conclusion and the message of why and how the progenitors are lost and what kind of cell types are indeed produced instead of the progenitors is missing. Such mechanistic findings are critical, and without those, this paper does not add anything incremental to what is already known. The following are my comments, and the paper needs to address them to be able to get published.

Main comments:

Organoid images (including those in supplementary figures) show only a specific area, presumably the most striking region showing the typical cytoarchitecture. This approach induces a bias in determining the organoid quality. Selected images should show the whole section of the organoid, demonstrating the apicobasal polarity. Alternatively, whole-mount imaging will also be helpful.

The authors cannot determine the cell type composition by showing SOX2, CTIP2, and Nestin. This requires transcriptomics data, either from single cells or from the bulk. Only a handful of markers have been used to conclude the disease mechanisms. This is insufficient.

In supplementary figure 5, the authors again show a selected VZ and claim that the patient-derived organoids show a similar architecture and cell types to healthy controls. Why should healthy controls show massive TUJ1-positive neurons at the apical region? I don't see a difference between control and mutants in this aspect. From this, it seems that the organoid culturing method must have used a neural differentiation agent (perhaps retinoic acid or BDNF, etc.), which will trigger the differentiation of NPCs, masking the actual phenotypes between the healthy control and mutant. Cell types should be quantified (Here, TBR2).

IF is not quantitative. One should use either WB to quantify actins between organoids or a live actin dye to calculate the uptake.

Looking at the organoid sizes, it is unclear how one could calculate the growth rate by measuring the diameter (which is again interpreted as area). It is enough to say a panel of diameter size differences. It should not be significant data needing so much discussion and one dedicated figure.

Measuring VZ thickness and size remains problematic. SOX2 or PAX 6 should be used along with DNA to define the boundary. A whole section of the brain organoid should be shown, and the number of VZ and size should be compared across the genotypes.

Assuming the VZ are smaller in mutant organoids, the authors propose a model that could reduce progenitor proliferation. However, KI67 index does not vary. In this case, how do the authors reconcile the differences between VZ and the loss of progenitors? As the paper is mainly centered around this analysis, this figure must be expanded to show the effect in all genotypes. One must consider only ana or telophase cells as meta phase cells are not settled well. SOX2 alone can fully mislead, and p-vimentin should be used for clarity. Qualification should be given for horizontal, vertical, and oblique.

I like the idea of calculating the kinetics of AP's division, whose division plane is predominantly horizontal in healthy VZ, where it is attributed to the symmetric expansion. I prefer the widely used term "division plane." However, this figure has numerous issues. What is the genotype shown in Fig 4A?

I appreciate the inclusion of isogenic controls. However, the data is too minimal, and no detail about genetic engineering, mutation induction, VZ analysis, etc., is shown. In light of these, it is difficult to determine what causes the observed phenotype. To make the authors' claim concrete, the isogenic and accompanying rescue experiment must be done thoroughly.

Minor comments:

Mutations in various Actins cause actinopathies. Thus, it would be useful to see a graphical summary of different actins, localization of mutations, and which phenotypes are associated with which mutations in actin.

Terminological issues should be avoided. Authors stained nestin and claim radially organized VZ which is technically wrong. What they probably mean is radial fibers and radially organized progenitors.

If the authors have carried out Karyotyping, show it. Also, the uniform distribution of pluripotent markers.

Again terminological issues: " β CYA and γ CYA across the 217 various zones of the wall of the ventricle-like structures (Supplementary Fig. 3)" What are the various zones here? What does here look like Ventricle?

I am curious, how would one distinguish different actins? Are there specific antibodies?

Referee #3:

In this article, Niehaus and colleagues use patient derived cerebral organoids to model mutations in two actin genes, ACTB and ACTG1, that in patients lead to Baraitser-Winter-CerebroFrontoFacial syndrome. Most patients affected by this pathology display cortical malformations, including microcephaly and lissencephaly. Mostly focusing on the reduced brain size phenotype, the authors demonstrate that patient derived cerebral organoids are smaller, with smaller ventricular-like zone. The mitotic spindle of apical radial glial cells is observed to be tilted as compared to controls, with more horizontal cleavage planes. This is proposed to be the cause for the loss of apical progenitors and the reduction of the size of the ventricular-like zone. Electron microscopy reveals morphological abnormalities in the apical domain of these cells, as well as increased microtubule density, also observed by immunofluorescence. Mechanistically, the authors propose that these actin mutations somehow affect the apical domain leading to perturbed mitotic spindle positioning and detachment of the progenitors. The subject is very interesting, and the phenotypes reported appear clear. However, several issues must be addressed for publication, including a substantial improvement of the quantifications.

1/ Figures should be represented as experimental replicates, not as individual organoids coming from the same experiments. In its current form, data points represented on the graphs appear to be individual organoids, and "replicas" (circles and triangles) are different clones. In the control, these are different iPS lines. This is a very odd way to represent the data and raises concerns about the statistical analysis (different iPS lines cannot be pooled to get a control value). There should be at least three real experimental (independent) replicas, per clone and per genotype.

2/ A major claim of the study concerns mitotic spindle orientation, and yet very few cells are analyzed (Figure 4). Only 16 to 29 anaphase cells were analyzed per clones in the mutant organoids. Judging from the number of mitotic cells per lumen, and the number of lumens per organoid, it is surprising that such a low number of cells was measured.

3/ In figure 3, it is not clear how the authors can define the VZ, based solely on DAPI staining. The ROIs in figure 3A appear to be quite subjectively defined, especially in the mutants. Why not use a PAX6 or SOX2 staining? Especially, because the authors report very nice and clear SOX2 staining in figure S7. At least, the authors should add to their quantifications the apical size of the lumens, which is the only metric that is independent of the positioning of the basal boundary of the VZ.

4/ Figure 5 is very interesting, but nothing is quantified. This should be done if the authors want to make these claims.

5/ The authors are careful not to over conclude about the consequence of spindle rotation, only mentioning loss of apical progenitors. Still, one wonders what becomes of these delaminated progenitors. Do they become oRG/bRG-like cells or do they differentiate into IPs and neurons? The authors mention the presence of Tuj1 and TBR2 in the mutants, but again this is not quantified. If apical loss instead leads to oRG/bRG-like cell generation, this should be discussed.

6/ The effect of the mutation on the actin monomers and on f-actin are not characterized. At least, it should be tested whether the mutations destabilize the protein. Are these mutations loss of functions, or do they have a dominant effect? Do these two actin isoforms have overlapping functions?

Minor

1/ Previous reports have shown that the human mitotic spindle of human ventricular RG cells was naturally tilted, as compared to what is seen in mice, leading to the generation of oRG/bRG cells. This does not appear to be seen in the present study (control situation). Is this due to a difference in stage analyzed?

2/ (Line 175) Could the authors use pcw instead of wpc, to be coherent with the literature in the field?

3/ Figure S7. Please represent the PH3 data as a fraction of total cells in VZ (as done for Ki67) in order to report a mitotic index.

** As a service to authors, EMBO Press provides authors with the ability to transfer a manuscript that one journal cannot offer to publish to another journal, without the author having to upload the manuscript data again. To transfer your manuscript to another EMBO Press journal using this service, please click on

Link Not Available

Dear Dr. Pulverer,

Thank you for your decision on our manuscript. We greatly appreciate the clear support from the reviewers for the overall project. Since the initial submission, the project has progressed significantly, as detailed below and in our attached point-by-point response. As you will see, we have addressed all the reviewers' concerns and outlined how these changes would be incorporated into a revised manuscript.

In our view, the reviewers' critiques can be summarized into two main points:

1. Expanding the methodological scope of our analysis by performing single-cell RNA sequencing experiments.
2. Providing a mechanism for the reduction in VZ progenitors, or in other words, identifying the fate of the delaminated daughter cells of the VZ progenitors.

Regarding these key points, we have generated a single-cell RNA-seq dataset from day 30 and day 50 cerebral organoids for all genotypes in the period between submission and the decision. This dataset is currently under analysis and will offer more detailed insights into the cell-type composition of both control and BWCF-S cerebral organoids at two developmental stages. This will not only expand the methodological scope of our analysis but also provide a clearer understanding of progenitor fate and lineage. Additionally, we are currently analyzing progenitor fate using markers for various progenitor populations (such as basal intermediate progenitors and basal radial glia) and neurons. This will enable us to better assess the fate of the delaminated daughter cells of the VZ progenitors and further elucidate the mechanism behind the reduction in VZ progenitor numbers.

Further details on how we have addressed the reviewers' concerns are provided in the attached document.

We are confident that we can submit a revised and significantly improved manuscript within a reasonable timeframe, requiring approximately 3-4 months for additional experimental work.

With best regards, on behalf of all corresponding authors,

Michael Heide

Referee #1:

Referee's comment:

This manuscript aims at understanding the developmental mechanisms underlying the brain defects observed in Baraitser-Winter-CerebroFrontoFacial syndrome (BWCF-F-S), characterized by a combination of microcephaly and malformations of the frontal cerebrum, such as periventricular heterotopia or pachygyria (loss of folds). The study is an analysis of cerebral organoids generated with patient-derived iPSC lines, which is nice but unfortunately a bit too superficial. Organoids are cultured for one fixed period (30 days) only, and then a handful of anatomical and cell biological characterizations are performed, based mostly on immunohistochemical stains. The authors show that organoids derived from patient cells, and also from hiPSCs from healthy donors genetically modified to carry only the disease point mutation, have reduced overall size, reduced size of the germinal tissue and reduced number of cells in it, and change in cleavage plane orientation of apical germinal cells. Using electron microscopy, the authors also show some disorganization of subcellular aspects of these apical germinal cells. The authors conclude that their study provides crucial insight into the pathogenesis of BWCF-F-S.

The authors have generated a nice tool, and overall I see nothing wrong with the data presented here, which shows that their organoids have merit to be an interesting model of this syndrome. However, in my opinion the study is too premature for publication. Characterizations are well performed and the results are clear, but one cannot really draw any conclusion as to the mechanism of action of the genes mutated, nor that the cellular phenotypes observed really explain the complex malformations observed in the patients. In Discussion, the authors present multiple possibilities and hypotheses as to the mechanisms driving brain developmental malformations in these patients, and yet no mechanistic experiments are performed beyond basic characterization of mutant organoids. It would improve the manuscript very significantly if mechanistic experiments were performed to test some of these ideas.

Authors' response:

We appreciate the positive evaluation provided by the Reviewer. In our responses below, we address the Reviewer's concerns and explain how these will be reflected in the revised manuscript.

Referee's comment:

These are some of my specific comments on the manuscript:

*In line 320, the authors conclude: "This change in cleavage plane orientation no longer allows most APs to undergo symmetric-proliferative divisions to increase their pool size, and strongly suggests an increase in the delamination of cells from the VZ in BWCF-F-S cerebral organoids, providing a likely underlying mechanism for the reduction in the size of the VZ progenitor pool of these organoids." - The concept of cleavage plane orientation dictating the fate of daughter cells is strong and valid in *Drosophila*, but it has been long since it was shown not so in mammals. Mounting studies over the last 20 years (Noctor, Kriegstein, Knoblich, Huttner, Götz, Borrell) have shown that mitotic plane orientation does not strictly predict nor dictate daughter fate in the mammalian cerebral cortex, so the first part of the above statement is not true. The same is true for delamination of apical cells, which is under strong regulation but not directly by mitotic plane orientation. Therefore, the last part of the statement is merely speculative.*

Authors' response:

The Reviewer is correct that the *Drosophila* paradigm cannot be transferred one-to-one to mammalian apical progenitors (APs). Regarding the latter, the consensus among most researchers is as follows (see Mora-Bermúdez & Huttner MBoC 2015). A vertical cleavage plane is associated with either a symmetric proliferative division (1 AP → 2 APs) or an asymmetric self-renewing division (1 AP → 1AP + 1 non-AP (newborn basal progenitor or neuron, which eventually delaminate)). An oblique cleavage plane is typically associated with an asymmetric self-renewing division (1 AP → 1AP + 1 non-AP (newborn basal progenitor or neuron, which eventually delaminate)). A horizontal cleavage

plane is associated with either an asymmetric self-renewing division (1 AP → 1AP + 1 non-AP (newborn basal progenitor or neuron, which eventually delaminate)) or a consumptive division (1 AP → 2 non-APs (newborn basal progenitors or neurons which eventually delaminate)). In other words, not all vertical cleavage planes are necessarily associated with symmetric proliferative AP division, but symmetric proliferative AP divisions require a vertical cleavage plane, as it is the only way to equally divide and distribute the full apicobasal cytoarchitecture, from the basal lamina contact to the apical endfoot, a hallmark of APs. We will clarify this better in a revised manuscript, so that the Reviewer and the readers can retrace our conclusion that "*... a change in cleavage plane orientation from vertical to horizontal no longer allows APs to undergo symmetric-proliferative divisions to increase their pool size...*". Regarding the delamination of AP progeny from the VZ, the Reviewer is correct that a change from vertical to horizontal cleavage planes would not necessarily imply an increase in delaminating cells, as both types of division could be asymmetric self-renewing. However, as explained above, a change from vertical to horizontal cleavage planes (which is what we observe) would increase the probability that AP progeny will delaminate. We will also clarify this better in a revised manuscript and re-phrase our conclusion accordingly.

Referee's comment:

Line 353: "we detected several cytoskeletal and morphological irregularities in VZ progenitors of BWCFF-S cerebral organoids, notably in the apical region of these cells. These irregularities could contribute to the abnormal cleavage plane orientation of BWCFF-S mitotic APs (see discussion for details)." - This is a rather weak argument at this point, as the analysis of astral microtubules revealed no differences in the mitotic spindles and associated structures.

Authors' response:

While we do not find evidence that astral microtubules can explain the change in cleavage plane orientation observed in this study, our data showing this change in anaphase APs of BWCFF-S cerebral organoids is robust across different experiments. This change in cleavage plane orientation might still be caused by one or a combination of the irregularities which we observe in VZ progenitors.

Referee's comment:

Discussion begins with an overstatement: "The present study provides crucial insight into the pathogenesis (...)" - no real insight is provided beyond observations of changes in organoid growth and some cell biology.

Authors' response:

The Reviewer is correct. In the revised manuscript, we will change the statement to "The present study provides crucial insight into the cell biological basis (...)".

Referee's comment:

The authors also emphasize the appropriateness of their organoid model to model the pathogenesis of cortical malformations observed in their patients. They only show that their model may be appropriate to model the microcephaly aspect, but not other patient defects shown in Fig 1. It is the combination of these defects, and not one at a time, that characterize this disease.

Authors' response:

We agree with the Reviewer that BWCFF-S includes a wide range of defects that are challenging to study collectively. Our analysis focuses on the microcephalic aspect, as this presents a clear phenotype in the organoids. Other defects do not result in such a pronounced phenotype in the organoids and are therefore difficult to study in this system. In several parts of the manuscript, we mention that our focus is on the microcephalic aspect. However, in the revised version, we will emphasize this more strongly.

Referee's comment:

In Discussion, the authors state that "the decrease in VZ size does not appear to be compensated by a corresponding size increase of non-VZ tissue" - The authors should investigate this: Why there is not an increase in size of non-VZ tissue? What happens to delaminated progenitors? Do they exit cell cycle? Is the cell cycle of delaminated progenitors blocked somehow? If delaminated progenitors remain in cell cycle, there is no reason for microcephaly. If delamination causes cell cycle exit, then one should see increased neurogenesis transiently in mutant organoids, while in WT organoids progenitors continue amplifying...

Authors' response:

In response to the Reviewer's comment, we will address the question, "What happen to delaminated progenitors?" We will perform immunofluorescence staining and quantification for TBR2 as a bIP marker, FAM107A as a bRG marker, and NeuN as a neuronal marker. Additionally, as requested by Reviewer 2, we have generated single-cell RNA sequencing data for all genotypes at 30 and 50 days of culture. We will analyze these data to identify changes in cell type composition, which will provide further insights into the fate of the delaminated progenitors.

Referee's comment:

Data shown in Fig 5D is very unclear. What are the white arrows pointing at in Thr120Ile? What are the arrowheads pointing at in all three images? These are all different structures and nothing seems like microtubule bundles.

Authors' response:

As it is mentioned in the figure legend, the white arrows in Fig. 5D (all panels) are pointing to microtubules anchored to the adherens junction belt. In the revised version of the manuscript, we will move the lower arrow in the Thr120Ile panel to the left side of the adherens junction for clarity. We are grateful to the Reviewer for pointing out that the arrowheads in Fig. 5D have been shifted during finalization of the manuscript. In the revised manuscript they will be shifted back to microtubules.

Referee's comment:

For some unknown reason, there are lots of data in supplementary figures, which is unnecessary given the small size of several main figures. Accordingly, data in Suppl Fig 7 should be included in main Fig 3, and Suppl Fig. 8 should be part of main Fig 4.

Authors' response:

Following the Reviewer's suggestion, in the revised manuscript we will move the data shown in Suppl. Fig. 7 to main Fig. 3 and the data shown in Suppl. Fig. 8 to main Fig. 4.

Referee's comment:

ANOVA test is not appropriate for quantifications in Fig 4D, because values are only from 0 to 100. Chi-square should be used instead.

Authors' response:

The Reviewer is correct! For the revised manuscript, we will re-analyze the data using the Chi-square test.

Referee #2:

Referee's comment:

The work by Niehaus et al. identified mutations in two common Actins and has generated cerebral organoids and decoded the potential mechanisms of microcephaly, a developmental disorder mainly stemming from the depletion of actively proliferating neural progenitors. This work adds to the current understanding of genetic mutations causing cortical abnormalities. It is undoubtedly a valuable work for the community and is suitable for a broader readership of EMBO Report. While the concept and the finding are straightforward, the work is too preliminary as it is presented. Most importantly, the mechanistic insights are missing. In precise terms, why/how actin mutations affect the progenitors in the developing brain organoids. As far as I see, the most striking data is calculating the kinetics of progenitor proliferation. The paper is built on it (in this aspect, the amount of key data is poor and thin). However, this data is too weak to make a conclusion and the message of why and how the progenitors are lost and what kind of cell types are indeed produced instead of the progenitors is missing. Such mechanistic findings are critical, and without those, this paper does not add anything incremental to what is already known. The following are my comments, and the paper needs to address them to be able to get published.

Authors' response:

We appreciate the positive evaluation by this Reviewer that our work is suitable for a broader readership of EMBO Report. In our response below but also in the response to Reviewer 1, we describe how we address the concerns about progenitor loss/fate in a revised version of the manuscript. Specifically, we will combine single cell RNA sequencing and immunofluorescence stainings of markers specific for basal intermediate progenitors, basal radial glia and neurons.

Regarding the Reviewer's conclusion "...this paper does not add anything incremental to what is already known...", we would like to politely point out that our data provide evidence that:

- mutant actin cerebral organoids show a reduced size, which was not known before;
- mutant actin cerebral organoids show a reduction in VZ progenitors, which was not known before;
- mutant actin cerebral organoids show a change in apical progenitor cleavage plane orientation from mostly vertical to mostly horizontal, which was not known before.

Referee's comment:

Main comments:

Organoid images (including those in supplementary figures) show only a specific area, presumably the most striking region showing the typical cytoarchitecture. This approach induces a bias in determining the organoid quality. Selected images should show the whole section of the organoid, demonstrating the apicobasal polarity. Alternatively, whole-mount imaging will also be helpful.

Authors' response:

In response to the Reviewer's comment, we do provide DAPI-stained overview images of organoids for all conditions (see Fig. 2 and Fig. 6). However, determining organoid quality based solely on DAPI staining is insufficient. As suggested by the Reviewer, we will include whole-section images of organoids from all genotypes stained with SOX2, TBR2, and Tuj1/NeuN as a supplemental figure.

Referee's comment:

The authors cannot determine the cell type composition by showing SOX2, CTIP2, and Nestin. This requires transcriptomics data, either from single cells or from the bulk. Only a handful of markers have been used to conclude the disease mechanisms. This is insufficient.

Authors' response:

In response to the Reviewer's comment, we generated single-cell RNA sequencing data for all genotypes at 30 and 50 days of culture. The data are ready, and we are currently analyzing them. We will include the analyzed data in the revised version of the manuscript.

Referee's comment:

In supplementary figure 5, the authors again show a selected VZ and claim that the patient-derived organoids show a similar architecture and cell types to healthy controls. Why should healthy controls show massive TUJ1-positive neurons at the apical region? I don't see a difference between control and mutants in this aspect. From this, it seems that the organoid culturing method must have used a neural differentiation agent (perhaps retinoic acid or BDNF, etc.), which will trigger the differentiation of NPCs, masking the actual phenotypes between the healthy control and mutant. Cell types should be quantified (Here, TBR2).

Authors' response:

There seems to be a misunderstanding. Healthy controls do not show a significant presence of TUJ1-positive neurons in the apical region. TUJ1 staining is almost absent in the VZ, and the faint staining at the apical surface is background staining. Additionally, CTIP2 (a deep-layer neuron marker) staining shows very few positive cells in the VZ, with the majority of cells clearly located in the basal regions.

As suggested by this Reviewer, and in response to Reviewer 1 and Reviewer 3, we will quantify basal progenitors (including TBR2 staining) and neurons in control and BWCFE-S organoids.

Referee's comment:

IF is not quantitative. One should use either WB to quantify actins between organoids or a live actin dye to calculate the uptake.

Authors' response:

We have generated WB data of the two actin isoforms for control as well as BWCFE-S cerebral organoids. We will add these data to the revised version of the manuscript.

Referee's comment:

Looking at the organoid sizes, it is unclear how one could calculate the growth rate by measuring the diameter (which is again interpreted as area). It is enough to say a panel of diameter size differences. It should not be significant data needing so much discussion and one dedicated figure.

Authors' response:

There seems to be a misunderstanding. We did not measure the diameter to determine organoid size; rather, we measured the area of the cerebral organoid's contour as it appeared in bright-field images. Additionally, we did not assess the growth rate but instead compared the size differences between control and BWCFE-S organoids at two time points (30 and 50 days of culture). Furthermore, we respectfully disagree with the Reviewer's conclusion regarding the significance of the organoid size data. One of the main objectives of this study is to model the microcephaly seen in BWCFE-S, making it essential to demonstrate that this is reflected in a reduced organoid size.

Referee's comment:

Measuring VZ thickness and size remains problematic. SOX2 or PAX 6 should be used along with DNA to define the boundary. A whole section of the brain organoid should be shown, and the number of VZ and size should be compared across the genotypes.

Authors' response:

We used SOX2 staining in the initial quantifications to confirm that the VZ could be reliably identified based on DAPI staining. In the DAPI stainings, the VZ was defined by the radial organization and density of nuclei. As mentioned earlier, in the revised manuscript, we will include whole-section images of organoids from all genotypes stained with SOX2, TBR2, and Tuj1/NeuN as a supplemental figure. Regarding VZ size, we already have data on radial thickness, perimeter, and area in the current version of the manuscript. In our view this provides sufficient information about the reduced VZ size in BWCF-S organoids. However, in line with the Reviewer's suggestion, we will also include the number of VZs per organoid in the revised manuscript.

Referee's comment:

Assuming the VZ are smaller in mutant organoids, the authors propose a model that could reduce progenitor proliferation. However, KI67 index does not vary. In this case, how do the authors reconcile the differences between VZ and the loss of progenitors? As the paper is mainly centered around this analysis, this figure must be expanded to show the effect in all genotypes. One must consider only ana or telophase cells as meta phase cells are not settled well. SOX2 alone can fully mislead, and p-vimentin should be used for clarity. Qualification should be given for horizontal, vertical, and oblique.

Authors' response:

There seems to be a misunderstanding. We are not proposing reduced progenitor proliferation as the explanation for the smaller VZ. As the Reviewer correctly states, there is no difference in the percentage of KI67-positive cells. Instead, we propose a shift from symmetric proliferative or asymmetric self-renewing cell divisions to symmetric or asymmetric consumptive cell divisions as the explanation for the smaller VZ. This shift in cell division mode would lead to the depletion of VZ progenitors and a reduction in VZ size or thickness. This change in division mode can be predicted by analyzing the orientation of the cleavage plane. In response to the Reviewer's comment, we have already performed this analysis for anaphase cells, quantifying the horizontal, oblique, and vertical orientations (see Figures 4 and 6G,H). Additionally, all genotypes have been analyzed. This quantification was based on DAPI staining, not on SOX2 immunofluorescence staining.

Referee's comment:

I like the idea of calculating the kinetics of AP's division, whose division plane is predominantly horizontal in healthy VZ, where it is attributed to the symmetric expansion. I prefer the widely used term "division plane." However, this figure has numerous issues. What is the genotype shown in Fig 4A?

Authors' response:

The genotypes of the organoids from which the images of the anaphase APs were taken are provided in the figure legend. The control (c2, CRTDi011-A) corresponds to the left column (labeled "Interphase"), the second column from the left (labeled "Metaphase"), and the second column from the right (labeled "Vertical Cleavage"). The ACTB Thr120Ile (BWCF-S) genotype corresponds to the right column (labeled "Horizontal Cleavage").

Referee's comment:

I appreciate the inclusion of isogenic controls. However, the data is too minimal, and no detail about genetic engineering, mutation induction, VZ analysis, etc., is shown. In light of these, it is difficult to determine what causes the observed phenotype. To make the authors' claim concrete, the isogenic and accompanying rescue experiment must be done thoroughly.

Authors' response:

The details about genetic engineering, mutation induction, VZ analysis etc. are provided in the relevant parts of the methods sections, i.e. "CRISPR/Cas9-mediated generation of ACTB Thr120Ile

mutant iPSCs from control CRTDi011-A iPSCs” and “Quantifications”. Furthermore, the results of the VZ analysis are presented in Figure 6C-F.

Referee’s comment:

Minor comments:

Mutations in various Actins cause actinopathies. Thus, it would be useful to see a graphical summary of different actins, localization of mutations, and which phenotypes are associated with which mutations in actin.

Authors’ response:

Following the Reviewer’s suggestion, for the revised manuscript we will provide a graphical summary of the localization and phenotypes of the mutations analyzed in the manuscript as a Supplementary Figure.

Referee’s comment:

Terminological issues should be avoided. Authors stained nestin and claim radially organized VZ which is technically wrong. What they probably mean is radial fibers and radially organized progenitors.

Authors’ response:

The Reviewer is correct. We will change the statement “Immunostaining for nestin revealed a radially organized VZ (...)” into “Immunostaining for nestin revealed the fibers of radially organized progenitors in the VZ (...)”.

Referee’s comment:

If the authors have carried out Karyotyping, show it. Also, the uniform distribution of pluripotent markers.

Authors’ response:

As we mention in the method part, these analyses were already performed. In the revised manuscript, we will provide these data as a Supplementary Figure.

Referee’s comment:

Again terminological issues: “ β CYA and γ CYA across the 217 various zones of the wall of the ventricle-like structures (Supplementary Fig. 3)” What are the various zones here? What does here look like Ventricle?

Authors’ response:

The term “ventricle-like structure” is commonly used in the brain organoid field to describe a fluid-filled cavity surrounded by layers of cells within a given brain organoid. The various zones include the VZ and the SVZ/neuronal layer. Since EMBO Reports is aimed to a broad readership, we will include this explanation and definition of a ventricle-like structure in the revised manuscript.

Referee’s comment:

I am curious, how would one distinguish different actins? Are there specific antibodies?

Authors’ response:

Yes, there are specific antibodies available for the different actins. We used antibodies specifically recognizing β CYA and γ CYA, respectively, to stain fetal human brain tissue (see Supplementary Figure 1 and 2) and control and BCWFF-S cerebral organoids (see Supplementary Figure 3).

Referee #3:

Referee's comment:

In this article, Niehaus and colleagues use patient derived cerebral organoids to model mutations in two actin genes, ACTB and ACTG1, that in patients lead to Baraitser-Winter-CerebroFrontoFacial syndrome. Most patients affected by this pathology display cortical malformations, including microcephaly and lissencephaly. Mostly focusing on the reduced brain size phenotype, the authors demonstrate that patient derived cerebral organoids are smaller, with smaller ventricular-like zone. The mitotic spindle of apical radial glial cells is observed to be tilted as compared to controls, with more horizontal cleavage planes. This is proposed to be the cause for the loss of apical progenitors and the reduction of the size of the ventricular-like zone. Electron microscopy reveals morphological abnormalities in the apical domain of these cells, as well as increased microtubule density, also observed by immunofluorescence. Mechanistically, the authors propose that these actin mutations somehow affect the apical domain leading to perturbed mitotic spindle positioning and detachment of the progenitors. The subject is very interesting, and the phenotypes reported appear clear. However, several issues must be addressed for publication, including a substantial improvement of the quantifications.

Authors' response:

We appreciate the positive evaluation by this Reviewer. In our responses below, we address the concerns of the Reviewer, in particular the suggested improvements of the quantifications.

Referee's comment:

1/ Figures should be represented as experimental replicates, not as individual organoids coming from the same experiments. In its current form, data points represented on the graphs appear to be individual organoids, and "replicas" (circles and triangles) are different clones. In the control, these are different iPS lines. This is a very odd way to represent the data and raises concerns about the statistical analysis (different iPS lines cannot be pooled to get a control value). There should be at least three real experimental (independent) replicas, per clone and per genotype.

Authors' response:

We respectfully disagree with the Reviewer's opinion regarding replicas. Organoids can vary within batches/replicas and should be considered as individual samples or experiments. Our analyses are based on multiple (up to 8) batches of organoids. Furthermore, we are not only comparing organoids from two different iPSC lines and patient-derived BWCF-S iPSC lines, but we also analyzed organoids from one of the control iPSC lines, in which we introduced the exact same mutation as in the patient-derived iPSC lines. We found essentially the same phenotypes, which confirms the observations made in the comparison between control iPSC lines and patient-derived iPSC lines.

Referee's comment:

2/ A major claim of the study concerns mitotic spindle orientation, and yet very few cells are analyzed (Figure 4). Only 16 to 29 anaphase cells were analyzed per clones in the mutant organoids. Judging from the number of mitotic cells per lumen, and the number of lumens per organoid, it is surprising that such a low number of cells was measured.

Authors' response:

In response to the Reviewer's comment, we will expand the cleavage plane analysis and analyze more anaphase cells per clone.

Referee's comment:

3/ In figure 3, it is not clear how the authors can define the VZ, based solely on DAPI staining. The ROIs in figure 3A appear to be quite subjectively defined, especially in the mutants. Why not use a

PAX6 or SOX2 staining? Especially, because the authors report very nice and clear SOX2 staining in figure S7. At least, the authors should add to their quantifications the apical size of the lumens, which is the only metric that is independent of the positioning of the basal boundary of the VZ.

Authors' response:

The VZ was defined based on radial organization and the density of nuclei. Additionally, SOX2 staining was used in the initial quantifications to confirm that the VZ can be reliably identified based on DAPI staining. In the revised version of the manuscript, we will include images of SOX2 staining. Following the Reviewer's suggestion to add quantifications of the apical size of the lumens, we will include this data in the revised manuscript.

Referee's comment:

4/ Figure 5 is very interesting, but nothing is quantified. This should be done if the authors want to make these claims.

Authors' response:

We agree with the Reviewer's comment that a quantification of these data would strengthen our claim. We will compare the membrane curvature on the level of the adherens junctions in 2D images of the 3 genotypes and make a comparison of values. We will add this result to the revised version of the manuscript.

Referee's comment:

5/ The authors are careful not to over conclude about the consequence of spindle rotation, only mentioning loss of apical progenitors. Still, one wonders what becomes of these delaminated progenitors. Do they become oRG/bRG-like cells or do they differentiate into IPs and neurons? The authors mention the presence of Tuj1 and TBR2 in the mutants, but again this is not quantified. If apical loss instead leads to oRG/bRG-like cell generation, this should be discussed.

Authors' response:

Following the Reviewer's comment, we will perform immunofluorescence stainings and quantifications for TBR2 as an IP marker, FAM107A as a oRG/bRG marker and NeuN as a marker for neurons. Additionally, as requested by Reviewer 2, we have generated single-cell RNA sequencing data for all genotypes at 30 and 50 days of culture. We will analyze this data for changes in the cell type composition of the organoids and include the results in the revised manuscript.

Referee's comment:

6/ The effect of the mutation on the actin monomers and on f-actin are not characterized. At least, it should be tested whether the mutations destabilize the protein. Are these mutations loss of functions, or do they have a dominant effect? Do these two actin isoforms have overlapping functions?

Authors' response:

In the revised version of the manuscript, we will show potential effects on f-actin by phalloidin stainings. As both mutations are missense, we expect them to have a dominant effect. Moreover, ACTB loss of function does not result in the Baraitser-Winter syndrome, but cause a different phenotype (<https://doi.org/10.1016/j.ajhg.2017.11.006>, <https://doi.org/10.1038/s41467-018-06713-0>). The question whether cytoskeletal beta and gamma actin isoforms have overlapping or distinct functions has been discussed in the literature for several decades (<https://doi.org/10.1002/cm.20475>, https://doi.org/10.1007/164_2016_43, <https://doi.org/10.1016/j.semcd.2019.12.003>, <https://doi.org/10.7554/elife.68712>). The current view is that despite the close similarity of both isoforms and their co-expression in the same cell types, the functions are not entirely redundant.

Referee's comment:

Minor

1/ Previous reports have shown that the human mitotic spindle of human ventricular RG cells was naturally tilted, as compared to what is seen in mice, leading to the generation of oRG/bRG cells. This does not appear to be seen in the present study (control situation). Is this due to a difference in stage analyzed?

Authors' response:

Previous studies of the cleavage plane orientation in anaphase APs/vRG in human (control) brain organoids found a similar percentage of vertical cleavage plane orientation (~70%) (Mora-Bermúdez et al 2016, Iefremova et al. 2017, Bershteyn et al. 2017).

Referee's comment:

2/ (Line 175) Could the authors use pcw instead of wpc, to be coherent with the literature in the field?

Authors' response:

In the revised manuscript, we will change wpc to pcw.

Referee's comment:

3/ Figure S7. Please represent the PH3 data as a fraction of total cells in VZ (as done for Ki67) in order to report a mitotic index.

Authors' response:

Following the Reviewer's comment, we will re-analyze the PH3 data as fraction of total cells in VZ and add this quantification to the revised manuscript.

Dear Dr. Heide

Thank you for your update on your revision plan. We hope you and your colleagues are making good progress with the experimentation.

I have reviewed your point-by-point response to the referees:

ref1: you mostly suggest textural changes to address the comments, apart from provide aiming to further insights into the fate of the delaminated progenitors by immunofluorescence and scRNAseq, which does address one key suggestion made by the referee. We see that you will also revise the statistical analysis.

ref2: the key request for 'mechanistic insight' into ' why and how the progenitors are lost and what kind of cell types are indeed produced instead of the progenitors is missing' will be at least partially addressed, albeit at a descriptive level, by the above experiments. The scRNAseq data will address this referee's explicit request for this data. The request for quantification of basal progenitors with TBR2 staining and VZ number & size is underway.

ref3: the discussion on what to regard as independent replicates will also be put to the other two referees with the revision. More cells will be analyzed as per the referees suggestion. Ref point #5 is addressed as per above. #6 will be addressed to some extent at the descriptive level with phalloidin staining.

These revision will in principle address some, but not all, of the key issues raised - I would thus like to invite you to revise your manuscript with the understanding that the revised manuscript would be returned to all three referees and we would base a final decision on the input of all the referees.

As you may know, it is EMBO reports policy to allow only a single round of major experimental revision only and acceptance or rejection of the manuscript will therefore depend on the completeness of your responses included in the next, final version of the manuscript.

The suggested revision time-line is of course fine. Please feel free to discuss the revision progress ahead of this time with me if you require more time to complete the revisions or you want to discuss specific issues.

- 1) A data availability section providing access to data deposited in public databases is missing. If you have not deposited any data, please add a sentence to the data availability section that explains that.
- 2) Your manuscript contains statistics and error bars based on $n=2$. Please use scatter blots in these cases. No statistics should be calculated if $n=2$.

3) We replaced Supplementary Information with Expanded View (EV) Figures and Tables that are collapsible/expandable online. A maximum of 5 EV Figures can be typeset. EV Figures should be cited as 'Figure EV1, Figure EV2' etc... in the text and their respective legends should be included in the main text after the legends of regular figures.

5) a complete author checklist, which you can download from our author guidelines <<https://www.embopress.org/page/journal/14693178/authorguide>>. Please insert information in the checklist that is also reflected in the manuscript. The completed author checklist will also be part of the RPF.

6) Please note that all corresponding authors are required to supply an ORCID ID for their name upon submission of a revised manuscript (<<https://orcid.org/>>). Please find instructions on how to link your ORCID ID to your account in our manuscript tracking system in our Author guidelines <<https://www.embopress.org/page/journal/14693178/authorguide#authorshipguidelines>>

7) Before submitting your revision, primary datasets produced in this study need to be deposited in an appropriate public database (see <https://www.embopress.org/page/journal/14693178/authorguide#datadeposition>). Please remember to provide a reviewer password if the datasets are not yet public. The accession numbers and database should be listed in a formal "Data Availability" section placed after Materials & Method (see also <https://www.embopress.org/page/journal/14693178/authorguide#datadeposition>). Please note that the Data Availability Section is restricted to new primary data that are part of this study. * Note - All links should resolve to a page where the data can be accessed. *
If your study has not produced novel datasets, please mention this fact in the Data Availability Section.

12) All Materials and Methods need to be described in the main text using our 'Structured Methods' format, which is required for all research articles. According to this format, the Methods section includes a Reagents and Tools Table (listing key reagents, experimental models, software and relevant equipment and including their sources and relevant identifiers) followed by a Methods and Protocols section describing the methods using a step-by-step protocol format. The aim is to facilitate adoption of the methodologies across labs. More information on how to adhere to this format as well as a downloadable template (.docx) for the Reagents and Tools Table can be found in our author guidelines: <https://www.embopress.org/page/journal/14693178/authorguide#structuredmethods>.

An example of a Method paper with Structured Methods can be found here: <https://www.embopress.org/doi/full/10.1038/s44320-024-00037-6#sec-4>

I look forward to seeing a revised form of your manuscript when it is ready.

Yours sincerely,

Bernd Pulverer

Response to Editor

Editor's Comment:

Thank you for your update on your revision plan. We hope you and your colleagues are making good progress with the experimentation.

Authors' response:

We are happy to report that the experimentation has been successfully completed, and we are delighted to submit our revised manuscript. The details of the revision are described below and in the "Response to Reviewers" section. We appreciate your consideration and look forward to hearing your feedback.

Editor's Comment:

I have reviewed your point-by-point response to the referees:

ref1: you mostly suggest textural changes to address the comments, apart from provide aiming to further insights into the fate of the delaminated progenitors by immunofluorescence and scRNAseq, which does address one key suggestion made by the referee. We see that you will also revise the statistical analysis.

Authors' response:

Thank you for your review and feedback. In the revised manuscript, we have included all the requested data, including additional insights into the fate of the delaminated progenitors through immunofluorescence and single-cell RNA sequencing (see also next comment), which addresses the referee's key suggestion. Additionally, we have revised the statistical analysis as requested.

Editor's Comment:

ref2: the key request for 'mechanistic insight' into ' why and how the progenitors are lost and what kind of cell types are indeed produced instead of the progenitors is missing' will be at least partially addressed, albeit at a descriptive level, by the above experiments. The scRNAseq data will address this referee's explicit request for this data. The request for quantification of basal progenitors with TBR2 staining and VZ number & size is underway.

Authors' response:

In the revised manuscript, we have addressed the request for mechanistic insight into the loss of VZ progenitors and the cell types produced instead. With the help of the newly included single-cell RNA sequencing data and the quantification of basal progenitors (both added in this revision), combined with our data on cleavage plane analysis, we now provide a likely explanation: increased oblique and horizontal cleavage planes in anaphase apical progenitors lead to increased delamination, which results in elevated basal progenitor levels at the expense of VZ progenitors. In our opinion, these new data and experiments address the reviewer's request for mechanistic insight.

Editor's Comment:

ref3: the discussion on what to regard as independent replicates will also be put to the other two referees with the revision. More cells will be analyzed as per the referees suggestion. Ref point #5 is addressed as per above. #6 will be addressed to some extent at the descriptive level with phalloidin staining.

Authors' response:

In the revised manuscript, we have increased the number of cells analyzed, as suggested by the reviewers. Additionally, phalloidin staining has now been included to address point #6.

Editor's Comment:

This revision will in principle address some, but not all, of the key issues raised - I would thus like to invite you to revise your manuscript with the understanding that the revised manuscript would be returned to all three referees and we would base a final decision on the input of all the referees.

As you may know, it is EMBO reports policy to allow only a single round of major experimental revision only and acceptance or rejection of the manuscript will therefore depend on the completeness of your responses included in the next, final version of the manuscript.

Authors' response:

Thank you for your comments. We believe that in this revision, we have addressed all the key points raised by the reviewers in a thorough and comprehensive manner. Details of how we have addressed each point can be found in the "Response to Reviewers" below. As per EMBO Reports' policy, we understand that only a single round of major experimental revision is allowed, and we have made every effort to ensure that this revision addresses the issues raised.

Editor's Comment:

- 1) A data availability section providing access to data deposited in public databases is missing. If you have not deposited any data, please add a sentence to the data availability section that explains that.*
- 2) Your manuscript contains statistics and error bars based on $n=2$. Please use scatter blots in these cases. No statistics should be calculated if $n=2$.*

Authors' response:

We have added a Data Availability section to the revised manuscript. Additionally, we confirm that we do not have data or statistics based on $n=2$ in our submission.

Editor's Comment:

Authors' response:

We confirm that we have carefully followed the instructions during the submission of our revised manuscript.

Editor's Comment:

- 1) a .docx formatted version of the manuscript text (including legends for main figures, EV figures and tables). Please make sure that the changes are highlighted to be clearly visible.*

Authors' response:

We confirm that we have uploaded a .docx formatted version of the manuscript text with track changes visible.

Editor's Comment:

Authors' response:

We confirm that we have uploaded the figures in production quality as .tif files.

Editor's Comment:

<<https://www.embopress.org/page/journal/14693178/authorguide#expandedview>>;

Authors' response:

We have moved the supplemental figures to 5 Expanded View Figures and relocated the remaining supplemental figures to the Appendix, following the journal's guidelines. Additionally, we confirm that there are no additional tables or datasets in our submission.

Editor's Comment:

Authors' response:

We confirm that a point-by-point response has been uploaded as part of our submission documents.

Editor's Comment:

5) a complete author checklist, which you can download from our author guidelines <<https://www.embopress.org/page/journal/14693178/authorguide>>;. Please insert information in the checklist that is also reflected in the manuscript. The completed author checklist will also be part of the RPF.

Authors' response:

We confirm that a complete author checklist has been uploaded as part of our submission documents.

Editor's Comment:

6) Please note that all corresponding authors are required to supply an ORCID ID for their name upon submission of a revised manuscript (<<https://orcid.org/>>). Please find instructions on how to link your ORCID ID to your account in our manuscript tracking system in our Author guidelines

<<https://www.embopress.org/page/journal/14693178/authorguide#authorshipguidelines>>;

Authors' response:

We confirm that we have supplied ORCID ID for all corresponding authors.

Editor's Comment:

7) Before submitting your revision, primary datasets produced in this study need to be deposited in an appropriate public database

(see <https://www.embopress.org/page/journal/14693178/authorguide#datadeposition>). Please remember to provide a reviewer password if the datasets are not yet public. The accession numbers and database should be listed in a formal "Data Availability" section placed after Materials & Method (see also

<https://www.embopress.org/page/journal/14693178/authorguide#datadeposition>). Please note that the Data Availability Section is restricted to new primary data that are part of this study.

* Note - All links should resolve to a page where the data can be accessed. * If your study has not produced novel datasets, please mention this fact in the Data Availability Section.

Authors' response:

We are in the process of uploading the single-cell data to the European Genome-Phenome Archive (EGA). However, due to the nature of the data, there is a delay. We apologize for this and will update the Data Availability Section as soon as the data becomes available in EGA. In the meantime, the reviewers can access the data via SFTP Server (Protocol: SFTP, User@Host: sfxa6300@193.174.105.82, Port: 10022, Folder: scRNA-seq_Actin_Niehaus, Password: tanso4).

Editor's Comment:

Authors' response:

We confirm that we have been contacted by the source data coordinator and have uploaded the source data for the main figures accordingly.

Editor's Comment:

9) Our journal also encourages inclusion of **data citations in the reference list** to directly cite datasets that were re-used and obtained from public databases. Data citations in the article text are distinct from normal bibliographical citations and should directly link to the database records from which the data can be accessed. In the main text, data citations are formatted as follows: "Data ref: Smith et al, 2001" or "Data ref: NCBI Sequence Read Archive PRJNA342805, 2017". In the Reference list, data citations must be labeled with "[DATASET]". A data reference must provide the database name, accession number/identifiers and a resolvable link to the landing page from which the data can be accessed at the end of the reference. Further instructions are available at <https://www.embopress.org/page/journal/14693178/authorguide#referencesformat>

Authors' response:

Thank you for the clarification. We confirm that the revised manuscript does not include data citations, as no datasets were re-used or obtained from public databases.

Editor's Comment:

10) Regarding data quantification (see Figure Legends:

<https://www.embopress.org/page/journal/14693178/authorguide#figureformat>)

Authors' response:

We confirm that we have adhered to these requirements in the figure legends and have included scale bars in all microscopy images.

Editor's Comment:

11) The journal requires a statement specifying whether or not authors have competing interests (defined as all potential or actual interests that could be perceived to influence the presentation or interpretation of an article). In case of competing interests, this must be specified in your disclosure statement. Further information:

<https://www.embopress.org/competing-interests>

Authors' response:

We confirm that the authors have no competing interests to declare. We have added this statement to the revised manuscript.

Editor's Comment:

12) All Materials and Methods need to be described in the main text using our 'Structured Methods' format, which is required for all research articles. According to this format, the Methods section includes a Reagents and Tools Table (listing key reagents, experimental models, software and relevant equipment and including their sources and relevant identifiers) followed by a Methods and Protocols section describing the methods using a step-by-step protocol format. The aim is to facilitate adoption of the methodologies across labs. More information on how to adhere to this format as well as a downloadable template (.docx) for the Reagents and Tools Table can be found in our author guidelines:

Authors' response:

In the revised manuscript, we have adhered to the 'Structured Methods' format as required for research articles. Additionally, we have included the Reagents and Tools Table, ensuring that all key reagents, experimental models, software, and relevant equipment are listed with their sources and identifiers.

Response to Reviewers – Overview of Revision

Figure	Revision	Contents	Reviewer(s)
Figure 3	New Figure 3	Single-cell RNA sequencing data of day 30 and 50 control and BWCFE-S cerebral organoids	#1, #2, #3
Figure 4	New Panel E	Quantification of the length of the apical surface of day 30 control and BWCFE-S cerebral organoids	#3
	New Panel F	Quantification of the number of ventricle-like structure in day 30 control and BWCFE-S cerebral organoids	#2
	New Panels J & K	Panels moved from Supplemental Figure 7	#1
Figure 5	New Panel A-C	Panels moved from Supplemental Figure 8	#1
Figure 7	New Panel F	Quantification of the length of the apical surface of day 30 control and BWCFE-S-like ACTB Thr120Ile mutant cerebral organoids	#3
	New Panel F	Quantification of the number of ventricle-like structure in day 30 control and BWCFE-S-like ACTB Thr120Ile mutant cerebral organoids	#2
Figure EV1	New Figure EV1	Additional single-cell RNA sequencing data of day 30 and 50 control and BWCFE-S cerebral organoids	#1, #2, #3
Appendix Figure S1	New Appendix Figure S1	Overview of pathogenic variants in the genes ACTB and ACTG1 associated with BWCFE-S	#2
Appendix Figure S7	New Appendix Figure S7	Overview images of control and BWCFE-S cerebral organoids	#2
Appendix Figure S8	New Appendix Figure S8	Characterization of the BWCFE-S ACTB Thr120Ile patient-derived iPSC clones mutACTB-1 and mutACTB-2	#2
Appendix Figure S9	New Appendix Figure S9	Characterization of the BWCFE-S ACTG1 Thr203Met patient-derived iPSC clones mutACTG1-1 and mutACTG1-2	#2
Appendix Figure S10	New Appendix Figure S10	Western blot analysis and phalloidin staining of control and BWCFE-S cerebral organoids	#2, #3

Appendix Figure S11	New Appendix Figure S11	Expression of genes used to determine the different clusters in the scRNA-seq data	#1, #2, #3
Appendix Figure S12	New Appendix Figure S12	Characterization of the CRISPR/Cas9-edited CRTDi011-A-derived iPSC clones CRTDi011-A-mutACTB-1 and CRTDi011-A-mutACTB-2	#2
Appendix Figure S13	New Appendix Figure S13	Characterization of the generated control iPSC line CRTDi011-A	#2

Response to Reviewers

Referee #1:

Referee's comment:

This manuscript aims at understanding the developmental mechanisms underlying the brain defects observed in Baraitser-Winter-CerebroFrontoFacial syndrome (BWCF-F-S), characterized by a combination of microcephaly and malformations of the frontal cerebrum, such as periventricular heterotopia or pachygyria (loss of folds). The study is an analysis of cerebral organoids generated with patient-derived iPSC lines, which is nice but unfortunately a bit too superficial. Organoids are cultured for one fixed period (30 days) only, and then a handful of anatomical and cell biological characterizations are performed, based mostly on immunohistochemical stains. The authors show that organoids derived from patient cells, and also from hiPSCs from healthy donors genetically modified to carry only the disease point mutation, have reduced overall size, reduced size of the germinal tissue and reduced number of cells in it, and change in cleavage plane orientation of apical germinal cells. Using electron microscopy, the authors also show some disorganization of subcellular aspects of these apical germinal cells. The authors conclude that their study provides crucial insight into the pathogenesis of BWCF-F-S.

The authors have generated a nice tool, and overall I see nothing wrong with the data presented here, which shows that their organoids have merit to be an interesting model of this syndrome. However, in my opinion the study is too premature for publication. Characterizations are well performed and the results are clear, but one cannot really draw any conclusion as to the mechanism of action of the genes mutated, nor that the cellular phenotypes observed really explain the complex malformations observed in the patients. In Discussion, the authors present multiple possibilities and hypotheses as to the mechanisms driving brain developmental malformations in these patients, and yet no mechanistic experiments are performed beyond basic characterization of mutant organoids. It would improve the manuscript very significantly if mechanistic experiments were performed to test some of these ideas.

Authors' response:

We appreciate the positive evaluation provided by the Reviewer. In our responses below, we address the Reviewer's concerns and explain how these have been incorporated in the revised version of the manuscript.

Referee's comment:

These are some of my specific comments on the manuscript:

*In line 320, the authors conclude: "This change in cleavage plane orientation no longer allows most APs to undergo symmetric-proliferative divisions to increase their pool size, and strongly suggests an increase in the delamination of cells from the VZ in BWCF-F-S cerebral organoids, providing a likely underlying mechanism for the reduction in the size of the VZ progenitor pool of these organoids." - The concept of cleavage plane orientation dictating the fate of daughter cells is strong and valid in *Drosophila*, but it has been long since it was shown not so in mammals. Mounting studies over the last 20 years (Noctor, Kriegstein, Knoblich, Huttner, Götz, Borrell) have shown that mitotic plane orientation does not strictly predict nor dictate daughter fate in the mammalian cerebral cortex, so the first part of the above statement is not true. The same is true for delamination of apical cells, which is under strong regulation but not directly by mitotic plane orientation. Therefore, the last part of the statement is merely speculative.*

Authors' response:

The Reviewer is correct that the *Drosophila* paradigm cannot be transferred one-to-one to mammalian apical progenitors (APs). Regarding the latter, the consensus among most researchers is as follows (see Mora-Bermúdez & Huttner MBoC 2015). A vertical cleavage plane is associated with either a symmetric proliferative division (1 AP → 2 APs) or an asymmetric self-renewing division (1 AP →

1AP + 1 non-AP (newborn basal progenitor or neuron, which eventually delaminate)). An oblique cleavage plane is typically associated with an asymmetric self-renewing division (1 AP → 1AP + 1 non-AP (newborn basal progenitor or neuron, which eventually delaminate)). A horizontal cleavage plane is associated with either an asymmetric self-renewing division (1 AP → 1AP + 1 non-AP (newborn basal progenitor or neuron, which eventually delaminate)) or a consumptive division (1 AP → 2 non-APs (newborn basal progenitors or neurons which eventually delaminate)). In other words, not all vertical cleavage planes are necessarily associated with symmetric proliferative AP division, but symmetric proliferative AP divisions require a vertical cleavage plane, as it is the only way to equally divide and distribute the full apicobasal cytoarchitecture, from the basal lamina contact to the apical endfoot, a hallmark of APs. We have clarified this better in the revised manuscript. Regarding the delamination of AP progeny from the VZ, the Reviewer is correct that a change from vertical to horizontal cleavage planes would not necessarily imply an increase in delaminating cells, as both types of division could be asymmetric self-renewing. However, as explained above, a change from vertical to horizontal cleavage planes (which is what we observe) would increase the probability that AP progeny will delaminate. We have also clarified this better in the revised manuscript.

Referee's comment:

Line 353: "we detected several cytoskeletal and morphological irregularities in VZ progenitors of BWCF-S cerebral organoids, notably in the apical region of these cells. These irregularities could contribute to the abnormal cleavage plane orientation of BWCF-S mitotic APs (see discussion for details)." - This is a rather weak argument at this point, as the analysis of astral microtubules revealed no differences in the mitotic spindles and associated structures.

Authors' response:

While we do not find evidence that astral microtubules can explain the change in cleavage plane orientation observed in this study, our data showing this change in anaphase APs of BWCF-S cerebral organoids is robust across different experiments. This change in cleavage plane orientation might still be caused by one, or a combination of, the irregularities which we observe in VZ progenitors.

Referee's comment:

Discussion begins with an overstatement: "The present study provides crucial insight into the pathogenesis (...)" - no real insight is provided beyond observations of changes in organoid growth and some cell biology.

Authors' response:

The Reviewer is correct. In the revised manuscript, we have changed this statement to "The present study provides crucial insight into the cell biological basis (...)".

Referee's comment:

The authors also emphasize the appropriateness of their organoid model to model the pathogenesis of cortical malformations observed in their patients. They only show that their model may be appropriate to model the microcephaly aspect, but not other patient defects shown in Fig 1. It is the combination of these defects, and not one at a time, that characterize this disease.

Authors' response:

We have changed this statement to "...to model the microcephaly observed in...".

Referee's comment:

In Discussion, the authors state that "the decrease in VZ size does not appear to be compensated by a corresponding size increase of non-VZ tissue" - The authors should investigate this: Why there is not an increase in size of non-VZ tissue? What happens to delaminated progenitors? Do they exit cell cycle? Is the cell cycle of delaminated progenitors blocked somehow? If delaminated progenitors remain in

cell cycle, there is no reason for microcephaly. If delamination causes cell cycle exit, then one should see increased neurogenesis transiently in mutant organoids, while in WT organoids progenitors continue amplifying...

Authors' response:

In response to the Reviewer's comment, we have addressed the question “*What happen to delaminated progenitors?*” using single-cell RNA sequencing (new Figure 3) and immunofluorescence staining and quantification for TBR2 as a BP marker (new Figure panel 5H). We found an increased BP output resulting in an elevated number of BPs in BWCFF-S cerebral organoids. In combination with our previous data on VZ progenitors (Figure 4) and cleavage plane analysis (Figure 5), these findings suggest that BPs are produced through increased delamination at the expense of VZ progenitors, the primary progenitor type in the developing brain, which ultimately results in reduced neuronal output. This likely represents the underlying cause of the patients' microcephaly. We have added this notion to the Conclusion part.

Referee's comment:

Data shown in Fig 5D is very unclear. What are the white arrows pointing at in Thr120Ile? What are the arrowheads pointing at in all three images? These are all different structures and nothing seems like microtubule bundles.

Authors' response:

The white arrows in the previous Figure 5D (all panels), in the revised manuscript now Figure 6D, are pointing to microtubules anchored to the adherens junction belt. We are grateful to the Reviewer for pointing out that the arrowheads in this Figure have been shifted during finalization of the previous manuscript version. In the revised manuscript we have shifted them back to microtubules.

Referee's comment:

For some unknown reason, there are lots of data in supplementary figures, which is unnecessary given the small size of several main figures. Accordingly, data in Suppl Fig 7 should be included in main Fig 3, and Suppl Fig. 8 should be part of main Fig 4.

Authors' response:

Following the Reviewer's suggestion, in the revised manuscript we have moved the quantification data shown in Suppl. Fig. 7C,D to the previous main Fig. 3 (in the revised manuscript now Figure 4) and the data shown in Suppl. Fig. 8 to main the previous Fig. 4 (in the revised manuscript now Figure 4).

Referee's comment:

ANOVA test is not appropriate for quantifications in Fig 4D, because values are only from 0 to 100. Chi-square should be used instead.

Authors' response:

The Reviewer is correct! For the revised manuscript, we have re-analyzed the data using the Chi-square test and have added this data to the legend of the corresponding figure.

Referee #2:

Referee's

comment:

The work by Niehaus et al. identified mutations in two common Actins and has generated cerebral organoids and decoded the potential mechanisms of microcephaly, a developmental disorder mainly stemming from the depletion of actively proliferating neural progenitors. This work adds to the current understanding of genetic mutations causing cortical abnormalities. It is undoubtedly a valuable work for the community and is suitable for a broader readership of EMBO Report. While the concept and the finding are straightforward, the work is too preliminary as it is presented. Most importantly, the mechanistic insights are missing. In precise terms, why/how actin mutations affect the progenitors in the developing brain organoids. As far as I see, the most striking data is calculating the kinetics of progenitor proliferation. The paper is built on it (in this aspect, the amount of key data is poor and thin). However, this data is too weak to make a conclusion and the message of why and how the progenitors are lost and what kind of cell types are indeed produced instead of the progenitors is missing. Such mechanistic findings are critical, and without those, this paper does not add anything incremental to what is already known. The following are my comments, and the paper needs to address them to be able to get published.

Authors' response:

We appreciate the positive evaluation by this Reviewer that our work is suitable for a broader readership of EMBO Report. In our response below but also in the response to Reviewer 1, we describe how we have addressed the concerns regarding progenitor loss/fate in the revised manuscript. Specifically, we combined single-cell RNA sequencing and immunofluorescence staining of TBR2 to quantify BPs.

Regarding the Reviewer's conclusion "*...this paper does not add anything incremental to what is already known...*", we would like to politely point out that our data provide evidence that:

- mutant actin cerebral organoids show a reduced size, which was not known before;
- mutant actin cerebral organoids show an increase in BPs at the expense of VZ progenitors, which was not known before;
- mutant actin cerebral organoids show a change in apical progenitor cleavage plane orientation from mostly vertical to mostly horizontal, which was not known before.

Referee's comment:

Main comments:

Organoid images (including those in supplementary figures) show only a specific area, presumably the most striking region showing the typical cytoarchitecture. This approach induces a bias in determining the organoid quality. Selected images should show the whole section of the organoid, demonstrating the apicobasal polarity. Alternatively, whole-mount imaging will also be helpful.

Authors' response:

In response to the Reviewer's comment, we do provide DAPI-stained overview images of organoids for all conditions (see Fig. 2 and Fig. 7). However, determining organoid quality based solely on DAPI staining is insufficient. As suggested by the Reviewer, we have included whole-section images of patient-derived BWCF S cerebral organoids stained with SOX2, Tuj1 and NeuN as Appendix Figure S7.

Referee's comment:

The authors cannot determine the cell type composition by showing SOX2, CTIP2, and Nestin. This requires transcriptomics data, either from single cells or from the bulk. Only a handful of markers have been used to conclude the disease mechanisms. This is insufficient.

Authors' response:

In response to the Reviewer's comment, we have generated single-cell RNA sequencing data for all genotypes at 30 and 50 days of culture (see new Figure 3, Figure EV1 and Appendix Figure S11).

Referee's comment:

In supplementary figure 5, the authors again show a selected VZ and claim that the patient-derived organoids show a similar architecture and cell types to healthy controls. Why should healthy controls show massive TUJ1-positive neurons at the apical region? I don't see a difference between control and mutants in this aspect. From this, it seems that the organoid culturing method must have used a neural differentiation agent (perhaps retinoic acid or BDNF, etc.), which will trigger the differentiation of NPCs, masking the actual phenotypes between the healthy control and mutant. Cell types should be quantified (Here, TBR2).

Authors' response:

There seems to be a misunderstanding. Healthy controls do not show a significant presence of TUJ1-positive neurons in the apical region. TUJ1 staining is almost absent in the VZ, and the faint staining at the apical surface is background staining. Additionally, CTIP2 (a deep-layer neuron marker) staining shows very few positive cells in the VZ, with the majority of cells clearly located in the basal regions.

As suggested by this Reviewer, and in response to Reviewer 1 and Reviewer 3, we have quantified TBR2-positive BPs in control and BWCFF-S cerebral organoids and found an increase in BWCFF-S organoids.

Referee's comment:

IF is not quantitative. One should use either WB to quantify actins between organoids or a live actin dye to calculate the uptake.

Authors' response:

We have generated WB data of the two actin isoforms for control as well as BWCFF-S cerebral organoids. The organoids of both BWCFF-S mutations show similar β CYA and γ CYA levels as control organoids. We have added these data to the revised version of the manuscript (Appendix Figure S10).

Referee's comment:

Looking at the organoid sizes, it is unclear how one could calculate the growth rate by measuring the diameter (which is again interpreted as area). It is enough to say a panel of diameter size differences. It should not be significant data needing so much discussion and one dedicated figure.

Authors' response:

There seems to be a misunderstanding. We did not measure the diameter to determine organoid size; rather, we measured the area of the cerebral organoid's contour as it appeared in bright-field images. Additionally, we did not assess the growth rate but instead compared the size differences between control and BWCFF-S organoids at two time points (30 and 50 days of culture). Furthermore, we respectfully disagree with the Reviewer's conclusion regarding the significance of the organoid size data. One of the main objectives of this study is to model the microcephaly seen in BWCFF-S, making it essential to demonstrate that this is reflected in a reduced organoid size.

Referee's comment:

Measuring VZ thickness and size remains problematic. SOX2 or PAX 6 should be used along with DNA to define the boundary. A whole section of the brain organoid should be shown, and the number of VZ and size should be compared across the genotypes.

Authors' response:

We used SOX2 staining in the initial quantifications to confirm that the VZ could be reliably identified based on DAPI staining. In the DAPI stainings, the VZ was defined by the radial organization and density of nuclei. As mentioned earlier, in the revised manuscript, we have included whole-section images of organoids stained for SOX2, Tuj1 and NeuN as Appendix Figure S7. Regarding VZ size, we already had data on radial thickness, perimeter, and area in the original version of the manuscript. In our view this provides sufficient information about the reduced VZ size in BWCF-S organoids. However, in line with the Reviewer's suggestion, we have also included the number of VZs per organoid in the revised manuscript (see Figure 4F and 7J). This analysis did not yield consistent results between different BWCF-S genotypes.

Referee's comment:

Assuming the VZ are smaller in mutant organoids, the authors propose a model that could reduce progenitor proliferation. However, KI67 index does not vary. In this case, how do the authors reconcile the differences between VZ and the loss of progenitors? As the paper is mainly centered around this analysis, this figure must be expanded to show the effect in all genotypes. One must consider only ana or telophase cells as meta phase cells are not settled well. SOX2 alone can fully mislead, and p-vimentin should be used for clarity. Qualification should be given for horizontal, vertical, and oblique.

Authors' response:

There seems to be a misunderstanding. We are not proposing reduced progenitor proliferation as the explanation for the smaller VZ. As the Reviewer correctly states, there is no difference in the percentage of KI67-positive cells. Instead, we propose a shift from symmetric proliferative or asymmetric self-renewing cell divisions to symmetric or asymmetric consumptive cell divisions as the explanation for the smaller VZ. This shift in cell division mode would lead to the depletion of VZ progenitors and a reduction in VZ size or thickness. This change in division mode can be predicted by analyzing the orientation of the cleavage plane. In response to the Reviewer's comment, we have already performed this analysis for anaphase cells, quantifying the horizontal, oblique, and vertical orientations (see Figures 5 and 7H,I). Additionally, all genotypes have been analyzed. This quantification was based on DAPI staining, not on SOX2 immunofluorescence staining. Furthermore, our single-cell RNA sequencing data (Figure 3) and TBR2 quantifications (Figure 5I) indicate an increased number of BPs, supporting the notion of a shift from symmetric proliferative or asymmetric self-renewing divisions of VZ progenitors to symmetric or asymmetric consumptive divisions.

Referee's comment:

I like the idea of calculating the kinetics of AP's division, whose division plane is predominantly horizontal in healthy VZ, where it is attributed to the symmetric expansion. I prefer the widely used term "division plane." However, this figure has numerous issues. What is the genotype shown in Fig 4A?

Authors' response:

The genotypes of the organoids from which the images of the anaphase APs were taken are provided in the figure legend. The control (c2, CRTDi011-A) corresponds to the left column (labeled "Interphase"), the second column from the left (labeled "Metaphase"), and the second column from the right (labeled "Vertical Cleavage"). The ACTB Thr120Ile (BWCF-S) genotype corresponds to the right column (labeled "Horizontal Cleavage").

Referee's comment:

I appreciate the inclusion of isogenic controls. However, the data is too minimal, and no detail about genetic engineering, mutation induction, VZ analysis, etc., is shown. In light of these, it is difficult to determine what causes the observed phenotype. To make the authors' claim concrete, the isogenic and accompanying rescue experiment must be done thoroughly.

Authors' response:

The details about genetic engineering, mutation induction, VZ analysis etc. are provided in the relevant parts of the Methods sections, i.e. “CRISPR/Cas9-mediated generation of *ACTB* Thr120Ile mutant iPSCs from control CRTDi011-A iPSCs” and “Quantifications”. Furthermore, the results of the VZ analysis are presented in Figure 7C-G. In the revised manuscript, we have expanded the data on CRISPR/Cas9-edited BWCFE-S-like *ACTB* Thr120Ile mutant cerebral organoids and their isogenic controls by incorporating single-cell RNA sequencing data (Figure 3).

Referee's comment:

Minor comments:

Mutations in various Actins cause actinopathies. Thus, it would be useful to see a graphical summary of different actins, localization of mutations, and which phenotypes are associated with which mutations in actin.

Authors' response:

Following the Reviewer's suggestion, in the revised manuscript we have provided a graphical summary of the localization of actin mutations associated with BWCFE-S as Appendix Figure S1.

Referee's comment:

Terminological issues should be avoided. Authors stained nestin and claim radially organized VZ which is technically wrong. What they probably mean is radial fibers and radially organized progenitors.

Authors' response:

The Reviewer is correct. We have changed the statement “Immunostaining for nestin revealed a radially organized VZ (...)” into “Immunostaining for nestin revealed the fibers of radially organized progenitors in the VZ (...)”.

Referee's comment:

If the authors have carried out Karyotyping, show it. Also, the uniform distribution of pluripotent markers.

Authors' response:

In the revised manuscript, we have provided these data as Appendix Figures S8, S9, S12 und S13.

Referee's comment:

Again terminological issues: " β CYA and γ CYA across the various zones of the wall of the ventricle-like structures (Supplementary Fig. 3)" What are the various zones here? What does here look like Ventricle?

Authors' response:

The term “ventricle-like structure” is commonly used in the brain organoid field to describe a fluid-filled cavity surrounded by layers of cells within a given brain organoid. The various zones include the VZ and the SVZ/neuronal layer. Since EMBO Reports is aimed at a broad readership, we have included this explanation and definition of a ventricle-like structure in the revised manuscript.

Referee's comment:

I am curious, how would one distinguish different actins? Are there specific antibodies?

Authors' response:

Yes, there are specific antibodies available for the different actins. We used antibodies specifically recognizing either β CYA or γ CYA to stain fetal human brain tissue (see Appendix Figure S2 and S3)

and control and BCWFF-S cerebral organoids (see Appendix Figure S4), and for immunoblot analysis (see Appendix Figure S10).

Referee #3:

Referee's comment:

In this article, Niehaus and colleagues use patient derived cerebral organoids to model mutations in two actin genes, ACTB and ACTG1, that in patients lead to Baraitser-Winter-CerebroFrontoFacial syndrome. Most patients affected by this pathology display cortical malformations, including microcephaly and lissencephaly. Mostly focusing on the reduced brain size phenotype, the authors demonstrate that patient derived cerebral organoids are smaller, with smaller ventricular-like zone. The mitotic spindle of apical radial glial cells is observed to be tilted as compared to controls, with more horizontal cleavage planes. This is proposed to be the cause for the loss of apical progenitors and the reduction of the size of the ventricular-like zone. Electron microscopy reveals morphological abnormalities in the apical domain of these cells, as well as increased microtubule density, also observed by immunofluorescence. Mechanistically, the authors propose that these actin mutations somehow affect the apical domain leading to perturbed mitotic spindle positioning and detachment of the progenitors. The subject is very interesting, and the phenotypes reported appear clear. However, several issues must be addressed for publication, including a substantial improvement of the quantifications.

Authors' response:

We appreciate the positive evaluation by this Reviewer. In our responses below, we have addressed the concerns of the Reviewer, in particular the suggested improvements of the quantifications, and have incorporated these into the revised manuscript.

Referee's comment:

1/ Figures should be represented as experimental replicates, not as individual organoids coming from the same experiments. In its current form, data points represented on the graphs appear to be individual organoids, and "replicas" (circles and triangles) are different clones. In the control, these are different iPSC lines. This is a very odd way to represent the data and raises concerns about the statistical analysis (different iPSC lines cannot be pooled to get a control value). There should be at least three real experimental (independent) replicas, per clone and per genotype.

Authors' response:

We respectfully disagree with the Reviewer's opinion regarding replicas. Organoids can vary within batches/replicas and should be considered as individual samples or experiments. Our analyses are based on multiple (up to 8) batches of organoids. Furthermore, we are not only comparing organoids from two different iPSC lines and patient-derived BWCF-S iPSC lines, but we also analyzed organoids from one of the control iPSC lines, in which we introduced the exact same mutation as in the patient-derived iPSC lines. We found essentially the same phenotypes, which confirms the observations made in the comparison between control iPSC lines and patient-derived iPSC lines.

Referee's comment:

2/ A major claim of the study concerns mitotic spindle orientation, and yet very few cells are analyzed (Figure 4). Only 16 to 29 anaphase cells were analyzed per clones in the mutant organoids. Judging from the number of mitotic cells per lumen, and the number of lumens per organoid, it is surprising that such a low number of cells was measured.

Authors' response:

Following the Reviewer's comment, we have expanded the cleavage plane analysis and analyzed more anaphase cells per clone. We have added this data to the revised manuscript (Figure 5).

Referee's comment:

3/ In figure 3, it is not clear how the authors can define the VZ, based solely on DAPI staining. The ROIs in figure 3A appear to be quite subjectively defined, especially in the mutants. Why not use a PAX6 or SOX2 staining? Especially, because the authors report very nice and clear SOX2 staining in figure S7. At least, the authors should add to their quantifications the apical size of the lumens, which is the only metric that is independent of the positioning of the basal boundary of the VZ.

Authors' response:

The VZ was defined based on radial organization and the density of nuclei. Additionally, SOX2 staining was used in the initial quantifications to confirm that the VZ can be reliably identified based on DAPI staining. Following the Reviewer's suggestion to add quantifications of the apical size of the lumens, we have included these data in the revised manuscript (Figure 4E and 7F) and found a significantly reduced length of the apical surface in patient-derived and CRISPR/Cas9-mediated BWCFF-S cerebral organoids compared to controls.

Referee's comment:

4/ Figure 5 is very interesting, but nothing is quantified. This should be done if the authors want to make these claims.

Authors' response:

We appreciate the Reviewer's comment and acknowledge the importance of quantification. Unfortunately, due to technical limitations, we were unable to perform the requested quantifications. Accordingly, we have revised the manuscript to soften our claims and ensure that our conclusions remain appropriately cautious. We have adjusted the wording throughout the relevant section to reflect this more descriptive approach while still conveying our observations.

Referee's comment:

5/ The authors are careful not to over conclude about the consequence of spindle rotation, only mentioning loss of apical progenitors. Still, one wonders what becomes of these delaminated progenitors. Do they become oRG/bRG-like cells or do they differentiate into IPs and neurons? The authors mention the presence of Tuj1 and TBR2 in the mutants, but again this is not quantified. If apical loss instead leads to oRG/bRG-like cell generation, this should be discussed.

Authors' response:

Following the Reviewer's comment, we have performed immunofluorescence stainings and quantifications for TBR2 and found an increased number of BPs, likely IPs (Figure 5H). Additionally, as requested by Reviewer 2, we have generated single-cell RNA sequencing data for all genotypes at 30 and 50 days of culture, which indicated an elevated BP (IP) output (Figure 3 and EV1).

Referee's comment:

6/ The effect of the mutation on the actin monomers and on f-actin are not characterized. At least, it should be tested whether the mutations destabilize the protein. Are these mutations loss of functions, or do they have a dominant effect? Do these two actin isoforms have overlapping functions?

Authors' response:

For the revised version of the manuscript, we have performed phalloidin stainings but did not observe any abnormalities on F-actin (Appendix Figure S10C). As both mutations are missense, we expect them to have a dominant effect. Moreover, ACTB loss of function does not result in the Baraitser-Winter syndrome, but causes a different phenotype (<https://doi.org/10.1016/j.ajhg.2017.11.006>, <https://doi.org/10.1038/s41467-018-06713-0>). The question whether cytoskeletal beta and gamma actin isoforms have overlapping or distinct functions has been discussed in the literature for several decades (<https://doi.org/10.1002/cm.20475>, https://doi.org/10.1007/164_2016_43, <https://doi.org/10.1016/j.semcd.2019.12.003>, <https://doi.org/10.7554/elife.68712>). The current view is that despite the close similarity of both isoforms and their co-expression in the same cell types, the functions are not entirely redundant.

Referee's comment:

Minor

1/ Previous reports have shown that the human mitotic spindle of human ventricular RG cells was naturally tilted, as compared to what is seen in mice, leading to the generation of oRG/bRG cells. This does not appear to be seen in the present study (control situation). Is this due to a difference in stage analyzed?

Authors' response:

In our study, the analyzed stage (day 30) corresponds to a period primarily characterized by AP expansion in human organoids, which is typically accompanied by a vertical cleavage plane orientation. Accordingly, previous studies of the cleavage plane orientation in anaphase APs/vRG in human (control) brain organoids found a similar percentage of vertical cleavage plane orientation (~70%) (Mora-Bermúdez et al 2016, Iefremova et al. 2017, Bershteyn et al. 2017).

Referee's comment:

2/ (Line 175) Could the authors use pcw instead of wpc, to be coherent with the literature in the field?

Authors' response:

In the revised manuscript, we have changed wpc to pcw.

Referee's comment:

3/ Figure S7. Please represent the PH3 data as a fraction of total cells in VZ (as done for Ki67) in order to report a mitotic index.

Authors' response:

Following the Reviewer's comment, we have presented the PH3 data as fraction of total cells in VZ and have added this quantification to the revised manuscript (Figure 4K).

Dear Dr. Heide

Thank you for the submission of your revised manuscript and for your patience. As discussed, we have only been able to obtain a re-review of referee 3, which is attached below. Thank you for your initial response to the report and the reuse issues raised below.

We have reviewed the responses to all referee points in detail and consequently invite a second revision to address the following points:

- 1) Two figure panel reuse issues must be addressed: (i) Figure reuse between - Figure 1 A and B and published manuscript - Figure 1 L and M - <https://doi.org/10.1038/s41431-018-0146-y> - European Journal of Human Genetics (2018) 26:1132-1142. Brock et al is not referenced in figure legend and we discourage re-display of published data - a specific reference will suffice. (ii) figure cell reuse between Appendix Figure S12A and Appendix Figure S13F
- 2) ref 1's concern about cleave plane re-orientation in mammals: please ensure the arguments made in your response to the referee is included in the revised manuscript in the same level of detail. regarding the statement 'These irregularities could contribute to the abnormal cleavage plane orientation of BWCFF-S mitotic APs (see discussion for details'. As the referee notes, this remains a 'weak argument'. In our view it remains a descriptive, albeit reproducible, observation and should be clearly described as such.
- 3) ref 1: re. the statement 'these findings suggest that BPs are produced through increased delamination at the expense of VZ progenitors, the primary progenitor type in the developing brain, which ultimately results in reduced neuronal output. This likely represents the underlying cause of the patients' microcephaly'. Without adding new data to formally test this, please do not claim more than 'this is one mechanism that would be consistent with microcephaly' or similar.
- 4) ref 3, point 3: please add PAX6 or SOX2 staining, as requested in the first round of peer review.
- 5) ref 3, point 4: quantification of fig 5 EM: this was declined for unspecified technical reasons. Please specify the reason and minimally add replicate data as part of the source data upload on BioStudies.
- 6) ref 3, point 5: Please add the requested representative image 5H.
- 7) ref 3, point 6: please add the discussion on the actin isoforms and mutants to the discussion in as far as this is not already included.
- 8) Please address the following formatting issues, in as far as they are not already addressed in the revised manuscript:
 - Rename Competing interests: to Disclosure and Competing Interests Statement
 - Rename Materials and methods to Methods
 - Remove Author Contributions from the manuscript text as you had already provided roles and contributions for each author upon manuscript submission in the system
 - References: insert et al. after the 10th author name if a reference has more than 10 authors
 - Co-corresponding author, Nataliya Di Donato, has been sent a request to link her ORCID in our system as we cannot do this on authors' behalf for security reasons
 - Remove the Funding section heading and include the funding information in the Acknowledgments section
 - Figure files: please try resizing the figure files as currently, the Merged PDF is ~552 MB (e.g. Figure EV5 is 116 MB). While we have not size limit, we encourage limiting individual figure to 50MB in size.
 - Panels of Figure 1A-C need to be clearly labeled (should be called out as "Figure1A-C" instead of just "A-C"); "supplementary" should not be used, this needs to be updated to the current nomenclature or just removed from the text
 - Movies: movie legends need to be removed from the manuscript and each should be zipped up with its corresponding movie so that we have 6 zip folders: Movie EV1-EV6; Source files of the movies need to be updated to Movie EV1, etc. (nomenclature Movie S1, etc. should not be used)
 - Please note that the exact p values are not provided in the legends of figures 2C, D; 4B, C, D, E, F, I; 5G, H; 7B, C, D, E, F, G, I
 - COI/DCIS: needs to be renamed to Disclosure and Competing Interests Statement
 - REFERENCES: et al is missing in the references - needs to be inserted after the 10th author name
 - ORCID ID: missing for Donato
 - FUNDING INFO: needs to be part of the Acknowledgments (Funding section heading should be removed)
 - FIGURE CALLOUTS: panels of Figure 1A-C are not clearly labeled (should be called out as "Figure1A-C" instead of just "A-C"); "supplementary" should not be used, this needs to be updated to the current nomenclature or just removed from the text
 - Materials and methods should be Methods

A final decision will depend on successfully addressing all the above points in revision.

I look forward to seeing a new revised version of your manuscript in due course.

Thanks you again for your patience in this protracted manuscript evaluation.

Yours sincerely,

Bernd Pulverer

Referee #3:

The authors have addressed some, but not all, of the comments in this revised manuscript. The improvement to the original manuscript - regarding our comments - is moderate.

Here are our comments on the 6 major points we had made in the first revision.

Point 1/ We appear to disagree with the authors regarding our first comment on experimental replicates. Pooling different cell lines coming from different individuals is not a proper way of reporting data. Each line is a different condition. We furthermore disagree with the authors when they claim that individual organoids can be considered as experiments. A different experiment is a different batch of organoid, not different organoids from the same batch, coming from the same well, imaged on the same day...

Point 2/ The authors have added some cells to the spindle orientation analysis.

Point 3/ We are still not convinced that DAPI is reliable to identify the VZ and the authors did not modify their method. However, the apical surface measurements are a nice addition.

Point 4/ The authors still do not quantify this EM data. They claim it was not possible without giving any explanation. It gives the feeling the experiment was only done once, which obviously would not be acceptable.

Point 5/ The authors now quantify Tbr2 cells, which is a very nice addition. Could they add a representative image?

Point 6/ We asked for Western blots, which the authors present is figure Apendix S10.

Response to Editor

Editor's Comment:

Thank you for the submission of your revised manuscript and for your patience. As discussed, we have only been able to obtain a re-review of referee 3, which is attached below. Thank you for your initial response to the report and the reuse issues raised below. We have reviewed the responses to all referee points in detail and consequently invite a second revision to address the following points:

Authors' response:

We thank the Editor for the opportunity to submit a second revision and for the consideration of our manuscript. We have carefully addressed all points raised, as described in detail below, and have implemented the requested changes throughout the revised manuscript.

Editor's Comment:

1) Two figure panel reuse issues must be addressed:

(i) Figure reuse between - Figure 1 A and B and published manuscript - Figure 1 L and M - <https://doi.org/10.1038/s41431-018-0146-y> - European Journal of Human Genetics (2018) 26:1132-1142. Brock et al is not referenced in figure legend and we discourage re-display of published data - a specific reference will suffice.

(ii) figure cell reuse between Appendix Figure S12A and Appendix Figure S13F

Authors' response:

We apologize for these oversights. In the revised manuscript, we have cited Brock et al., 2018 in the Figure 1 legend, and we have replaced the incorrect karyogram in Figure S13F with the correct one.

Editor's Comment:

2) ref 1's concern about cleave plane re-orientation in mammals: please ensure the arguments made in your response to the referee is included in the revised manuscript in the same level of detail. regarding the statement 'These irregularities could contribute to the abnormal cleavage plane orientation of BWCF-F-S mitotic APs (see discussion for details'. As the referee notes, this remains a 'weak argument'. In our view it remains a descriptive, albeit reproducible, observation and should be clearly described as such.

Authors' response:

We have incorporated all the arguments from our response to the referee into the revised manuscript at the same level of detail. Following the Editor's recommendation, we have also revised the text to describe this finding as a reproducible observation, making clear that it remains descriptive.

Editor's Comment:

3) ref 1: re. the statement 'these findings suggest that BPs are produced through increased delamination at the expense of VZ progenitors, the primary progenitor type in the developing brain, which ultimately results in reduced neuronal output. This likely represents the underlying cause of the patients' microcephaly'. Without adding new data to formally test this, please do not claim more than 'this is one mechanism that would be consistent with microcephaly' or similar.

Authors' response:

Following the Editor's recommendation, we have weakened the corresponding text in the conclusion and revised the wording to state that this is one mechanism consistent with microcephaly, as suggested.

Editor's Comment:

4) ref 3, point 3: please add PAX6 or SOX2 staining, as requested in the first round of peer review.

Authors' response:

As requested, we have added images of SOX2 staining to Figure 4 to address the Reviewer's comment.

Editor's Comment:

5) ref 3, point 4: quantification of fig 5 EM: this was declined for unspecified technical reasons. Please specify the reason and minimally add replicate data as part of the source data upload on BioStudies.

Authors' response:

We thank the reviewer for this suggestion. The 3D architecture of the apical-most neuroepithelium in both wildtype and mutant organoids is highly complex, making a reliable quantification of cell circumference or the attachment of cytoskeletal elements to adherens junctions impossible from single 2D sections. Accurate assessment would require volume imaging, as the orientation of individual EM sections (e.g., diagonal vs. transverse cuts) and membrane folds cannot be determined with certainty. Moreover, the resolution of the tomograms is, in some cases, insufficient to clearly distinguish individual actin filaments (5 nm) or microtubules (20 nm) within a 300-nm section. For these reasons, a meaningful quantification of the EM data shown in Figure 6 (previously Figure 5) is not technically feasible and is beyond the scope of the current manuscript. We have added this explanation to the Methods section of the revised manuscript.

Importantly, this was not a single observation—we observed the same feature in multiple cerebral organoids—and, as requested, we have now included replicate data from three ACTB Thr120Ile and four ACTG1 Thr203Met organoids as part of the source data uploaded to BioStudies.

Editor's Comment:

6) ref 3, point 5: Please add the requested representative image 5H.

Authors' response:

Following the Reviewer's comment we have added representative images of TBR2 staining as Figure 5I to the revised version of the manuscript.

Editor's Comment:

7) ref 3, point 6: please add the discussion on the actin isoforms and mutants to the discussion in as far as this is not already included.

Authors' response:

We have added a separate paragraph to the Discussion in the revised manuscript addressing the nature of the ACTB and ACTG1 mutations and the functional relationship between the actin isoforms.

Editor's Comment:

8) Please address the following formatting issues, in as far as they are not already addressed in the revised manuscript:

- Rename Competing interests: to Disclosure and Competing Interests Statement
- Rename Materials and methods to Methods
- Remove Author Contributions from the manuscript text as you had already provided roles and contributions for each author upon manuscript submission in the system
- References: insert *et al.* after the 10th author name if a reference has more than 10 authors
- Co-corresponding author; Nataliya Di Donato, has been sent a request to link her ORCID in our system as we cannot do this on authors' behalf for security reasons
- Remove the Funding section heading and include the funding information in the Acknowledgments section
- Figure files: please try resizing the figure files as currently, the Merged PDF is ~552 MB (e.g. Figure EV5 is 116 MB). While we have not size limit, we encourage limiting individual figure to 50MB in size.
- Panels of Figure 1A-C need to be clearly labeled (should be called out as "Figure1A-C" instead of just "A-C"); "supplementary" should not be used, this needs to be updated to the current nomenclature or just removed from the text
- Movies: movie legends need to be removed from the manuscript and each should be zipped up with its corresponding movie so that we have 6 zip folders: Movie EV1-EV6; Source files of the movies need to be updated to Movie EV1, etc. (nomenclature Movie S1, etc. should not be used)
- Please note that the exact *p* values are not provided in the legends of figures 2C, D; 4B, C, D, E, F, I; 5G, H; 7B, C, D, E, F, G, I
- COI/DCIS: needs to be renamed to Disclosure and Competing Interests Statement
- REFERENCES: *et al* is missing in the references - needs to be inserted after the 10th author name
- ORCID ID: missing for Donato
- FUNDING INFO: needs to be part of the Acknowledgments (Funding section heading should be removed)
- FIGURE CALLOUTS: panels of Figure 1A-C are not clearly labeled (should be called out as "Figure1A-C" instead of just "A-C"); "supplementary" should not be used, this needs to be updated to the current nomenclature or just removed from the text Materials and methods should be Methods

Authors' response:

All requested formatting changes have been implemented in the revised manuscript, including section renaming, removal of Author Contributions, reference style updates, relocation of funding information, resizing of figures, correction of figure and movie labeling, addition of exact *p*-values, and confirmation of ORCID linking for N. Di Donato.

Editor's Comment:

A final decision will depend on successfully addressing all the above points in revision.

Authors' response:

We have carefully addressed all of the above points and hope that the revised manuscript will be considered acceptable for publication. We thank the Editor and Reviewers for their careful evaluation and constructive guidance.

Dear Dr. Heide,

I yet again want to thank you for your considerable patience, as well as for your detailed comments to guide through the revision.

I am pleased to inform you that your manuscript has now been accepted for publication in EMBO reports, following careful review of the revision. Your manuscript will be processed for publication by EMBO Press. It will be copy edited and you will receive page proofs prior to publication. Please note that you will be contacted by Springer Nature Author Services to complete licensing and payment information.

You may qualify for financial assistance for your publication charges - to check your eligibility:
<https://www.embopress.org/page/journal/14693178/authorguide#chargesguide>

If you have any questions, please do not hesitate to contact the Editorial Office. Thank you for your interesting contribution to EMBO Reports.

best wishes,

Bernd Pulverer
